# Why Do We Need Warm-up? A Theoretical Perspective

## Abstract

Learning rate warm-up – increasing the learning rate at the beginning of training – has become a ubiquitous heuristic in modern deep learning, yet its theoretical foundations remain poorly understood. In this work, we provide a principled explanation for why warm-up improves training. We rely on a generalization of the $(L_0, L_1)$-smoothness condition, which bounds local curvature as a linear function of the loss sub-optimality and exhibits desirable closure properties. We demonstrate both theoretically and empirically that this condition holds for common neural architectures trained with mean-squared error and cross-entropy losses. Under this assumption, we prove that Gradient Descent with a warm-up schedule achieves faster convergence than with a fixed step-size, establishing upper and lower complexity bounds. Finally, we validate our theoretical insights through experiments on language and vision models, confirming the practical benefits of warm-up schedules.

## 1 Introduction

Training modern machine learning models requires a careful choice of hyperparameters. A common practice for setting the learning rate (LR) is to linearly increase the LR in the beginning (*warm-up stage*) (Goyal et al., 2017; Vaswani et al., 2017) and gradually decrease at the end of the training (*decay stage*) (Loshchilov & Hutter, 2016; Vaswani et al., 2017; Hoffmann et al., 2022b; Zhang et al., 2023; Dremov et al., 2025).

Decaying the LR is a classical requirement in the theoretical analysis of `SGD`, ensuring convergence under broad conditions (Defazio et al., 2023; Gower et al., 2021), and it has been consistently observed to improve empirical performance (Loshchilov & Hutter, 2016; Hu et al., 2024; Hägele et al., 2024). Recent work further demonstrates that *decaying* step sizes can improve theoretical guarantees by yielding tighter bounds (Schaipp et al., 2025). By contrast, the practice of linearly increasing the LR at the start of training (warm-up phase) has become nearly ubiquitous in modern deep learning (He et al., 2016; Hu et al., 2024; Hägele et al., 2024), yet a clear theoretical understanding of why it helps optimization remains elusive. This raises the central question we address in this paper:

> *Why does LR warm-up improve training, and under what conditions can its benefits be theoretically justified?*

A growing body of empirical work points to several advantages of warm-up, including: (*i*) mitigating training instabilities (Kosson et al., 2024; Goyal et al., 2017; Zhang et al., 2023), reducing the variance of stochastic gradients (Liu et al., 2019), and improving the robustness to the choice of the peak LR (Wortsman et al., 2023; Kalra & Barkeshli, 2024). However, these explanations remain fragmented and do not clarify why warm-up is effective, nor to what extent it is actually necessary.

In order to provide a theoretical justification for warm-up, we will rely on a special smoothness condition that relates curvature to sub-optimality. We then demonstrate how this condition naturally provides an explanation for the benefits of warm-up schedules. Specifically, we make the following contributions:

1. We discuss a natural extension of $(L_0, L_1)$-smoothness, which we call $(H_0, H_1)$-smoothness, where the local smoothness is bounded by a linear function of the loss sub-optimality. This extension enjoys desirable properties such as closeness under finite sums and affine transformations.

2. We provide both theoretical and empirical evidence that the $(H_0, H_1)$-smoothness condition holds for various neural network architectures trained with mean-squared error (MSE) and cross-entropy (CE) losses.

3. We theoretically demonstrate that, in the function class defined by our proposed condition, Gradient Descent (GD) achieves faster convergence with a warm-up step-size than with a fixed step-size. We do that by obtaining both upper complexity bounds for GD with a warm-up step-size and lower complexity bounds for GD with a fixed step size.

4. Finally, we provide empirical evidence that the theoretical warm-up scheme is also useful in training language and vision models.

## 2 RELATED WORKS

**Warm-up.** LR scheduling plays a central role in the success of modern deep learning training pipelines. A wide range of scheduling strategies, including LR decay, annealing, and warm-up, have been developed to improve convergence and generalization (McCandlish et al., 2018; Sutskever et al., 2013; Touvron et al., 2023).

Among these different strategies, warm-up has become a key component in modern training pipelines, particularly for Transformers (Vaswani et al., 2017; Goyal et al., 2017). It is commonly credited with enhancing training stability (Kosson et al., 2024; Gotmare et al., 2018), improving robustness to the choice of LR (Wortsman et al., 2023), and enabling the use of larger peak LR (Kalra & Barkeshli, 2024). Warm-up has also been linked to improved generalization, either by reducing mini-batch gradient noise (Liu et al., 2019), encouraging convergence to flatter minima (Smith et al., 2020), or by complementing other scheduling techniques (Huang et al., 2020; Xiong et al., 2020; Wortsman et al., 2023). From a geometric perspective, Gilmer et al. (2021); Roulet et al. (2024) observed that warm-up induces a sharpness reduction phase in which the largest Hessian eigenvalue decreases.

Although warm-up is well supported by empirical evidence (Vaswani et al., 2017; Wortsman et al., 2023; Dremov et al., 2025), its theoretical foundations remain limited. Most existing convergence analyses of (stochastic) gradient-based optimizers focus on the decay phase. For example, Wen et al. (2024) uses a river-valley model to study neural loss landscapes, but their framework focuses on the stable and decay stages of the LR. Likewise, Schaipp et al. (2025); Attia & Koren (2025) showed that decaying LR provides theoretical benefits and that convergence bounds closely align with empirical training curves, yet their analysis does not account for the warm-up phase. Kondo & Iiduka (2025) analyze a scheme with exponentially increasing batch size and LR, showing faster convergence for gradient descent (GD). Yet, the requirement of rapidly growing batches limits its practicality.

Finally, several complementary explanations for the role of warm-up have been proposed. For instance, Xiong et al. (2020) attribute the necessity of warm-up in Transformer training primarily to the placement of layer normalization. In a different vein, Kosson et al. (2024) demonstrate that explicitly constraining the norm of parameter updates—similar to gradient clipping—can only partially reduce the reliance on warm-up.

Despite extensive prior research on warm-up, we are not aware of any theoretical framework that explains its benefits in terms of convergence. In this work, we address this gap by relying on a smoothness-type condition that upper bounds the curvature of the landscape using an affine expression of the function sub-optimality. Training under such condition turns out to be benefited by LR warm-up.

**Generalized Smoothness.** The conventional smoothness assumption in optimization theory requires the Hessian to satisfy a uniform bound $\|\nabla^2 f(w)\| \leq L$, but this constraint proves to be overly restrictive when applied to neural network training, as noted

by Zhang et al. (2019). To address this limitation, they introduced the more flexible $(L_0, L_1)$-smoothness condition, which allows the Hessian norm to grow linearly with the gradient magnitude: $\|\nabla^2 f(w)\| \leq L_0 + L_1 \|\nabla f(w)\|$ for non-negative constants $L_0, L_1 \geq 0$. This relaxed framework naturally motivates gradient normalization techniques—both soft normalization and hard clipping—as optimal LR strategies that can significantly improve gradient descent convergence rates (Zhang et al., 2020; Zhao et al., 2021; Faw et al., 2023; Wang et al., 2023; Gorbunov et al., 2024; Vankov et al., 2024; Li et al., 2023).

Despite its advantages, the $(L_0, L_1)$-smoothness condition suffers from several shortcomings that limit its practical applicability, especially in explaining warm-up schedules. From a theoretical perspective, the class of $(L_0, L_1)$-smooth functions does not possess the closeness property under fundamental operations such as summation and affine transformations (see Section 3). Since these operations are ubiquitous in neural network architectures, this limitation restricts the framework's general applicability.

More problematically, at the beginning of training, the gradient-dependent nature of the $(L_0, L_1)$-smoothness condition leads to counterintuitive implications for LR scheduling. In some cases, the gradient norm is observed to increase during the early iterations (Xie et al., 2023; Defazio et al., 2023; Defazio, 2025). As a result, the $(L_0, L_1)$-bound becomes increasingly loose, which theoretically prescribes *decreasing* step sizes through gradient clipping. This stands in direct contrast to empirical best practices, where *increasing* LR are typically employed at the beginning of training. We emphasize that this issue is specific to the beginning of training; beyond the warm-up phase, decreasing step sizes is consistent with the theoretical condition.

These theoretical and practical inconsistencies highlight the need for a more sophisticated smoothness characterization that can adequately capture and explain LR warm-up dynamics. Since the gradient norm is problematic in the $(L_0, L_1)$-smoothness condition, a natural candidate to replace it is the function value sub-optimality, which decays monotonically and gives a direct measure of the optimization target. We name this modified smoothness class as $(H_0, H_1)$-smoothness. Interestingly, a recent work by Vaswani & Babanezhad (2025) made a similar observation in a different context, showing that Armijo line search can achieve faster convergence than GD with a constant step-size. Their analysis verifies this condition for several simple models but relies on additional assumptions from Taheri & Thrampoulidis (2023): (*i*) bounding the gradient norm by the function sub-optimality, (*ii*) adopting the unrealistic exponential loss, (*iii*) assuming data separability, and (*iv*) restricting trainability to the input layer. In contrast, our analysis establishes the validity of the $(H_0, H_1)$-smoothness condition under a mild regularity assumption on the weights, which can be ensured either implicitly through gradient-based optimization or explicitly via standard L2 regularization. Although this work is not the first to propose extending $(L_0, L_1)$-smoothness, we go beyond prior work to demonstrate the applicability of this condition when training neural networks (see Section 3) and by establishing key properties of these functions (see Appendix B). [1]

## 3 The $(H_0, H_1)$-smoothness condition

Building on our observation that function value sub-optimality is more suitable than the gradient norm to measure curvature, we will focus on the following smoothness condition.

**Definition 3.1.** *A function $f \colon \mathbb{R}^d \to \mathbb{R}$ with minimum $f^* > -\infty$ is called $(H_0, H_1)$-smooth for some $H_0, H_1 \geq 0$, if for any $w \in \mathbb{R}^d$ we have*

$$\|\nabla^2 f(w)\|_2 \leq H_0 + H_1(f(w) - f^*).$$

$\mathcal{H} := \{f \colon \mathbb{R}^d \to \mathbb{R} \mid f \text{ is } (H_0, H_1)\text{-smooth}\}$ *denotes the class of all $(H_0, H_1)$-smooth functions.*

Based on simple derivations, we can check that any $(L_0, L_1)$-smooth function also satisfies $(H_0, H_1)$-smoothness. Hence, the $(H_0, H_1)$-smoothness class contains the previously studied $(L_0, L_1)$-smooth class. In addition, we show that $\mathcal{H}$ is closed under finite sums and affine

---

[1] A recent work (Liu et al., 2025) that appeared online on 09.09.2025 studies a warm-up stage using a similar condition. We discuss the differences in Appendix A.

transformations, in contrast to the $(L_0, L_1)$-smooth class, for which simple counterexamples demonstrate that neither operation is preserved. Formal statements and proofs of the aforementioned claims are deferred to Appendix B. Finally, Definition 3.1 admits a natural extension in which the linear dependence on sub-optimality $f(w) - f^*$ is replaced by any monotone increasing function $\mathcal{L}$ of $f(w) - f^*$, in the spirit of Li et al. (2023). We leave the study of this generalization to future work.

## 3.1 Theoretical Justification of $(H_0, H_1)$-smoothness

### 3.1.1 Standard Feedforward Networks

In this section, we demonstrate that under mild regularity conditions on the weights – enforced either implicitly by constraining the weight space or explicitly via L2 regularization – the $(H_0, H_1)$-smoothness condition holds for a range of basic deep learning architectures. Detailed proofs are provided in Appendix C.

**Results under Balancedness.** A known property of gradient flow in feedforward neural networks is that the weight matrices $\{W_i\}_{i=1}^{\ell}$ evolve in a balanced manner, satisfying $W_i(t)^\top W_i(t) = W_{i+1}(t)W_{i+1}(t)^\top$ for linear networks and $\|W_i(t)\|_{\mathrm{F}} = \|W_{i+1}(t)\|_{\mathrm{F}}$ for non-linear networks (Du et al., 2018, Theorem 2.2 and Corollary 2.1). Note that the second property is weaker than the first. The "strong" balancedness property holds even in non-linear networks if the activation between the layers $W_i$ and $W_{i+1}$ is linear.

**Proposition 3.1.** *Consider a deep linear network with $\ell$ layers and MSE loss:*

$$f(W) \equiv f(W_1, \ldots, W_\ell) = \|Y - W_1 W_2 \ldots W_\ell X\|_{\mathrm{F}}^2,$$

*where $Y \in \mathbb{R}^{c \times m}$ are the labels, $X \in \mathbb{R}^{d \times m}(d \leq m)$ is the input, and $W_i \in \mathbb{R}^{n_{i-1} \times n_i}$, where $n_0 = c$ and $n_\ell = d$ are networks' weights. In the space of strongly balanced weights, i.e., when $W_i^\top W_i = W_{i+1}W_{i+1}^\top$ for all $i \in [\ell - 1]$, it holds that*

$$\|\nabla^2 f(W)\|_2 \leq H_0 + H_1(f(W) - f^*),$$

*where exact forms of $H_0$ and $H_1$ are provided in equations (5) and (6) in the Appendix.*

We further discuss the case of deep non-linear networks with only one leaky ReLU non-linearity preceding the output layer.

**Proposition 3.2.** *Let $f$ be defined as*

$$f(W) \equiv f(W_1, \ldots, W_\ell) = \|Y - W_1 \phi(W_2 X_3 \ldots W_\ell X)\|_{\mathrm{F}}^2$$

*where $\phi$ is leaky-ReLU activation function with slopes 1 and $b$, i.e., $\phi(x) = \max\{bx, x\}$, $0 < b \leq 1$, and matrices $Y, X, \{W_i\}_{i=1}^{\ell}$ defined as before. Assume that over the course of GD:*
- *$\lambda_{\min}(W_1^\top W_1) \geq h > 0$.*
- *The layers $\{W_i\}_{i=1}^{\ell}$ are weakly balanced, i.e., $\|W_1\|_{\mathrm{F}} = \ldots = \|W_\ell\|_{\mathrm{F}}$.*
- *The layers $\{W_i\}_{i=2}^{\ell}$ are strongly balanced, i.e., $W_i^\top W_i = W_{i+1}W_{i+1}^\top$, for $i \in \{2, \ldots, \ell\}$.*
*Then it holds that*

$$\|\nabla^2 f(W)\|_2 \leq H_0 + H_1 f(W) \ \ (= (H_0 + H_1 f^*) + H_1(f(W) - f^*)),$$

*where the exact forms of $H_0$ and $H_1$ are provided in equations (17) and (18) in the Appendix.*

In the Appendix, we present a generalization of Proposition 3.2 in the case that the network has $(\ell - 1)$ non-linearities (Proposition C.1). In this case though, we need to raise $f(W) - f^*$ to a power depending on the depth of the network. We can still use our theory to explain the benefit of warm-up even in this case, as explained in Appendix A (see equation (3)).

**Results under L2 Regularization.** Analogous to balancedness, another approach to constraining the weight space is through L2 regularization. In this section, we present results that validate the $(H_0, H_1)$-smoothness condition for two-layer neural networks with general activation functions, considering both MSE and cross-entropy losses under L2 regularization.

**Proposition 3.3.** *Consider a 2-layer neural network with MSE loss and L2 regularization:*

$$f(W) \equiv f(W_1, W_2) = \|Y - W_1\phi(W_2 X)\|_{\mathrm{F}}^2 + \frac{\lambda_1}{2}\|W_1\|_{\mathrm{F}}^2 + \frac{\lambda_2}{2}\|W_2\|_{\mathrm{F}}^2,$$

*where $\phi$ is an activation function, such that $|\phi(x)| \leq C_1|x|, |\phi'(x)| \leq C_2$ and $|\phi''(x)| \leq C_3$ for all $x \in \mathbb{R}$, and matrices $Y, W_1, W_2$ are defined as before. Then, it holds*

$$\|\nabla^2 f(W)\|_2 \leq H_0 + H_1 f(W) \ (= H_0 + H_1 f^* + H_1(f(W) - f^*)),$$

*for $H_0$ and $H_1$ defined as in equations (31) and (32) respectively.*

We conclude our discussion of this section with the case of binary classification.

**Proposition 3.4.** *Consider a 2-layer non-linear model with cross-entropy loss and L2 regularization:*

$$f(W) \equiv f(W_1, W_2) = -Y\log(P)^\top - (\mathbb{1} - Y)\log(\mathbb{1} - P)^\top + \frac{\lambda_1}{2}\|W_1\|_{\mathrm{F}}^2 + \frac{\lambda_2}{2}\|W_2\|_{\mathrm{F}}^2,$$

*where $Y \in \mathbb{R}^{1 \times m}$ are true labels, and $P = \sigma(W_1\phi(W_2 X))$ is the output of the model with the activation function $\phi$ such that $|\phi(x)| \leq C_1|x|, |\phi'(x)| \leq C_2$ and $|\phi''(x)| \leq C_3$ for all $x \in \mathbb{R}$, sigmoid function $\sigma$, and weight matrices $W_1 \in \mathbb{R}^{1 \times n_1}, W_2 \in \mathbb{R}^{n_1 \times d}$. Then, it holds*

$$\|\nabla^2 f(W)\|_2 \leq H_0 + H_1 f(W) \ (= H_0 + H_1 f^* + H_1(f(W) - f^*))$$

*for $H_0$ and $H_1$ defined as in equations (36) and (37) respectively.*

**Remark 3.1.** *The results of Propositions 3.3 and 3.4 can be extended to a more general class of activations that satisfy $|\phi(x)| \leq C_0 + C_1|x|$, which covers more practical examples such as sigmoid.*

In Appendix E, we show that $(L_0, L_1)$-smoothness fails to hold even for simple two-layer networks under L2 regularization or weight balancedness, thereby highlighting its limitations in capturing the loss landscape of neural networks.

### 3.1.2 TRANSFORMERS

In this section, we study a simple transformer architecture with a single attention layer trained under L2 regularization, following the setup of Zhang et al. (2024). We show that its in-context loss function is $(H_0, H_1)$-smooth.

The input data is encoded into a single matrix $Z_0 \in \mathbb{R}^{(d+1)\times(n+1)}$. This matrix contains $n$ training tokens and one query token. The training tokens cover the first $n$ columns of the matrix, while the query token the last one. The label of the query's feature is initialized at 0.

$$Z_0 = \begin{bmatrix} x_1 & x_2 & \dots & x_n & x_{\mathrm{query}} \\ y_1 & y_2 & \dots & y_n & 0 \end{bmatrix} \in \mathbb{R}^{(d+1)\times(n+1)}.$$

The model's objective is to predict the true value for this entry. The model is defined by $Z_1 = Z_0 + \frac{1}{n}PZ_0 M \cdot \phi(Z_0^T Q Z_0)$, where the trainable parameters are $P$ and $Q$ ($Q$ is a re-parametrization of the standard Key and Query matrices). $M = \mathrm{diag}(1, 1, \dots, 1, 0)$ is a fixed mask and $\phi$ is a general activation applied to the attention scores. The output of the model is $\hat{y} = [Z_1]_{(d+1),(n+1)}$ and the cost function for one task with true target $y_{\mathrm{true}}$ is $f(P, Q) = \ell(\hat{y}, y_{\mathrm{true}})$, where $\ell$ is some loss function (MSE for continuous variables or cross-entropy for binary ones). The most interesting property of transformers is their ability to learn in-context, i.e., minimize an in-context cost function defined below.

**Definition 3.2.** *Let $D_x$ be a distribution over an input space $X$, $H$ a set of functions $X \to Y$, and $D_H$ a distribution over functions in $H$. Let $\ell : Y \times Y \to \mathbb{R}$ be a loss function, $S = \{(x_1, y_1, \dots, x_n, y_n) : x_i \in X, y_i \in Y\}$ be the set of finite-length sequences of $(x, y)$ pairs and $F_\theta = \{f_\theta : S \times X \to Y, \theta \in \Theta\}$ be a class of functions parameterized by $\theta$ in some set $\Theta$. For $n > 0$, we say that a model $f : S \times X \to Y$ is trained on in-context examples of functions in $H$ under loss $\ell$ w.r.t. $(D_x, D_H)$ if $f = f_\theta$, where $\theta$ satisfies*

$$\theta \in \mathrm{argmin}_{\theta \in \Theta} \mathbb{E}_{j=(x_1, h(x_1), \dots, x_n, h(x_n), x_{query})}[\ell(f_\theta(j), h(x_{query}))]$$

*where $x_i, x_{query}$ are chosen i.i.d. from $D_x$ and $h \sim D_H$ is independent. $j$ represents a prompt.*

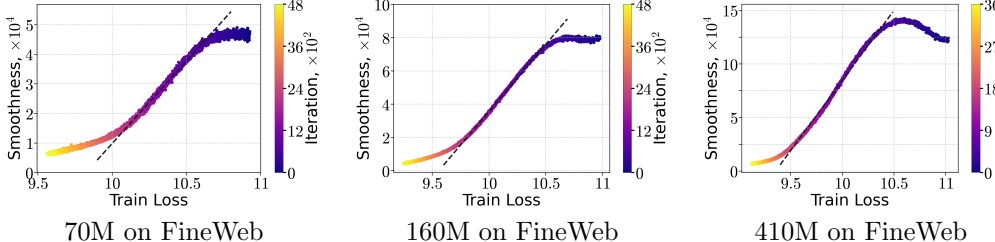

Figure 1: Local smoothness approximation versus training loss for language models of varying sizes on the FineWeb dataset, using `SGD` at a constant LR of $10^{-4}$. Each dot represents estimated local smoothness and stochastic training loss, with color indicating training progress, while the black dashed line shows the best linear fit. For much of early training, the relation is well-approximated by a line, aside from the very initial phase where smoothness behaves differently. This deviation likely arises because the linear fit reflects only an upper bound, suggesting that a more complex functional dependence may be necessary.

The following result holds under mild conditions on the distribution of the training tokens:

**Proposition 3.5.** *Consider the transformer model as described before, for in-context learning in continuous variables with MSE loss, or in binary variables with CE loss. $\phi$ is assumed to satisfy the conditions of Proposition 3.4 and $f_j(P,Q)$ is the L2 regularized loss corresponding to the j-th prompt. Consider the regularized in-context loss function*

$$f(P,Q) = \mathbb{E}_j[f_j(P,Q)].$$

*Assume that $D_x$ is sub-gaussian and the distribution of $y = h(x)$ is sub-exponential. Then, it holds*

$$\|\nabla^2 f(P,Q)\|_2 \leq H_0 + H_1 f(P,Q)(= H_0 + H_1 f^* + H_1(f(P,Q) - f^*))$$

*for some positive and finite constants $H_0$ and $H_1$.*

The proof can be found in Appendix D.

### 3.2 Empirical Justification of $(H_0, H_1)$-smoothness

We next turn to verifying the proposed condition in practical settings. Specifically, we examine Transformer-based language models with 70M, 160M, and 410M parameters trained using the NanoGPT implementation (Radford et al., 2019; Karpathy, 2022). Experiments are carried out on the FineWeb dataset (Penedo et al., 2024) with `SGD` and a small constant LR of $10^{-4}$. Using such a conservative LR allows the optimizer to progress slowly, thereby probing the landscape around initialization in more detail. To approximate the local smoothness at iteration $k$, we compute $\frac{\|\nabla f_{S_k}(w_{k+1}) - \nabla f_{S_{k-1}}(w_k)\|}{\|w_{k+1} - w_k\|}$, where $S_k$ denotes the mini-batch at iteration $k$, following prior work (Zhang et al., 2019; Riabinin et al., 2025). As shown in Figure 1, the estimated smoothness decays approximately linearly, indicating that the proposed condition provides a reasonable smoothness approximation for real-world models. The only exception is a brief initial phase where the trend deviates from linearity, likely because the condition acts as an upper bound, implying that a more expressive functional form may be needed to describe the behavior fully.

We next turn to image classification on ImageNet32 (Chrabaszcz et al., 2017), training both ResNet50 (He et al., 2016) and ViT-Tiny (Dosovitskiy et al., 2020). The results, shown in Figure 2, indicate that a linear function provides a good approximation of the relationship between local smoothness and training loss. Compared to language models, however, the points are more widely dispersed and have larger variance. Taken together, Figures 1 and 2 support the view that $(H_0, H_1)$-smoothness offers a reasonable approximation of smoothness in the early stages of training.

An important point to notice in both Figures 1 and 2 is that by selecting a sufficiently small learning rate, we restrict `SGD` to the sharpness-reduction phase identified in prior work (Kalra & Barkeshli, 2024; Kalra et al., 2023). Our condition provides an accurate characterization of this phase, as the smoothness constant decreases roughly linearly with the loss. Once this

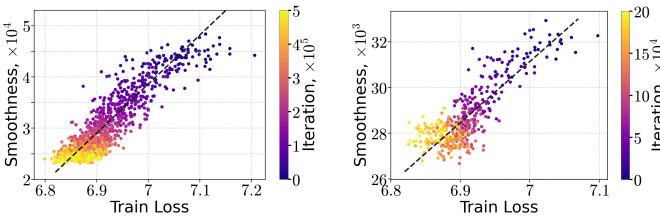

Figure 2: Local smoothness approximation against train loss during training a ResNet50 (left) and ViT-Tiny (right) on ImageNet32, using `SGD` with a constant LR $10^{-4}$.

phase concludes, `SGD` transitions into the progressive sharpening phase (Cohen et al., 2021), during which the smoothness constant rises again; see Figure K.1. In the latter regime, our condition no longer reliably predicts smoothness behavior (it applies only to the reduction phase).

## 4 THEORETICAL ANALYSIS UNDER $(H_0, H_1)$-SMOOTHNESS

We study the minimization problem $\min_w f(w)$, which appears in various machine learning applications. Here $w \in \mathbb{R}^d$ denotes parameters of some model, $d$ is the number of parameters, and $f$ is the loss that measures the performance. We define $f^* := \min_w f(w) > -\infty$ as the optimal loss. The set $\mathcal{S}$ contains all global minimizers of the objective $f$. The proofs of this section are deferred to Appendix G and I.

### 4.1 NOTATION AND ASSUMPTIONS

We conduct our analysis for well-known classes of non-convex functions, presented below.

**Definition 4.1** (Liu et al. (2023)). *A function $h$ satisfies the Aiming condition with a constant $\theta > 0$ around the set $\mathcal{X}$, if $\langle \nabla h(w), w - \pi_\mathcal{X}(w) \rangle \geq \theta(h(w) - h^*)$ holds for all $w \in \mathbb{R}^d$. Here, $\pi_\mathcal{X}(w)$ is the projection of $w$ onto the set $\mathcal{X}$, and $h^* := \min_{w \in \mathbb{R}^d} h(w)$.*

**Definition 4.2** (Polyak (1963)). *A function $h$ satisfies Polyak-Łojasiewicz (PL) condition with a constant $\mu > 0$, if $\|\nabla h(w)\|^2 \geq 2\mu(h(w) - h^*)$ holds for all $w \in \mathbb{R}^d$.*

### 4.2 LOWER BOUNDS AND CONVERGENCE OF GD WITH CONSTANT STEP-SIZE

To enable a meaningful comparison between the step-size schedule suggested by the $(H_0, H_1)$-condition and an alternative fixed step-size strategy, we derive lower complexity bounds for the latter. The approach follows the idea of Theorem 4 in Zhang et al. (2019): we first consider a rapidly growing function and show that, for `GD` to converge, the step-size must be sufficiently small. Next, we examine a slowly growing function and demonstrate that this previously derived step-size constraint leads to slow convergence of the algorithm. The complete proof can be found in Appendix I.

**Theorem 4.1.** *Let $f$ belong to the class $\mathcal{H}$ of $(H_0, H_1)$-smooth functions. Then it holds:*

1. *To satisfy $\|\nabla f(w_K)\| \leq \varepsilon$ for a general non-convex function $f$, `GD` with constant step-size initialized at $w_0$, needs at least*

$$K \geq \frac{H_1(f(w_0) - f^*)}{\log(f(w_0) - f^*) + 1} \frac{f(w_0) - f^* - 2\epsilon^2}{8\epsilon^2} \quad \text{iterations.}$$

2. *To satisfy $f(w_K) - f^* \leq \varepsilon$ for convex function $f$, `GD` with constant step-size initialized at $w_0$, needs at least*

$$K \geq \frac{H_1(f(w_0) - f^*)}{\log(f(w_0) - f^*) + 1} \frac{f(w_0) - f^* - \epsilon}{4\epsilon} \quad \text{iterations.}$$

3. *To satisfy $f(w_K) - f^* \leq \varepsilon$ for $\mu$-PL function $f$ (but not necessarily convex), `GD` with constant step-size initialized at $w_0$, needs at least*

$$K \geq \frac{H_1}{4\mu} \frac{(f(w_0) - f^*)}{\log(f(w_0) - f^*) + 1} \log\left(\frac{f(w_0) - f^*}{\epsilon}\right) \quad \text{iterations.}$$

This result covers the one in (Zhang et al., 2019) as a special case, and it also covers convex (thus also functions that satisfy the Aiming condition) and $\mu$-PL functions.

### 4.3 Convergence of GD with Adaptive Warm-up Step-size

Next, we turn to the analysis of GD under Assumption 3.1 with an adaptive step-size of the form

$$\eta_k := \frac{1}{10H_0 + 20H_1(f(w_k) - f^*)} \tag{1}$$

prescribed by $(H_0, H_1)$-smoothness. Since the function sub-optimality decreases at the beginning of training, the theoretical step-size follows a warm-up-like scheme. In the general non-convex case, the derived upper bound in Theorem G.1 provides only numerical improvement over a constant schedule.

To achieve tangible improvements, additional convexity-like assumptions are necessary. The loss landscape of neural networks exhibits additional structure. Prior studies indicate that, near a minimizer, neural network loss surfaces often display a convex-like geometry (Kleinberg et al., 2018; Guille-Escuret et al., 2023; Islamov et al., 2024; Tran et al., 2024). This observation has motivated relaxations of convexity, such as the aiming condition (Liu et al., 2023) and quasar-convexity (Hardt et al., 2018), which have been leveraged in the analysis of various gradient-based algorithms (Gower et al., 2021; Hinder et al., 2020; Fu et al., 2023). Importantly, these conditions are satisfied by certain classes of non-convex functions (Hardt et al., 2018; Liu et al., 2023).

**Theorem 4.2.** *Assume that $f$ is $(H_0, H_1)$-smooth, and it satisfies the Aiming condition with constant $\theta$ around the set of global minimizers $\mathcal{S}$. Then the iterates of GD with adaptive step-size $\theta \cdot \eta_k$ satisfy*

$$f(w_K) - f^* \le \varepsilon \quad \text{after at most} \quad \frac{40H_0 \operatorname{dist}(w_0, \mathcal{S})^2}{\theta^2 \varepsilon} + \frac{40H_1 \operatorname{dist}(w_0, \mathcal{S})^2}{\theta^2} \quad \text{iterations.}$$

To derive a tighter convergence rate, we split the iterations into two parts – small and large function values – and analyze them separately. The convergence rate in the convex setting is recovered by setting $\theta = 1$. Notably, the $1/\varepsilon$ term depends only on $H_0$, as in the standard convex GD theory, while $H_1$ influences only the constant term. Comparing the bounds in Theorem 4.2 and Theorem 4.1, we observe that GD **with a warm-up adaptive step-size outperforms the fixed step-size version when** $H_1(f(w_0) - f^*)/\varepsilon$ **is large**, i.e., when the algorithm is poorly initialized or a high precision solution is required. This factor can be significant, potentially even exponential in $H_1 \operatorname{dist}(w_0, \mathcal{S})$ (Gaash et al., 2025). These findings offer a theoretical justification for the practical need for a warm-up when network initialization is sub-optimal.

Next, we consider another widely studied class of structured non-convex functions, which encompasses the $\mu$-PL functions–known to hold for sufficiently over-parameterized networks (Liu et al., 2022). Moreover, PL is considered the weakest sufficient condition ensuring linear convergence of GD (Karimi et al., 2016).

**Theorem 4.3.** *Assume that $f$ is $(H_0, H_1)$-smooth, and it satisfies $\mu$-PL condition. Then the iterates of GD with adaptive step-size $\eta_k$ satisfy*

$$f(w_K) - f^* \le \varepsilon \quad \text{after at most} \quad \frac{40H_1}{\mu}(f(w_0) - f^*) + \frac{20H_0}{\mu} \log \frac{H_0}{2H_1 \varepsilon} \quad \text{iterations.}$$

Similar to the convex case, the $\varepsilon$-dependent term in GD with a warm-up adaptive step-size leads to faster convergence whenever $H_1(f(w_0) - f^*)$ is substantially larger than $H_0$.

In Appendix A, we demonstrate that our proof techniques in both Theorems 4.2 and 4.3 can be used for a more general class of functions, where the function sub-optimality in Definition 3.1 is raised to the power $\rho \ge 1$, extending the benefits of the theoretical warm-up to a broader class of functions.

### 4.4 Extension to the Stochastic Setting

In a standard training setup, the function $f$ has a finite sum structure, namely,

$$f(w) := \frac{1}{n} \sum_{i=1}^{n} f_i(w) \tag{*}$$

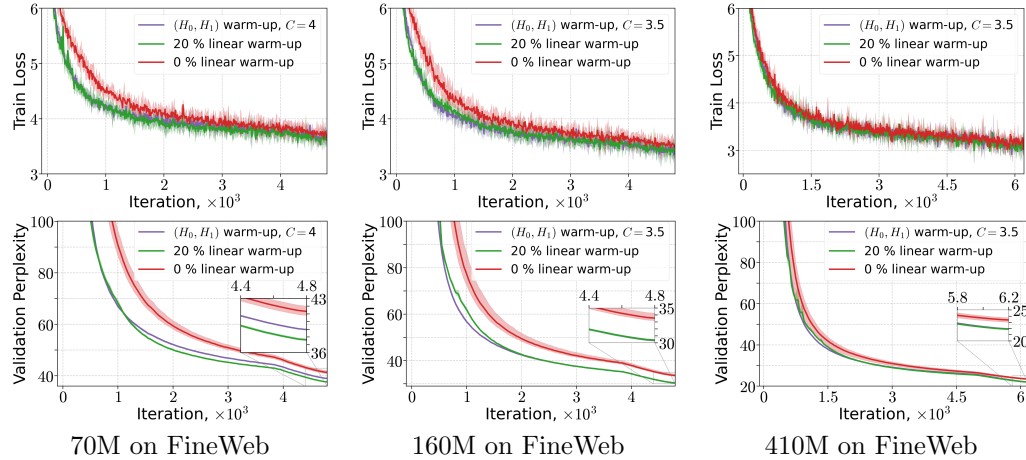

Figure 3: Performance of `Adam` (for 70M and 160M) and `AdamW` (for 410M with weight decay $\lambda = 0.1$) when training language models with three warm-up strategies: $(H_0, H_1)$ warm-up with tuned $C$, tuned linear warm-up, and no warm-up. The last 20% of iterations is a linear decay from the peak LR to $10^{-5}$ in all cases.

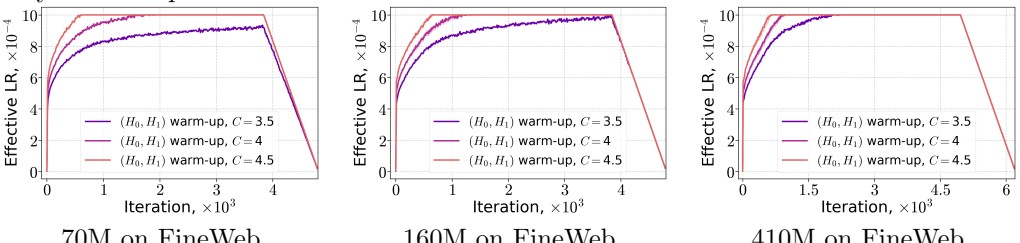

Figure 4: Effective LR with $(H_0, H_1)$ warm-up when training language models on the FineWeb dataset for the peak LR $10^{-3}$, varying parameter in $(H_0, H_1)$ warm-up.

where $n$ is the size of the training dataset, and each $f_i$ represents a loss on $i$-th sample. We define the minimum of each loss $f_i^* = \min_w f_i(w)$. To study the convergence in the stochastic setting, we need an interpolation condition, which is typically satisfied for over-parameterized networks (Ma et al., 2018). Analytically, it means that $f^* = f_i^*$ for all $i \in [n]$.

**Theorem 4.4.** *Assume that the problem* ($*$) *satisfies the interpolation condition. Assume that each $f_i$ is $(H_0, H_1)$-smooth and satisfies the Aiming condition around the set of global minimizers $\mathcal{S}$. Then the iterates of `SGD` $w_{k+1} = w_k - \eta_k \nabla f_{S_k}(w_k)$ with a step-size $\eta_k = \frac{\theta}{10H_0 + 20H_1(f_{S_k}(w_k) - f_{S_k}^*)}$ and batch $S_k \subseteq [n]$ satisfy*

$$\frac{1}{K+1} \sum_{k=0}^{K} \mathbb{E}\left[\min\left\{f(w_k) - f^*, \frac{H_0}{2nH_1}\right\}\right] \leq \frac{20H_0 \text{dist}(w_0, \mathcal{S})^2}{\theta^2(K+1)}.$$

We observe that the convergence rate depends on $H_0$, mirroring the deterministic result in Theorem 4.2. The convergence metric we use is non-standard, adopted because uniform convergence over all component functions $\{f_i\}_{i=1}^n$ cannot be ensured. With probability at most $\frac{40nH_0 \text{dist}(w_0, \mathcal{S})^2}{\theta^2(K+1)}$ the sub-optimality $f(w_k) - f^*$ can be larger than $\frac{H_0}{2nH_1}$ for any $k \in \{0, \ldots, K\}$. Nonetheless, the failure probability vanishes with $K \to \infty$, implying convergence after a sufficiently large number of iterations with high probability.

## 5 EXPERIMENTS

We next evaluate the warm-up schedule derived from $(H_0, H_1)$-smoothness on two benchmarks: transformer language modeling on FineWeb and ViT-Tiny training on ImageNet-32, both of which are known to benefit from warm-up. This section aims to highlight the merits of warm-up, particularly the gains obtained from the $(H_0, H_1)$-smoothness–driven schedule rather than to achieve state-of-the-art performance. To demonstrate the validity of the theoretical warm-up schedule, we compare linear and no warm-up with the following $(H_0, H_1)$

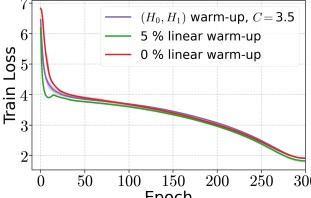 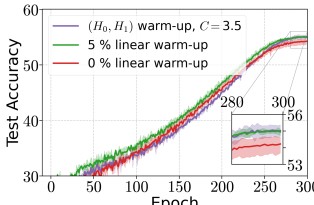 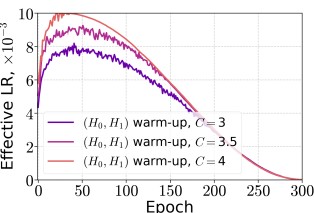

Figure 5: Performance of `AdamW` with weight decay $\lambda = 0.05$ when training ViT model on the ImageNet32 dataset with three warm-up strategies: $(H_0, H_1)$ warm-up with tuned $C$, tuned linear warm-up, and no warm-up. All LR schedules follow cosine decay after the warm-up phase.

warm-up scheduling: $\frac{\eta_k}{\max\{1, f_{S_k}(w_k)/C\}}$, where $f_{S_k}(w_k)$ is the stochastic loss at iteration $k$, $C$ is the parameter of $(H_0, H_1)$ warm-up, that controls the warm-up length. Here $\eta_k$ follows the WSD schedule (language modeling) or cosine annealing (ViT training) with no warm-up. All training details are reported in Appendix J.

**Language Modeling.** We train language models of three sizes: 70M, 160M, and 410M near Chinchilla optimum (Hoffmann et al., 2022a). When training 70M and 160M models, two baselines are `Adam` (Kingma & Ba, 2014) with WSD schedule (Hu et al., 2024) with 20 % decay stage and tuning the warm-up stage in $\{0\%, 10\%, 20\%\}$. When training 410M model, we also add weight decay $\lambda = 0.1$ (Loshchilov & Hutter, 2017). We report the mean of 3 runs, with the shaded area showing the min–max range.

In this setup, $\eta_k$ in $(H_0, H_1)$ warm-up follows the WSD schedule without warm-up (i.e., the LR starts directly at its peak) with a 20% decay phase. This can be viewed as a hard counterpart of the theoretical step-size considered in our convergence analysis. We tune the parameter $C$, which determines the length of the $(H_0, H_1)$ warm-up, over the set $\{3.5, 4, 4.5\}$ (which was found to yield good results empirically). For all warm-up schedules, we tune the peak LR over $\{3 \cdot 10^{-4}, 10^{-3}, 3 \cdot 10^{-3}, 10^{-2}\}$. Figure 3 shows that the theoretically motivated $(H_0, H_1)$ warm-up performs competitively with linear warm-up, which is the standard choice in practice, and both warm-up schedules improve over training without warm-up. We also demonstrate the evaluation of the effective LR in Figure 4. We observe that $(H_0, H_1)$ has a significantly different warm-up shape than a linear one.

**Image Classification.** Next, we repeat the study on ViT-Tiny using cosine annealing for $\eta_k$ (replacing WSD) while keeping the same warm-up mechanisms. For the $(H_0, H_1)$ warm-up, we sweep $C \in \{3, 3.5, 4\}$; for linear warm-up, we vary the warm-up length in $\{0\%, 5\%, 10\%\}$. For each schedule, we grid-search the peak LR over $\{3 \cdot 10^{-4}, 10^{-3}, 3 \cdot 10^{-3}, 10^{-2}, 3 \cdot 10^{-2}\}$. As in the previous setting, Figure 5 shows that $(H_0, H_1)$ warm-up matches linear warm-up, and both outperform training with no warm-up. The right sub-figure in Figure 5 presents the effective LR. Similar to the previous case, the warm-up substantially differs from the linear warm-up. We report the mean of three runs, with the shaded area showing the min–max range.

## 6 LIMITATIONS AND FUTURE WORK

Our experiments show that the $(H_0, H_1)$ condition provides a relatively tight curvature bound at the start of training. However, we observe that $(i)$ the bound can be improved in the initial iterations for some architectures, particularly LLMs, and $(ii)$ a phase transition occurs after warm-up, where the bound begins to deteriorate. A promising direction for future work would be to identify curvature upper bounds that remain valid across the entire training trajectory, therefore going beyond the warm-up phase. Another promising direction for tightening the smoothness bound is to extend the proposed condition to a layer-wise setting, since different network blocks may exhibit varying conditioning. This would necessitate a deeper understanding of how the final loss depends on each block. Finally, our experiments show that the theoretically motivated LR warm-up can match the performance of linear warm-up, though further investigation is needed before it could be applied as a practical replacement – an objective beyond the scope of this work.

## Reproducibility Statement

We detail model configurations and training pipelines in Appendix J. Our code base is built upon publicly available repositories, which we link to for reference. All experiments utilize publicly available datasets, cited accordingly.

## Ethics Statement

This paper presents work whose goal is to advance the field of Machine Learning. There are many potential societal consequences of our work, none of which we feel must be specifically highlighted here.

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

# Appendix

## Contents

Table A.1: Summary of the comparison between our work and (Liu et al., 2025).

| | This Work | (Liu et al., 2025) |
|---|---|---|
| **Proposed condition** | We study the condition: $\|\nabla^2 f(w)\|_2 \le H_0 + H_1(f(w) - f^*)$ | They propose the condition: $\|\nabla^2 f(w)\|_2 \le H_0 + H_1(f(w) - f^*)^\rho$ |
| **Theoretical Evidence** | 1. General deep linear network of arbitrary width with MSE loss under balancedness 2. Deep non-linear network of arbitrary width a single non-linearity under balancedness in the second last layer with MSE loss 3. 2-layer network with arbitrary activation MSE and CE losses with L2 regularization | 1. Toy MLP model with 4 parameters, no activation between middle layers, and with CE loss 2. Toy recurrent model with 2 parameters, no activation between middle layers, and CE loss |
| **Empirical Evidence** | GPT-2-like language models of size 70M, 160M, 410M on FineWeb dataset, ResNet50 and ViT models on ImageNet32, where warm-up **does bring** better performance | ResNet18 on CIFAR10 and small NanoGPT-like on Tiny Shakespeare, which can be trained **without** warm-up |
| **Theoretical Analysis** | **General non-convex setting:** with numerical improvements **Structured non-convex setting:** (under the Aiming and PL conditions) with the benefits of a warm-up schedule (Convex case is a special case of analysis under the Aiming condition) **Stochastic setting:** in-expectation analysis under Aiming and interpolation with the benefits of a warm-up schedule **Lower bounds**: substantially different from each other "hard-to-optimize" functions | **General non-convex setting:** without improvement in the worst case **Convex setting:** with the benefits of a warm-up schedule **Stochastic setting:** high-probability analysis for general non-convex functions without improvement in the worst case |
| **Experimental results** | **Tested theoretical warm-up schedule:** $\frac{\eta_k}{\max\{1, f_{S_k}(w_k)/C\}}$, where $\eta_t$ follows WSD or cosine annealing **Tested workloads:** Training of GPT-2-like language models of size 70M, 160M, 410M on the FineWeb dataset, and ViT-Tiny on ImageNet32 **with benefits** from the theoretical warm-up schedule comparable to a linear warm-up | **Tested theoretical warm-up schedule:** $\eta_k = \frac{1}{4\sqrt{2}+4} \min\left\{\frac{1}{K_0}, \frac{1}{3K_1 f(w_k)}\right\}$ with tuned $K_0, K_1$ **Tested workloads:** ResNet18 on CIFAR10 **without benefits** from the theoretical warm-up |

# A COMPARISON TO LIU ET AL. (2025)

In this section, we provide a detailed comparison against a concurrent work by Liu et al. (2025). In the following section, we discuss in more detail the differences between our work and their work. The summary of the discussion is presented in Table A.1.

## A.1 THE PROPOSED CONDITIONS

Liu et al. (2025) proposed the following condition with a general power $\rho > 0$:

$$\|\nabla^2 f(w)\| \le K_0 + K_1(f(w) - f^*)^\rho. \tag{2}$$

The condition we study in the main part of the paper is a special case of (2) with $\rho = 1$. Liu et al. (2025) proves the convergence in the convex setting under (2), demonstrating benefits of the theoretical warm-up schedule. The proposed theoretical step-size is similar to ours in (1). However, their results can be simply recovered from our analysis for the $\rho = 1$ case.

Indeed, assuming $\rho > 1$ and that the iterates $\{w_k\}_{k=0}^K$ stay in the set $\{w \mid f(w) - f^* \le f(w_0) - f^*\}$, which is the case for GD, we can simplify (2) as follows

$$\|\nabla^2 f(w)\|_2 \le K_0 + K_\rho(f(w) - f^*)^\rho \le K_0 + K_\rho(f(w_0) - f^*)^{\rho-1}(f(w) - f^*), \tag{3}$$

i.e., Definition 3.1 holds with $H_0 = K_0$ and $H_1 = K_\rho(f(w_0) - f^*)^{\rho-1}$. Therefore, the results of Theorem 4.2 apply, leading to the iteration complexity of GD with adaptive warm-up schedule of the form

$$K = \mathcal{O}\left(\frac{K_0 \text{dist}(w_0, \mathcal{S})^2}{\theta^2 \varepsilon} + \frac{K_\rho(f(w_0) - f^*)^{\rho-1}\text{dist}(w_0, \mathcal{S})^2}{\theta^2}\right).$$

This matches the bound in Liu et al. (2025) up to constants when $\theta = 1$, and shows that the adaptive schedule converges faster whenever $(f(w_0) - f^*)/\varepsilon \gg 1$. Given the simplification in (3), it remains open whether the convergence under the general condition (2) can be further tightened.

In Proposition C.1, we show that deep non-linear networks with Leaky-ReLU activations satisfy (2), albeit under stronger assumptions than Proposition 3.2. Moreover, Proposition 3.3 covers L2-regularized networks with two layers and arbitrary activations. If one considers deeper networks, $\rho$ increases with the number of layers $\ell$.

## A.2 Theoretical Evidence

We show that Definition 3.1 holds for several standard architectures: ($i$) an $\ell$-layer linear network with MSE loss under balancedness (Proposition 3.1); ($ii$) an $\ell$-layer nonlinear network with a single activation before the output and MSE loss under balancedness with leaky ReLU (Proposition 3.2); and ($iii$) two-layer networks with MSE (Proposition 3.3) and cross-entropy losses (Proposition 3.4) under L2 regularization. These results provide clear theoretical evidence that the proposed condition is a good proxy for neural-network smoothness. By contrast, concurrent work by Liu et al. (2025) validates the condition only on a toy MLP model and a recurrent model with fewer than four parameters and sigmoid activation, leaving its extension beyond such simple cases unclear.

## A.3 Experimental Evidence

Liu et al. (2025) empirically test the proposed condition on two setups: ResNet18 trained on CIFAR-10 (Krizhevsky, 2009) and a small NanoGPT-style transformer (6 blocks, 384-dimensional embeddings, 6 attention heads) trained on the Tiny Shakespeare dataset (Karpathy, 2015). They present smoothness versus training loss in a log-log scale, which leaves open how well the condition aligns with empirical behavior. Moreover, they show that these models train efficiently without learning-rate warm-up, and that warm-up offers no benefit (see Table 1 in Liu et al. (2025)). This might question whether the proposed condition still holds for models where warm-up is essential. In contrast, we validate the condition on much larger models, where warm-up is crucial for achieving improved performance (see Figures 3 and 5).

## A.4 Theoretical Analysis

**Deterministic Setting.** Liu et al. (2025) establish convergence guarantees for GD in both convex and general non-convex regimes. In the non-convex case, their worst-case analysis shows that an adaptive warm-up schedule offers no advantage over a constant step size beyond numerical factors, which is consistent with our findings. Benefits emerge in the convex setting, which is again similar to our results. In contrast, we extend the analysis beyond convexity, demonstrating the benefits of warm-up under Aiming and PL conditions, which are known to hold for sufficiently wide networks.

**Stochastic Setting.** The convergence guarantees in Liu et al. (2025) and our work are not directly comparable. We focus on finite-sum minimization in expectation under the Aiming condition and interpolation, showing that, under these assumptions, SGD with an adaptive warm-up schedule attains performance comparable to GD in the deterministic case. By contrast, Liu et al. (2025) analyze general non-convex objectives under almost surely bounded variance or the almost-sure ABD assumption (Khaled & Richtárik, 2020) with high probability, where their results do not demonstrate warm-up benefits analogous to those in the deterministic setting.

**Lower Bounds.** Both works provide lower bounds for constant step-size GD, however, the "hard-to-optimize" functions and proof techniques are different.

## A.5 Empirical Results

We test the theoretically inspired LR warm-up schedule of the form $\frac{\eta_k}{\max\{1, f_{S_k(w_k)}/C\}}$, where $\eta_k$ follows WSD or cosine annealing schedule, $f_{S_k}(w_k)$ is the current stochastic loss and $C$ is the parameter of $(H_0, H_1)$ warm-up schedule. The proposed $(H_0, H_1)$ warm-up schedule is used when training language models of size 70M, 160M, and 410M on the FineWeb dataset

and the ViT-Tiny model on the ImageNet32 dataset. Our empirical results demonstrate that the proposed theoretical warm-up with tuned $C$ matches the performance of a linear warm-up and improves over a no-warm-up baseline. In contrast, Liu et al. (2025) conducts experiments when training the ResNet18 model on the CIFAR10 dataset. Their results demonstrate that the theoretical warm-up schedule $\frac{1}{4\sqrt{2}+4} \max\{\frac{1}{K_0}, \frac{1}{3K_1 f_{S_k}(w_k)}\}$ with tuned $K_0$ and $K_1$ matches the performance of linear and no warm-up baselines.

# B ARITHMETICS OF $(H_0, H_1)$-SMOOTH FUNCTIONS

First, we provide a formal proof of the conjecture mentioned in Section 3. In other words, the following result demonstrates that the class of $(H_0, H_1)$-smooth functions contains all $(L_0, L_1)$-smooth functions.

**Proposition B.1.** *Assume that $f$ is $(L_0, L_1)$-smooth and bounded from below, i.e., $\|\nabla^2 f(w)\| \leq L_0 + L_1\|\nabla f(w)\|$ and $f^* > -\infty$. Then $f$ satisfies Definition 3.1 with*

$$H_0 = L_0 + \frac{L_0 L_1}{\nu}, \quad H_1 = \frac{4L_1^2 + \nu L_1}{2\nu},$$

*where $\nu$ satisfies the equality $\nu = e^{-\nu^2}$.*

*Proof.* We start with Lemma 2.2 in Gorbunov et al. (2024)

$$\|\nabla f(w)\|^2 \leq \frac{2}{\nu}(L_0 + L_1\|\nabla f(w)\|)(f(w) - f^*)$$

$$\Leftrightarrow \|\nabla f(w)\|^2 - \frac{2L_1}{\nu}\|\nabla f(w)\|(f(w) - f^*) - \frac{2L_0}{\nu}(f(w) - f^*) \leq 0.$$

We need to solve this quadratic inequality w.r.t. $\|\nabla f(w)\|$. The discriminant is

$$\frac{4L_1^2}{\nu^2}(f(w) - f^*)^2 + 4 \cdot 1 \cdot \frac{2L_0}{\nu}(f(w) - f^*) > 0 = \frac{4L_1^2}{\nu^2}(f(w) - f^*)^2 + \frac{8L_0}{\nu}(f(w) - f^*) > 0,$$

i.e., it is positive. Since $\|\nabla f(w)\| \geq 0$, we should also satisfy

$$\|\nabla f(w)\| \leq \frac{\frac{2L_1}{\nu}(f(w) - f^*) + \sqrt{\frac{4L_1^2}{\nu^2}(f(w) - f^*)^2 + \frac{8L_0}{\nu}(f(w) - f^*)}}{2}$$

$$\overset{(i)}{\leq} \frac{L_1}{\nu}(f(w) - f^*) + \sqrt{\frac{L_1^2}{\nu^2}(f(w) - f^*)^2} + \sqrt{\frac{2L_0}{\nu}(f(w) - f^*)}$$

$$\overset{(ii)}{\leq} \frac{2L_1}{\nu}(f(w) - f^*) + \frac{L_0}{\nu} + \frac{1}{2}(f(w) - f^*)$$

$$= \frac{L_0}{\nu} + \frac{4L_1 + \nu}{2\nu}(f(w) - f^*),$$

where $(i)$ follows from the inequality $\sqrt{a + b} \leq \sqrt{a} + \sqrt{b}$ for any $a, b \geq 0$, $(ii)$ – from the inequality $\sqrt{ab} \leq \frac{a}{2} + \frac{b}{2}$ for any $a, b \geq 0$. Therefore, we obtain

$$\|\nabla^2 f(w)\| \leq L_0 + L_1\|\nabla f(w)\|$$

$$\leq L_0 + \frac{L_0 L_1}{\nu} + \frac{4L_1^2 + \nu L_1}{2\nu}(f(w) - f^*),$$

which means that the function $f$ is $(H_0, H_1)$-smooth.

$\square$

Next, we demonstrate that operations like summation preserve $(H_0, H_1)$-smoothness. First, we show that the class of $(H_0, H_1)$-smooth functions is closed under summation.

---

[2]One can check numerically that $\nu \in (0.56, 0.57)$.

**Proposition B.2.** *Let $f$ and $g$ be $(H_0^f, H_1^f)$- and $(H_0^g, H_1^g)$-smooth respectively. Then $h := f + g$ is $(H_0, H_1)$-smooth with*

$$H_0 = (H_0^f + H_0^g + \max\{H_1^f, H_1^g\}h^* - H_1^f f^* - H_1^g g^*), \quad \text{and} \quad H_1 = \max\{H_1^f, H_1^g\}.$$

*Proof.* By Definition 3.1, we have

$$\|\nabla^2 f(w)\| \leq H_0^f + H_1^f(f(w) - f^*), \quad \|\nabla^2 g(w)\| \leq H_0^g + H_1^g(g(w) - g^*).$$

$\square$

Therefore, we have

$$
\begin{aligned}
\|\nabla^2 h(w)\| &= \|\nabla^2 f(w) + \nabla^2 g(w)\| \\
&\leq \|\nabla^2 f(w)\| + \|\nabla^2 g(w)\| \\
&\leq H_0^f + H_1^f(f(w) - f^*) + H_0^g + H_1^g(g(w) - g^*) \\
&\leq (H_0^f + H_0^g) + \max\{H_1^f, H_1^g\}(f(w) + g(w)) - H_1^f f^* - H_1^g g^* \\
&= \underbrace{(H_0^f + H_0^g + \max\{H_1^f, H_1^g\}h^* - H_1^f f^* - H_1^g g^*)}_{:=H_0} + \underbrace{\max\{H_1^f, H_1^g\}}_{:=H_1}(h(w) - h^*).
\end{aligned}
$$

Note that $h^* \geq f^* + g^*$. Therefore, we have

$$\max\{H_1^f, H_1^g\}h^* - H_1^f f^* - H_1^g g^* \geq H_1^f h^* + H_1^g h^* - H_1^f f^* - H_1^g g^* \geq 0,$$

i.e., $H_0 \geq 0$.

The next proposition shows that the class of $(H_0, H_1)$-smooth functions is closed under affine transformation.

**Proposition B.3.** *Let $g: \mathbb{R}^q \to \mathbb{R}$ be $(H_0^g, H_1^g)$-smooth, $A \in \mathbb{R}^{q \times p}$ be an arbitrary matrix, and $b \in \mathbb{R}^q$ be an arbitrary vector. We define $f: \mathbb{R}^p \to \mathbb{R}$ as $f(w) := g(Aw + b)$. Then $f$ is $(H_0^f, H_1^f)$-smooth with*

$$H_0^f = \|A\|^2(H_0^g + H_1(f^* - g^*)), \quad H_1^f = \|A\|^2 H_1^g,$$

*where $f^* = \min_{w \in \mathbb{R}^p} f(w), g^* = \min_{y \in \mathbb{R}^q} g(y)$.*

*Proof.* First, note that

$$f^* = \min_{w \in \mathbb{R}^p} g(Aw + b) \geq \min_{y \in \mathbb{R}^q} g(y) = g^*,$$

since the first minimum is taken in $\text{Im}(A)$. Second, note that $\nabla^2 f(w) = A^\top \nabla^2 g(Aw + b)A$. Therefore,

$$
\begin{aligned}
\|\nabla^2 f(w)\| &= \|A^\top \nabla^2 g(Aw + b)A\| \\
&\leq \|A^\top\| \cdot \|\nabla^2 g(Aw + b)\| \cdot \|A\| \\
&\leq \|A\|^2 \cdot (H_0^g + H_1^g(g(Aw + b) - g^*)) \\
&= \|A\|^2 H_0^g + \|A\|^2 H_1^g(f(w) - f^* + f^* - g^*) \\
&= \|A\|^2(H_0^g + H_1(f^* - g^*)) + \|A\|^2 H_1^g(f(w) - f^*).
\end{aligned}
$$

$\square$

In the next proposition, we demonstrate that the class of $(L_0, L_1)$-smooth functions is not closed under summation.

**Proposition B.4.** *There exist two $(L_0, L_1)$-smooth functions $f_1, f_2: \mathbb{R} \to \mathbb{R}$ such that their sum $f = f_1 + f_2$ does not belong to the class of $(L_0, L_1)$-smooth functions.*

*Proof.* Let us consider two functions $f_1$ and $f_2$ defined as

$$f_1(w) = \int_0^w (u + \sin(u^2)) \mathrm{d}u, \quad f_2(w) = \int_0^w (-v + \sin(v^2)) \mathrm{d}v.$$

Then we have

$$f_1'(w) = w + \sin(w^2), \quad f_1''(w) = 1 + 2w\cos(w^2), \quad f_2'(w) = -w + \sin(w^2), \quad f_2''(w) = -1 + 2w\cos(w^2).$$

Therefore, we have

$$|f_{1,2}''(w)| \le 1 + |2w\cos(w^2)| \le 1 + 2|w|,$$

and

$$|f_{1,2}'(w)| \ge |\pm w + \sin(w^2)| \ge |w| - |\sin(w^2)| \ge |w| - 1.$$

This implies that for $|w| \ge 1$

$$|f_{1,2}''(w)| \le 1 + 2|w| \le 3 + 3(|w| - 1) \le 3 + 3|f_{1,2}'(w)|.$$

For $|w| \le 1$, we have $|f_{1,2}''(w)| \le 3$. Thus, both functions are $(L_0, L_1)$-smooth with $L_0 = L_1 = 3$. They sum is $f(w) = 2\sin(w^2)$, for which we have

$$f'(w) = 2\sin(w^2), \quad f''(w) = 4w\cos(w^2).$$

Now we consider points $\{w_m\}_{m=1}^\infty$ with $w_m = \sqrt{m\pi}$. At these points, we have

$$f'(w_m) = 0, \quad f''(w_m) = 4w_m \to \infty.$$

If $f$ were $(L_0, L_1)$-smooth, then we would have

$$|f''(w_m)| \le L_0 + L_1|f'(w_m)| \le L_0.$$

This contradiction concludes the proof. $\qquad\square$

We now show that there exists an affine transformation that does not preserve $(L_0, L_1)$-smoothness.

**Proposition B.5.** *There exist a $(L_0, L_1)$-smooth function $g \colon \mathbb{R}^2 \to \mathbb{R}$ and a matrix $A \in \mathbb{R}^{2\times 1}$ such that a function $f(w) = g(Aw)$ does not belong to the class of $(L_0, L_1)$-smooth functions.*

*Proof.* Let us consider $A = \begin{pmatrix} 1 \\ 0 \end{pmatrix}$, $b = 0$, and $g(y_1, y_2) = h(y_1)e^{y_2}$ with $h(y_1) = \cos(y_1)e^{y_1}$.

We know that

$$h'(y_1) = e^{y_1}(\cos(y_1) - \sin(y_1)), \quad h''(y_1) = -2\sin(y_1)e^{y_1}.$$

Therefore,

$$\nabla g(y) = e^{y_2} \begin{pmatrix} h'(y_1) \\ h(y_1) \end{pmatrix}, \quad \nabla^2 g(y) = e^{y_2} \begin{pmatrix} h''(y_1) & h'(y_1) \\ h'(y_1) & h(y_1) \end{pmatrix}.$$

Note that

$$|h''(y_1)| = 2e^{y_1}|\sin(y_1)| \le 2e^{y_1}|\cos(y_1)| + 2e^{y_1}|\cos(y_1) - \sin(y_1)| = 2|h(y_1)| + 2|h'(y_1)|.$$

Therefore, we have

$$\begin{aligned}
\|\nabla^2 g(y)\|_2 &\le \|\nabla^2 g(y)\|_{\mathrm{F}} \\
&= e^{y_2}\sqrt{(h''(y_1))^2 + 2(h'(y_1))^2 + (h(y_1))^2} \\
&\le e^{y_2}\sqrt{4(h(y_1) + h'(y_1))^2 + 2(h'(y_1))^2 + (h(y_1))^2} \\
&\le e^{y_2}\sqrt{8(h(y_1))^2 + 8(h'(y_1))^2 + 2(h'(y_1))^2 + (h(y_1))^2} \\
&\le \sqrt{10}e^{y_2}\sqrt{(h(y_1))^2 + (h'(y_1))^2}
\end{aligned}$$

Note that $\|\nabla g(y)\| = e^{y_2}\sqrt{(h(y_1))^2 + (h'(y_1))^2}$. Therefore, we obtain the bound $\|\nabla^2 g(y)\|_2 \le \sqrt{10}\|\nabla g(y)\|$. Now we consider the function $f(w) = g(Aw) = g(w, 0) = h(w)$. For $f$, we have

$$f'(w) = e^w(\cos(w) - \sin(w)), \quad f''(w) = -2\sin(w)e^w.$$

We consider the points $\{w_m\}_{m=1}\infty$ with $w_m = \frac{\pi}{4} + 2\pi m$. Therefore, $\cos(w_m) = \sin(w_m) = \sqrt{2}/2$. This implies, that at these points $f'(w_m) = e^{w_m}(\sqrt{2}/2 - \sqrt{2}/2) = 0$ and $f''(w_m) = -\sqrt{2}e^{w_m}$. Thus, we obtain that $|f''(w_m)| \to \infty$ with $m \to \infty$, while $|f'(w_m)| = 0$. This implies that $f$ does not satisfy $(L_0, L_1)$-smoothness for any $L_0, L_1 \ge 0$.

$\qquad\square$

## C    Missing Proofs for Section 3.1.1

**Proposition 3.1.** *Consider a deep linear network with $\ell$ layers and MSE loss:*

$$f(W) \equiv f(W_1, \dots, W_\ell) = \|Y - W_1 W_2 \dots W_\ell X\|_{\mathrm{F}}^2,$$

*where $Y \in \mathbb{R}^{c \times m}$ are the labels, $X \in \mathbb{R}^{d \times m}(d \leq m)$ is the input, and $W_i \in \mathbb{R}^{n_{i-1} \times n_i}$, where $n_0 = c$ and $n_\ell = d$ are networks' weights. In the space of strongly balanced weights, i.e., when $W_i^\top W_i = W_{i+1} W_{i+1}^\top$ for all $i \in [\ell - 1]$, it holds that*

$$\|\nabla^2 f(W)\|_2 \leq H_0 + H_1(f(W) - f^*),$$

*where exact forms of $H_0$ and $H_1$ are provided in equations (5) and (6) in the Appendix.*

$$\bar{H}_0 := 4\ell^2 \left( (2d^{\frac{\ell-1}{2}})^{\frac{2\ell-2}{\ell}} \left( \frac{1}{\lambda_{\min}(XX^\top)} \right)^{\frac{2\ell-2}{2\ell}} \|Y\|_{\mathrm{F}}^{\frac{2\ell-2}{\ell}} \|X\|_2^2 + (2d^{\frac{\ell-1}{2}})^{\frac{\ell-2}{\ell}} \left( \frac{1}{\lambda_{\min}(XX^\top)} \right)^{\frac{\ell-2}{2\ell}} \|Y\|_{\mathrm{F}}^{\frac{\ell-2}{\ell}} \|X\|_2 \right),$$

$$\tag{4}$$

$$H_0 := 2\bar{H}_0 + H_1(1 + f^*) \tag{5}$$

and

$$\begin{aligned}
H_1 := 4\ell^2 \Bigg( & (2d^{\frac{\ell-1}{2}})^{\frac{2\ell-2}{\ell}} \left( \frac{1}{\lambda_{\min}(XX^\top)} \right)^{\frac{2\ell-2}{2\ell}} \|X\|_2^2 + (2d^{\frac{\ell-1}{2}})^{\frac{\ell-2}{\ell}} \left( \frac{1}{\lambda_{\min}(XX^\top)} \right)^{\frac{\ell-2}{2\ell}} \|X\|_2 \\
& + (2d^{\frac{\ell-1}{2}})^{\frac{\ell-2}{\ell}} \left( \frac{1}{\lambda_{\min}(XX^\top)} \right)^{\frac{\ell-2}{2\ell}} \|Y\|_{\mathrm{F}}^{\frac{\ell-2}{\ell}} \|X\|_2 \Bigg).
\end{aligned} \tag{6}$$

*Proof.* The proof is split in two parts: first we obtain an upper bound for the norm of the Hessian and second a lower bound for the loss value.

**Upper bound for the Hessian norm:**    One can find an explicit formula for the Hessian of such neural network in Kawaguchi (2016), Lemma 4.3.

The Hessian of $f$ in vectorized form has blocks in the $(i, j)$ position for $j < i$, that are of the form

$$\begin{aligned}
\frac{\partial^2 f}{\partial \mathrm{vec}(W_i)\mathrm{vec}(W_j)} = {} & 2((W_1 \dots W_{i-1}) \otimes (W_{i+1} \dots W_\ell X)^\top)^\top ((W_1 \dots W_{j-1}) \otimes (W_{j+1} \dots W_\ell X)^\top) \\
& + 2((W_{j+1} \dots W_{i-1})^\top \otimes (W_{i+1} \dots W_\ell X))(I_{n_j} \otimes ((W_1 \dots W_\ell X - Y)^\top W_1 \dots W_{j-1})),
\end{aligned}$$

where $W_1 W_0, W_{\ell+1} W_\ell := I$.

For $j = i$, we have

$$\frac{\partial^2 f}{\partial \mathrm{vec}(W_i)\mathrm{vec}(W_j)} = 2((W_1 \dots W_{i-1}) \otimes (W_{i+1} \dots W_\ell X)^\top)^\top ((W_1 \dots W_{j-1}) \otimes (W_{j+1} \dots W_\ell X)^\top).$$

The spectral norm of the Hessian in vectorized form is upper bounded by the sum of the spectral norms of each such block. Indeed, let $M$ be an $N \times N$ block symmetric matrix:

$$M = \begin{pmatrix} M_{11} & M_{12} & \cdots & M_{1N} \\ M_{12}^\top & M_{22} & \cdots & M_{2N} \\ \vdots & \vdots & \ddots & \vdots \\ M_{1N}^\top & M_{2N}^\top & \cdots & M_{NN} \end{pmatrix}$$

where each $M_{ij}$ is a matrix block.

A fundamental result for block matrices states that the spectral norm of a block matrix is bounded by the spectral norm of the matrix formed by the spectral norms of its blocks. Let

us define a real symmetric $N \times N$ matrix $\tilde{M}$ where each element $(\tilde{M})_{ij}$ is the spectral norm of the corresponding block $M_{ij}$:

$$\tilde{M} = \begin{pmatrix} \|M_{11}\|_2 & \|M_{12}\|_2 & \cdots & \|M_{1N}\|_2 \\ \|M_{12}\|_2 & \|M_{22}\|_2 & \cdots & \|M_{2N}\|_2 \\ \vdots & \vdots & \ddots & \vdots \\ \|M_{1N}\|_2 & \|M_{2N}\|_2 & \cdots & \|M_{NN}\|_2 \end{pmatrix}$$

The inequality is then:

$$\|M\|_2 \leq \|\tilde{M}\|_2$$

Since the spectral norm is always upper bounded by the Frobenius norm, it holds

$$\|\tilde{M}\|_2 \leq \|\tilde{M}\|_{\mathrm{F}} = \sqrt{\sum_{i=1}^{N} \sum_{j=1}^{N} \|M_{ij}\|_2^2} \leq \sum_{i=1}^{N} \sum_{j=1}^{N} \|M_{ij}\|_2.$$

Thus, indeed, it holds

$$\|M\|_2 \leq \sum_{i=1}^{N} \sum_{j=1}^{N} \|M_{ij}\|_2. \tag{7}$$

Going back to the Hessian, we can upper bound the spectral norm of the $(i,j)$ block using only the weak form of balancedness $\|W_i\|_{\mathrm{F}} = \|W_{i+1}\|_{\mathrm{F}}$ (which is implied by the strong form of balancedness).

For $1 < j < i < \ell$, we have

$$\left\| \frac{\partial^2 f}{\partial \mathrm{vec}(W_i)\mathrm{vec}(W_j)} \right\|_2 = 2\|((W_1 \ldots W_{i-1}) \otimes (W_{i+1} \ldots W_\ell X)^\top)^\top ((W_1 \ldots W_{j-1}) \otimes (W_{j+1} \ldots W_\ell X)^\top)$$

$$+ 2((W_{j+1} \ldots W_{i-1})^\top \otimes (W_{i+1} \ldots W_\ell X))(I_{n_j} \otimes ((W_1 \ldots W_\ell X - Y)^\top W_1 \ldots W_{j-1})\|_2$$

$$\leq 2\|((W_1 \ldots W_{i-1}) \otimes ((W_{i+1} \ldots W_\ell X)^\top)^\top (W_1 \ldots W_{j-1}) \otimes (W_{j+1} \ldots W_\ell X)^\top)\|_2$$

$$+ 2\|((W_{j+1} \ldots W_{i-1})^\top \otimes (W_{i+1} \ldots W_\ell X))(I_{n_j} \otimes ((W_1 \ldots W_\ell X - Y)^\top W_1 \ldots W_{j-1})\|_2$$

$$\leq 2\|W_1\|_{\mathrm{F}}^{2\ell-2}\|X\|_2^2 + 2\|W_1\|_{\mathrm{F}}^{\ell-2}\|X\|_2\sqrt{f(W)}.$$

For the last inequality, we used that for matrices $A$ and $B$

- $\|A \otimes B\|_2 = \|A\|_2\|B\|_2$.

- $\|A\|_2 = \|A^\top\|_2$

- $\|AB\|_2 \leq \|A\|_2\|B\|_2$.

- $\|A\|_2 \leq \|A\|_{\mathrm{F}}$.

For $j = 1$ and $1 < i < \ell$, we have

$$\frac{\partial^2 f}{\partial \mathrm{vec}(W_i)\mathrm{vec}(W_1)} = 2((W_1 \ldots W_{i-1}) \otimes (W_{i+1} \ldots W_\ell X)^\top)^\top (I_c \otimes (W_2 \ldots W_\ell X)^\top)$$

$$+ 2((W_2 \ldots W_{i-1})^\top \otimes (W_{i+1} \ldots W_\ell X))(I_{n_j} \otimes (W_1 \ldots W_\ell X - Y)^\top),$$

thus

$$\left\| \frac{\partial^2 f}{\partial \mathrm{vec}(W_i)\mathrm{vec}(W_1)} \right\|_2 \leq 2\|W_1\|_{\mathrm{F}}^{2\ell-2}\|X\|_2^2 + 2\|W_1\|_{\mathrm{F}}^{\ell-2}\|X\|_2\sqrt{f(W)}.$$

For $j = 1$ and $i = \ell$, it holds

$$\frac{\partial^2 f}{\partial \mathrm{vec}(W_i)\mathrm{vec}(W_1)} = 2((W_1 \ldots W_{\ell-1}) \otimes X)(I_c \otimes (W_2 \ldots W_\ell X)^\top)$$

$$+ 2((W_2 \ldots W_{\ell-1})^\top \otimes X)(I_{n_1} \otimes ((W_1 \ldots W_\ell X - Y)^\top),$$

thus again

$$\left\|\frac{\partial^2 f}{\partial \text{vec}(W_\ell)\text{vec}(W_1)}\right\|_2 \leq 2\|W_1\|_{\text{F}}^{2\ell-2}\|X\|_2^2 + 2\|W_1\|_{\text{F}}^{\ell-2}\|X\|_2\sqrt{f(W)}.$$

For the case that $1 < j < \ell$ and $i = \ell$, we have

$$\frac{\partial^2 f}{\partial \text{vec}(W_i)\text{vec}(W_j)} = 2((W_1 \ldots W_{\ell-1}) \otimes X)((W_1 \ldots W_{j-1}) \otimes (W_{j+1} \ldots W_\ell X)^\top)$$
$$+ 2((W_{j+1} \ldots W_{\ell-1})^\top \otimes X)(I_{n_j} \otimes ((W_1 \ldots W_\ell X - Y)^\top W_1 \ldots W_{j-1}).$$

Again, we have

$$\left\|\frac{\partial^2 f}{\partial \text{vec}(W_\ell)\text{vec}(W_j)}\right\|_2 \leq 2\|W_1\|_{\text{F}}^{2\ell-2}\|X\|_2^2 + 2\|W_1\|_{\text{F}}^{\ell-2}\|X\|_2\sqrt{f(W)}.$$

Similarly, we have for the diagonal blocks that

$$\left\|\frac{\partial^2 f}{\partial \text{vec}(W_i)\text{vec}(W_j)}\right\|_2 \leq 2\|W_1\|_{\text{F}}^{2\ell-2}\|X\|_2^2.$$

In summary, since we have $(\ell^2 - \ell)$-many off-diagonal blocks and $\ell$-many diagonal blocks in the Hessian, its norm is bounded as

$$\|\nabla^2 f(W)\|_2 \leq 2\ell^2\|W_1\|_{\text{F}}^{2\ell-2}\|X\|_2^2 + 2(\ell^2 - \ell)\|W_1\|_{\text{F}}^{\ell-2}\|X\|_2\sqrt{f(W)}. \tag{8}$$

**Lower bound for the loss value:** It holds

$$\|W_1 \ldots W_\ell X\|_{\text{F}}^2 = \text{Tr}(X^\top W_\ell^\top \ldots W_2^\top W_1^\top W_1 W_2 \ldots W_\ell X)$$
$$\geq \lambda_{\min}(XX^\top)\text{Tr}(W_\ell^\top \ldots W_2^\top W_1^\top W_1 W_2 \ldots W_\ell). \tag{9}$$

In order to deal with the last term, we use the strong balancedness assumption:

$W_\ell^\top \ldots W_4^\top W_3^\top W_2^\top W_1^\top W_1 W_2 W_3 W_4 \ldots W_\ell = W_\ell^\top \ldots W_4^\top W_3^\top W_2^\top W_2 W_2^\top W_2 W_3 W_4 \ldots W_\ell = W_\ell^\top \ldots W_4^\top W_3^\top W_3 W_3^\top W_3 W_3^\top W_3 W_4 \ldots W_\ell = W_\ell^\top \ldots W_4^\top W_4 W_4^\top W_3^\top W_3 W_4 W_4^\top W_4 \ldots W_\ell = W_\ell^\top \ldots W_5 W_5^\top W_4^\top W_3^\top W_3 W_4 W_5 W_5^\top \ldots W_\ell$

and the process continuous until we reach the expression

$$(W_\ell^\top W_\ell)W_\ell^\top W_{\ell-1}^\top \ldots W_6^\top W_5^\top W_4^\top W_3^\top W_3 W_4 W_5 W_6 \ldots W_{\ell-1}W_\ell(W_\ell^\top W_\ell).$$

We can now do the same process starting from $W_3$ and so on. Repeating this process $\ell/2$ times if $\ell$ is even and $(\ell-1)/2$ if $\ell$ is odd, we arrive to the expression

$$\underbrace{(W_\ell^\top W_\ell) \ldots (W_\ell^\top W_\ell)}_{\ell-\text{times}} = (W_\ell^\top W_\ell)^\ell.$$

Since the eigenvalues of $(W_\ell^\top W_\ell)^\ell$ are $\ell$ powers of the eigenvalues of $W_\ell^\top W_\ell$, we can use the generalized mean inequality and derive

$$\frac{\text{Tr}((W_\ell^\top W_\ell)^\ell)}{d} \geq \frac{\text{Tr}((W_\ell^\top W_\ell))^\ell}{d^\ell} = \frac{\|W_\ell\|_{\text{F}}^{2\ell}}{d^\ell} = \frac{\|W_1\|_{\text{F}}^{2\ell}}{d^\ell},$$

thus

$$\text{Tr}((W_\ell^\top W_\ell)^\ell) \geq \frac{\|W_1\|_{\text{F}}^{2\ell}}{d^{\ell-1}}. \tag{10}$$

Notice that we made use of the weak balancedness assumption $\|W_\ell\|_{\text{F}} = \|W_1\|_{\text{F}}$.

Combining inequalities (9) and (10), we get

$$\|W_1 \ldots W_\ell X\|_{\text{F}} \geq \sqrt{\lambda_{\min}(XX^\top)}\frac{\|W_1\|_{\text{F}}^\ell}{d^{\frac{\ell-1}{2}}}. \tag{11}$$

Now, we take the following cases:

- If $\|W_1 \ldots W_\ell X\|_{\mathrm{F}} \leq 2\|Y\|_{\mathrm{F}}$, then, by inequality (11), we have

$$\|W_1\|_{\mathrm{F}}^\ell \leq 2d^{\frac{\ell-1}{2}} \frac{1}{\sqrt{\lambda_{\min}(XX^\top)}} \|Y\|_{\mathrm{F}},$$

thus

$$\|W_1\|_{\mathrm{F}}^{2\ell-2} \leq \left(2d^{\frac{\ell-1}{2}} \frac{1}{\sqrt{\lambda_{\min}(XX^\top)}} \|Y\|_{\mathrm{F}}\right)^{\frac{2\ell-2}{\ell}}$$

and

$$\|W_1\|_{\mathrm{F}}^{\ell-2} \leq \left(2d^{\frac{\ell-1}{2}} \frac{1}{\sqrt{\lambda_{\min}(XX^\top)}} \|Y\|_{\mathrm{F}}\right)^{\frac{\ell-2}{\ell}}.$$

In this case, we have by equation (8) that

$$\|\nabla^2 f(W)\|_{\mathrm{F}} \leq 2\ell^2 \left(2d^{\frac{\ell-1}{2}} \frac{1}{\sqrt{\lambda_{\min}(XX^\top)}} \|Y\|_{\mathrm{F}}\right)^{\frac{2\ell-2}{\ell}} \|X\|_2^2$$

$$+ 2(\ell^2 - \ell) \left(2d^{\frac{\ell-1}{2}} \frac{1}{\sqrt{\lambda_{\min}(XX^\top)}} \|Y\|_{\mathrm{F}}\right)^{\frac{\ell-2}{\ell}} \|X\|_2 \sqrt{f(W)}. \quad (12)$$

- If $\|W_1 \ldots W_\ell X\|_{\mathrm{F}} > 2\|Y\|_{\mathrm{F}}$, then

$$\sqrt{f(W)} = \|W_1 \ldots W_\ell X - Y\|_{\mathrm{F}} \geq \|W_1 \ldots W_\ell X\|_{\mathrm{F}} - \|Y\|_{\mathrm{F}}$$

$$\geq \frac{\|W_1 \ldots W_\ell X\|_{\mathrm{F}}}{2} \geq \sqrt{\lambda_{\min}(XX^\top)} \frac{\|W_1\|_{\mathrm{F}}^\ell}{2d^{\frac{\ell-1}{2}}}.$$

The last inequality follows by inequality (11).

In this case, it holds

$$\|W_1\|_{\mathrm{F}}^{2\ell-2} \leq (2d^{\frac{\ell-1}{2}})^{\frac{2\ell-2}{\ell}} \left(\frac{1}{\lambda_{\min}(XX^\top))}\right)^{\frac{2\ell-2}{2\ell}} f(W)^{\frac{2\ell-2}{2\ell}}$$

and

$$\|W_1\|_{\mathrm{F}}^{\ell-2} \leq (2d^{\frac{\ell-1}{2}})^{\frac{\ell-2}{\ell}} \left(\frac{1}{\lambda_{\min}(XX^\top))}\right)^{\frac{\ell-2}{2\ell}} f(W)^{\frac{\ell-2}{2\ell}}.$$

By equation (8), we have

$$\|\nabla^2 f(W)\|_2 \leq 2\ell^2 (2d^{\frac{\ell-1}{2}})^{\frac{2\ell-2}{\ell}} \left(\frac{1}{\lambda_{\min}(XX^\top)}\right)^{\frac{2\ell-2}{2\ell}} f(W)^{\frac{2\ell-2}{2\ell}} \|X\|_2^2$$

$$+ 2(\ell^2 - \ell)(2d^{\frac{\ell-1}{2}})^{\frac{\ell-2}{\ell}} \left(\frac{1}{\lambda_{\min}(XX^\top)}\right)^{\frac{\ell-2}{2\ell}} f(W)^{\frac{2\ell-2}{2\ell}} \|X\|_2$$

$$= \left(2\ell^2 (2d^{\frac{\ell-1}{2}})^{\frac{2\ell-2}{\ell}} \left(\frac{1}{\lambda_{\min}(XX^\top)}\right)^{\frac{2\ell-2}{2\ell}} \|X\|_2^2\right.$$

$$\left. + 2(\ell^2 - \ell)(2d^{\frac{\ell-1}{2}})^{\frac{\ell-2}{\ell}} \left(\frac{1}{\lambda_{\min}(XX^\top)}\right)^{\frac{\ell-2}{2\ell}} \|X\|_2\right) f(W)^{\frac{2\ell-2}{2\ell}}. \quad (13)$$

In general, we can sum the left hand sides of equations (12) and (13) and obtain

$$
\begin{aligned}
\|\nabla^2 f(W)\|_2 \le{} & 2\ell^2 (2d^{\frac{\ell-1}{2}})^{\frac{2\ell-2}{\ell}} \left( \frac{1}{\lambda_{\min}(XX^\top)} \right)^{\frac{2\ell-2}{2\ell}} \|Y\|_{\mathrm{F}}^{\frac{2\ell-2}{\ell}} \|X\|_2^2 \\
& + 2(\ell^2 - \ell)(2d^{\frac{\ell-1}{2}})^{\frac{\ell-2}{\ell}} \left( \frac{1}{\lambda_{\min}(XX^\top)} \right)^{\frac{\ell-2}{2\ell}} \|Y\|_{\mathrm{F}}^{\frac{\ell-2}{\ell}} \|X\|_2 \sqrt{f(W)} \\
& + \left( 2\ell^2 (2d^{\frac{\ell-1}{2}})^{\frac{2\ell-2}{\ell}} \left( \frac{1}{\lambda_{\min}(XX^\top)} \right)^{\frac{2\ell-2}{2\ell}} \|X\|_2^2 \right. \\
& \left. + 2(\ell^2 - \ell)(2d^{\frac{\ell-1}{2}})^{\frac{\ell-2}{\ell}} \left( \frac{1}{\lambda_{\min}(XX^\top)} \right)^{\frac{\ell-2}{2\ell}} \|X\|_2 \right) f(W)^{\frac{2\ell-2}{2\ell}}. \qquad (14)
\end{aligned}
$$

If $f(W) < 1$, then $\sqrt{f(W)} < 1$ and inequality (14) becomes

$$
\begin{aligned}
\|\nabla^2 f(W)\|_2 \le{} & \left( 2\ell^2 (2d^{\frac{\ell-1}{2}})^{\frac{2\ell-2}{\ell}} \left( \frac{1}{\lambda_{\min}(XX^\top)} \right)^{\frac{2\ell-2}{2\ell}} \|Y\|_{\mathrm{F}}^{\frac{2\ell-2}{\ell}} \|X\|_2^2 \right. \\
& \left. + 2(\ell^2 - \ell)(2d^{\frac{\ell-1}{2}})^{\frac{\ell-2}{\ell}} \left( \frac{1}{\lambda_{\min}(XX^\top)} \right)^{\frac{\ell-2}{2\ell}} \|Y\|_{\mathrm{F}}^{\frac{\ell-2}{\ell}} \|X\|_2 \right) \\
& + \left( 2\ell^2 (2d^{\frac{\ell-1}{2}})^{\frac{2\ell-2}{\ell}} \left( \frac{1}{\lambda_{\min}(XX^\top)} \right)^{\frac{2\ell-2}{2\ell}} \|X\|_2^2 \right. \\
& \left. + 2(\ell^2 - \ell)(2d^{\frac{\ell-1}{2}})^{\frac{\ell-2}{\ell}} \left( \frac{1}{\lambda_{\min}(XX^\top)} \right)^{\frac{\ell-2}{2\ell}} \|X\|_2 \right) f(W)^{\frac{2\ell-2}{2\ell}}. \qquad (15)
\end{aligned}
$$

It holds that $\frac{2\ell-2}{2\ell} \ge \frac{1}{2}$, thus, if $f(W) \ge 1$, we have $\sqrt{f(W)} \le f(W)^{\frac{2\ell-2}{2\ell}}$ and inequality (14) becomes

$$
\begin{aligned}
\|\nabla^2 f(W)\|_2 \le{} & 2\ell^2 (2d^{\frac{\ell-1}{2}})^{\frac{2\ell-2}{\ell}} \left( \frac{1}{\lambda_{\min}(XX^\top)} \right)^{\frac{2\ell-2}{2\ell}} \|Y\|_{\mathrm{F}}^{\frac{2\ell-2}{\ell}} \|X\|_2^2 \\
& + \left( 2\ell^2 (2d^{\frac{\ell-1}{2}})^{\frac{2\ell-2}{\ell}} \left( \frac{1}{\lambda_{\min}(XX^\top)} \right)^{\frac{2\ell-2}{2\ell}} \|X\|_2^2 \right. \\
& + 2(\ell^2 - \ell)(2d^{\frac{\ell-1}{2}})^{\frac{\ell-2}{\ell}} \left( \frac{1}{\lambda_{\min}(XX^\top)} \right)^{\frac{\ell-2}{2\ell}} \|X\|_2 \\
& \left. + 2(\ell^2 - \ell)(2d^{\frac{\ell-1}{2}})^{\frac{\ell-2}{\ell}} \left( \frac{1}{\lambda_{\min}(XX^\top)} \right)^{\frac{\ell-2}{2\ell}} \|Y\|_{\mathrm{F}}^{\frac{\ell-2}{\ell}} \|X\|_2 \right) f(W)^{\frac{2\ell-2}{2\ell}}. \qquad (16)
\end{aligned}
$$

Summing the right hand sides of (15) and (16) and using $\ell^2 - \ell \le \ell^2$, we obtain

$$\|\nabla^2 f(W)\|_2 \le 4\ell^2 \left( (2d^{\frac{\ell-1}{2}})^{\frac{2\ell-2}{\ell}} \left( \frac{1}{\lambda_{\min}(XX^\top)} \right)^{\frac{2\ell-2}{2\ell}} \|Y\|_{\mathrm{F}}^{\frac{2\ell-2}{\ell}} \|X\|_2^2 \right.$$

$$\left. + (2d^{\frac{\ell-1}{2}})^{\frac{\ell-2}{\ell}} \left( \frac{1}{\lambda_{\min}(XX^\top)} \right)^{\frac{\ell-2}{2\ell}} \|Y\|_{\mathrm{F}}^{\frac{\ell-2}{\ell}} \|X\|_2 \right)$$

$$+ 4\ell^2 \left( (2d^{\frac{\ell-1}{2}})^{\frac{2\ell-2}{\ell}} \left( \frac{1}{\lambda_{\min}(XX^\top)} \right)^{\frac{2\ell-2}{2\ell}} \|X\|_2^2 \right.$$

$$+ (2d^{\frac{\ell-1}{2}})^{\frac{\ell-2}{\ell}} \left( \frac{1}{\lambda_{\min}(XX^\top)} \right)^{\frac{\ell-2}{2\ell}} \|X\|_2$$

$$\left. + (2d^{\frac{\ell-1}{2}})^{\frac{\ell-2}{\ell}} \left( \frac{1}{\lambda_{\min}(XX^\top)} \right)^{\frac{\ell-2}{2\ell}} \|Y\|_{\mathrm{F}}^{\frac{\ell-2}{\ell}} \|X\|_2 \right) f(W)^{\frac{2\ell-2}{2\ell}}.$$

The above imply that $f$ satisfies

$$\|\nabla^2 f(W)\|_2 \le \bar{H}_0 + H_1 \nabla f(W)^{\frac{\ell-1}{\ell}}$$

for $\bar{H}_0$ and $H_1$ defined as in equations (4) and (6).

It is easy to see that if $\|\nabla^2 f(W)\|_2 \le \bar{H}_0 + H_1 f(W)^c$ for some $c < 1$, it holds $\|\nabla^2 f(W)\|_2 \le \bar{H}_0 + H_1$, if $f(W) < 1$, and $\|\nabla^2 f(W)\|_2 \le \bar{H}_0 + H_1 f(W)$, if $f(W) \ge 1$. In both cases, it folds $\|\nabla^2 f(W)\|_2 \le (2\bar{H}_0 + H_1) + H_1 f(W)$. We can also add and subtract $H_1 f^*$ in the right-hand side and get the desired result.

$\square$

**Proposition 3.2.** *Let $f$ be defined as*

$$f(W) \equiv f(W_1, \ldots, W_\ell) = \|Y - W_1 \phi(W_2 X_3 \ldots W_\ell X)\|_{\mathrm{F}}^2$$

*where $\phi$ is leaky-ReLU activation function with slopes $1$ and $b$, i.e., $\phi(x) = \max\{bx, x\}$, $0 < b \le 1$, and matrices $Y, X, \{W_i\}_{i=1}^\ell$ defined as before. Assume that over the course of GD:*
- *$\lambda_{\min}(W_1^\top W_1) \ge h > 0$.*
- *The layers $\{W_i\}_{i=1}^\ell$ are weakly balanced, i.e., $\|W_1\|_{\mathrm{F}} = \ldots = \|W_\ell\|_{\mathrm{F}}$.*
- *The layers $\{W_i\}_{i=2}^\ell$ are strongly balanced, i.e., $W_i^\top W_i = W_{i+1} W_{i+1}^\top$, for $i \in \{2, \ldots, \ell\}$.*

*Then it holds that*

$$\|\nabla^2 f(W)\|_2 \le H_0 + H_1 f(W) \quad (= (H_0 + H_1 f^*) + H_1(f(W) - f^*)),$$

*where the exact forms of $H_0$ and $H_1$ are provided in equations (17) and (18) in the Appendix.*

$$H_0 := \ell^2 \left( \frac{16d^{\ell-2}\|Y\|_{\mathrm{F}}^2}{hb^2\lambda_{\min}(XX^\top)} \|X\|_2^2 + 2 \left( \frac{4d^{\ell-2}\|Y\|_{\mathrm{F}}^2}{hb^2\lambda_{\min}(XX^\top)} \right)^{\frac{\ell-2}{2\ell-2}} \|X\|_2 + 2 \left( \frac{4}{hb^2\lambda_{\min}(XX^\top)} d^{\ell-2} \right)^{\frac{\ell-2}{2\ell-2}} \|X\|_2 \right)$$

(17)

and

$$H_1 := \ell^2 \left( \frac{16d^{\ell-2}}{hb^2\lambda_{\min}(XX^\top)} \|X\|_2^2 + 2 \left( \frac{4d^{\ell-2}\|Y\|_{\mathrm{F}}^2}{hb^2\lambda_{\min}(XX^\top)} \right)^{\frac{\ell-2}{2\ell-2}} \|X\|_2 + 2 \left( \frac{4}{hb^2\lambda_{\min}(XX^\top)} d^{\ell-2} \right)^{\frac{\ell-2}{2\ell-2}} \|X\|_2 \right).$$

(18)

*Proof.* The proof is divided into two parts, similarly to the proof of Proposition 3.1: the first obtains an upper bound for the norm of the Hessian, while the second obtains a lower bound on the loss value.

The first part in the proof of Proposition 3.1 was easy, as one has ready formulas for the Hessian. In this case, the situation is more involved and we come up with a more general process to estimate the spectral norm of the Hessian based on the gradient finite differences. This process works for any non-linear network with activations $\phi_i$ that are either ReLU or leaky-ReLU (we re-use this calculation in Proposition C.1).

**Upper bound for the Hessian norm:** To simplify the notation, we set

$$Z_\ell = W_\ell X$$
$$A_{\ell-1} = \phi_{\ell-1}(Z_\ell)$$
$$Z_{\ell-1} = W_{\ell-1}A_{\ell-1}$$
$$\vdots$$
$$Z_2 = W_2 A_2$$
$$A_1 = \phi_1(Z_2)$$
$$Z_1 = W_1 A_1 = F.$$

By the backpropagation algorithm for the gradient, we have that the gradient of $f$ can be computed as

$$\frac{\partial f}{\partial W_i} = \delta_i A_i^\top$$

where $\delta_i$ is defined recursively as

$$\delta_1 = -2(Y - F)$$
$$\delta_2 = W_1^\top \delta_1 \odot \phi_1'(Z_2)$$
$$\vdots$$
$$\delta_i = W_{i-1}^\top \delta_{i-1} \odot \phi_{i-1}'(Z_i).$$

We need to upper bound the difference of the gradient defined in two distinct, sufficiently close points $W = (W_1, \ldots, W_\ell)$ and $\bar{W} = (\bar{W}_1, \ldots, \bar{W}_\ell)$. We also define

$$\text{dist}(W, \bar{W}) := \sqrt{\sum_{i=1}^{\ell} \|W_i - \bar{W}_i\|_{\text{F}}^2}.$$

It holds that

$$\|\nabla f(W) - \nabla f(\bar{W})\|_{\text{F}} \le \sum_{i=1}^{\ell} \left\| \frac{\partial f}{\partial W_i}(W) - \frac{\partial f}{\partial W_i}(\bar{W}) \right\|_{\text{F}}.$$

We have

$$\left\| \frac{\partial f}{\partial W_i}(W) - \frac{\partial f}{\partial W_i}(\bar{W}) \right\|_{\text{F}} = \|\delta_i A_i^\top - \bar{\delta}_i \bar{A}_i^\top\|_{\text{F}} \le \|\delta_i\|_{\text{F}} \|A_i - \bar{A}_i\|_{\text{F}} + \|\bar{A}_i\|_{\text{F}} \|\delta_i - \bar{\delta}_i\|_{\text{F}}. \quad (19)$$

Here we use a bar to denote the sequences of matrices related to the point $\bar{W}$. We deal with the four sequences appearing in this upper bound one by one, starting from $\bar{A}_i$. We can equivalently deal with $\bar{A}_i$ as the only difference will be to substitute $\bar{W}$ in place of $W$.

We have

$$A_i = \phi_i(W_{i+1}A_{i+1}), \text{ for } i = 1, \ldots, \ell - 2,$$

thus

$$\|A_i\|_{\text{F}} = \|\phi_i(W_{i+1}A_{i+1})\|_{\text{F}} \le \|W_{i+1}A_{i+1}\|_{\text{F}} = \|W_1\|_{\text{F}} \|A_{i+1}\|_{\text{F}}.$$

The inequality follows from the fact that $\phi_i$ is leaky-ReLU, thus $|\phi_i(x)| \le |x|$ and the last equality by the weakly balanced assumption, i.e. that $\|W_i\|_{\text{F}} = \|W_1\|_{\text{F}}$.

This implies that

$$\|A_i\|_{\mathrm{F}} \le \|W_1\|^{\ell-1-i}\|A_{\ell-1}\| = \|W_1\|^{\ell-1-i}\|\phi_{\ell-1}(W_\ell X)\|_{\mathrm{F}} \le \|W_1\|_{\mathrm{F}}^{\ell-i}\|X\|_2. \qquad (20)$$

Similarly, it holds

$$\|\bar{A}_i\|_{\mathrm{F}} \le \|\bar{W}_1\|_{\mathrm{F}}^{\ell-i}\|X\|_2. \qquad (21)$$

Now, we deal with $A_i - \bar{A}_i$:

$$\|A_i - \bar{A}_i\|_{\mathrm{F}} = \|\phi_i(W_{i+1}A_{i+1}) - \phi_i(\bar{W}_{i+1}\bar{A}_{i+1})\|_{\mathrm{F}} \le \|W_{i+1}A_{i+1} - \bar{W}_{i+1}\bar{A}_{i+1}\|_{\mathrm{F}} \le$$

$$\|A_{i+1}\|_{\mathrm{F}}\|W_{i+1} - \bar{W}_{i+1}\|_{\mathrm{F}} + \|\bar{W}_{i+1}\|_{\mathrm{F}}\|A_{i+1} - \bar{A}_{i+1}\|_{\mathrm{F}} \le$$

$$\|A_{i+1}\|_{\mathrm{F}}\mathrm{dist}(W, \bar{W}) + \|\bar{W}_1\|_{\mathrm{F}}\|A_{i+1} - \bar{A}_{i+1}\|_{\mathrm{F}}.$$

By an induction argument, we can get the bound

$$\|A_i - \bar{A}_i\|_{\mathrm{F}} \le \left( \sum_{k=i+1}^{\ell-1} \|A_k\|\|\bar{W}_1\|^{k-i-1} \right) \mathrm{dist}(W, \bar{W}) + \|\bar{W}_1\|_{\mathrm{F}}^{\ell-i-1}\|A_{\ell-1} - \bar{A}_{\ell-1}\|_{\mathrm{F}}$$

and by inequality (20), we have

$$\|A_i - \bar{A}_i\|_{\mathrm{F}} \le \left( \sum_{k=i+1}^{\ell-1} \|W_1\|_{\mathrm{F}}^{\ell-k}\|\bar{W}_1\|^{k-i-1} \right) \mathrm{dist}(W, \bar{W})\|X\|_2 + \|\bar{W}_1\|_{\mathrm{F}}^{\ell-i-1}\|W_\ell - \bar{W}_\ell\|_{\mathrm{F}}\|X\|_2$$

$$\le \left( \sum_{k=i+1}^{\ell-1} \|W_1\|_{\mathrm{F}}^{\ell-k}\|\bar{W}_1\|^{k-i-1} \right) \mathrm{dist}(W, \bar{W})\|X\|_2 + \|\bar{W}_1\|_{\mathrm{F}}^{\ell-i-1}\mathrm{dist}(W, \bar{W})\|X\|_2.$$

$$(22)$$

Now we move to $\delta_i$. It holds

$$\|\delta_i\|_{\mathrm{F}} = \|W_{i-1}^\top \delta_{i-1} \odot \phi'_{i-1}(Z_i)\|_{\mathrm{F}} \le \|W_{i-1}\|_{\mathrm{F}}\|\delta_{i-1}\|_{\mathrm{F}} = \|W_1\|_{\mathrm{F}}\|\delta_{i-1}\|_{\mathrm{F}}.$$

This implies that

$$\|\delta_i\|_{\mathrm{F}} \le \|W_1\|_{\mathrm{F}}^{i-1}\|\delta_1\|_{\mathrm{F}} = 2\|W_1\|_{\mathrm{F}}^{i-1}\sqrt{f(W)}$$

and similarly

$$\|\bar{\delta}_i\|_{\mathrm{F}} \le 2\|\bar{W}_1\|_{\mathrm{F}}^{i-1}\sqrt{f(\bar{W})}. \qquad (23)$$

For the sequence $\delta_i - \bar{\delta}_i$, we have

$$\|\delta_i - \bar{\delta}_i\|_{\mathrm{F}} = \|W_{i-1}^\top \delta_{i-1} \odot \phi'_{i-1}(Z_i) - \bar{W}_{i-1}^\top \bar{\delta}_{i-1} \odot \phi'_{i-1}(\bar{Z}_i)\|_{\mathrm{F}}$$

and since all entries of $Z_i$ are non-zero and $\bar{Z}_i$ is taken sufficiently close to $Z_i$, these two points feature the same activation pattern, thus $\phi'_{i-1}(Z_i) = \phi'_{i-1}(\bar{Z}_i)$. This gives

$$\|\delta_i - \bar{\delta}_i\|_{\mathrm{F}} \le \|W_{i-1}^\top \delta_{i-1} - \bar{W}_{i-1}^\top \bar{\delta}_{i-1}\|_{\mathrm{F}} \le \|W_{i-1}\|_{\mathrm{F}}\|\delta_{i-1} - \bar{\delta}_{i-1}\|_{\mathrm{F}} + \|\bar{\delta}_{i-1}\|_{\mathrm{F}}\|W_{i+1} - \bar{W}_{i+1}\|_{\mathrm{F}}$$

$$\le \|W_1\|_{\mathrm{F}}\|\delta_{i-1} - \bar{\delta}_{i-1}\|_{\mathrm{F}} + \|\bar{\delta}_{i-1}\|_{\mathrm{F}}\mathrm{dist}(W, \bar{W}).$$

By induction, we have

$$\|\delta_i - \bar{\delta}_i\|_{\mathrm{F}} \le \sum_{k=i-1}^{1} \|\bar{\delta}_k\|_{\mathrm{F}}\|W_1\|_{\mathrm{F}}^{i-1-k}\mathrm{dist}(W, \bar{W}) + \|W_1\|_{\mathrm{F}}^{i-1}\|\delta_1 - \bar{\delta}_1\|_{\mathrm{F}}$$

$$\le 2\sqrt{f(\bar{W})} \sum_{k=i-1}^{1} \|\bar{W}_1\|_{\mathrm{F}}^{k-1}\|W_1\|_{\mathrm{F}}^{i-1-k}\mathrm{dist}(W, \bar{W}) + \|W_1\|_{\mathrm{F}}^{i-1}\|\delta_1 - \bar{\delta}_1\|_{\mathrm{F}}.$$

The second inequality in the previous derivation follows by inequality (23).

For $\|\delta_1 - \bar{\delta}_1\|_{\mathrm{F}}$, we have

$$\|\delta_1 - \bar{\delta}_1\|_{\mathrm{F}} = 2\|W_1 A_1 - \bar{W}_1 \bar{A}_1\|_{\mathrm{F}} \leq 2\|W_1\|_{\mathrm{F}}\|A_1 - \bar{A}_1\|_{\mathrm{F}} + 2\|\bar{A}_1\|_{\mathrm{F}}\|W_1 - \bar{W}_1\|_{\mathrm{F}} \leq$$

$$2\|W_1\|_{\mathrm{F}}\left(\left(\sum_{k=2}^{\ell-1}\|W_1\|_{\mathrm{F}}^{\ell-k}\|\bar{W}_1\|^{k-2}\right) + \|\bar{W}_1\|_{\mathrm{F}}^{\ell-2}\right)\mathrm{dist}(W,\bar{W})\|X\|_2 + 2\|\bar{W}_1\|_{\mathrm{F}}^{\ell-1}\mathrm{dist}(W,\bar{W})\|X\|_2 =$$

$$2\left(\|W_1\|_{\mathrm{F}}\left(\left(\sum_{k=2}^{\ell-1}\|W_1\|_{\mathrm{F}}^{\ell-k}\|\bar{W}_1\|^{k-2}\right) + \|\bar{W}_1\|_{\mathrm{F}}^{\ell-2}\right) + \|\bar{W}_1\|_{\mathrm{F}}^{\ell-1}\right)\mathrm{dist}(W,\bar{W})\|X\|_2.$$

Thus,

$$\|\delta_i - \bar{\delta}_i\|_{\mathrm{F}} \leq 2\sqrt{f(\bar{W})}\sum_{k=i-1}^{1}\|\bar{W}_1\|_{\mathrm{F}}^{k-1}\|W_1\|_{\mathrm{F}}^{i-1-k}\mathrm{dist}(W,\bar{W}) +$$

$$2\|W_1\|_{\mathrm{F}}^{i-1}\left(\|W_1\|_{\mathrm{F}}\left(\left(\sum_{k=2}^{\ell-1}\|W_1\|_{\mathrm{F}}^{\ell-k}\|\bar{W}_1\|^{k-2}\right) + \|\bar{W}_1\|_{\mathrm{F}}^{\ell-2}\right) + \|\bar{W}_1\|_{\mathrm{F}}^{\ell-1}\right)\mathrm{dist}(W,\bar{W})\|X\|_2. \tag{24}$$

Combining inequalities (19),(21),(22),(23) and (24), we get

$$\left\|\frac{\partial f}{\partial W_i}(W) - \frac{\partial f}{\partial W_i}(\bar{W})\right\|_{\mathrm{F}} \leq$$

$$2\|\bar{W}_1\|_{\mathrm{F}}^{i-1}\sqrt{f(\bar{W})}\left(\left(\sum_{k=i+1}^{\ell-1}\|W_1\|_{\mathrm{F}}^{\ell-k}\|\bar{W}_1\|^{k-i-1}\right) + \|\bar{W}_1\|_{\mathrm{F}}^{\ell-i-1}\right)\mathrm{dist}(W,\bar{W})\|X\|_2 +$$

$$2\|\bar{W}_1\|_{\mathrm{F}}^{\ell-i}\|X\|_2\sqrt{f(\bar{W})}\sum_{k=i-1}^{1}\|\bar{W}_1\|_{\mathrm{F}}^{k-1}\|W_1\|_{\mathrm{F}}^{i-1-k}\mathrm{dist}(W,\bar{W}) +$$

$$2\|\bar{W}_1\|_{\mathrm{F}}^{\ell-i}\|W_1\|_{\mathrm{F}}^{i-1}\left(\|W_1\|_{\mathrm{F}}\left(\left(\sum_{k=2}^{\ell-1}\|W_1\|_{\mathrm{F}}^{\ell-k}\|\bar{W}_1\|^{k-2}\right) + \|\bar{W}_1\|_{\mathrm{F}}^{\ell-2}\right) + \|\bar{W}_1\|_{\mathrm{F}}^{\ell-1}\right)\mathrm{dist}(W,\bar{W})\|X\|_2^2,$$

thus

$$\frac{\left\|\frac{\partial f}{\partial W_i}(W) - \frac{\partial f}{\partial W_i}(\bar{W})\right\|_{\mathrm{F}}}{\mathrm{dist}(W,\bar{W})} \leq$$

$$2\|\bar{W}_1\|_{\mathrm{F}}^{i-1}\sqrt{f(\bar{W})}\left(\left(\sum_{k=i+1}^{\ell-1}\|W_1\|_{\mathrm{F}}^{\ell-k}\|\bar{W}_1\|^{k-i-1}\right) + \|\bar{W}_1\|_{\mathrm{F}}^{\ell-i-1}\right)\|X\|_2 +$$

$$2\|\bar{W}_1\|_{\mathrm{F}}^{\ell-i}\|X\|_2\sqrt{f(\bar{W})}\sum_{k=i-1}^{1}\|\bar{W}_1\|_{\mathrm{F}}^{k-1}\|W_1\|_{\mathrm{F}}^{i-1-k} +$$

$$2\|\bar{W}_1\|_{\mathrm{F}}^{\ell-i}\|W_1\|_{\mathrm{F}}^{i-1}\left(\|W_1\|_{\mathrm{F}}\left(\left(\sum_{k=2}^{\ell-1}\|W_1\|_{\mathrm{F}}^{\ell-k}\|\bar{W}_1\|^{k-2}\right) + \|\bar{W}_1\|_{\mathrm{F}}^{\ell-2}\right) + \|\bar{W}_1\|_{\mathrm{F}}^{\ell-1}\right)\|X\|_2^2$$

and taking the limit as $\bar{W} \longrightarrow W$, we get

$$\lim_{\bar{W}\to W}\frac{\left\|\frac{\partial f}{\partial W_i}(W) - \frac{\partial f}{\partial W_i}(\bar{W})\right\|_{\mathrm{F}}}{\mathrm{dist}(W,\bar{W})} \leq$$

$$2(\ell-i)\|X\|_2\|W_1\|_{\mathrm{F}}^{\ell-2}\sqrt{f(W)} + 2(i-1)\|X\|_2\|W_1\|_{\mathrm{F}}^{\ell-2}\sqrt{f(W)} + 2(\ell-1)\|W_1\|_{\mathrm{F}}^{2\ell-2}\|X\|_2^2 =$$

$$2(\ell-1)\|X\|_2\sqrt{f(W)}\|W_1\|_{\mathrm{F}}^{\ell-2} + 2\ell\|W_1\|_{\mathrm{F}}^{2\ell-2}\|X\|_2^2.$$

This is because, when $\bar{W} \longrightarrow W$, it holds $\bar{W}_1 \longrightarrow W_1$.

For the total gradient difference, we have

$$\lim_{\bar{W} \to W} \frac{\left\| \nabla f(W) - \nabla f(\bar{W}) \right\|_{\mathrm{F}}}{\mathrm{dist}(W, \bar{W})} \leq \sum_{i=1}^{\ell} \lim_{\bar{W} \to W} \frac{\left\| \frac{\partial f}{\partial W_i}(W) - \frac{\partial f}{\partial W_i}(\bar{W}) \right\|_{\mathrm{F}}}{\mathrm{dist}(W, \bar{W})}$$

$$\leq 2\ell(\ell - 1)\|W_1\|_{\mathrm{F}}^{\ell-2}\|X\|_2\sqrt{f(W)} + 2\ell^2\|W_1\|_{\mathrm{F}}^{2\ell-2}\|X\|_2^2.$$

It holds

$$\|\nabla^2 f(W)\|_2 = \lim_{\bar{W} \to W} \frac{\left\| \nabla f(W) - \nabla f(\bar{W}) \right\|_{\mathrm{F}}}{\mathrm{dist}(W, \bar{W})},$$

thus

$$\|\nabla^2 f(W)\|_2 \leq 2\ell(\ell - 1)\|W_1\|_{\mathrm{F}}^{\ell-2}\|X\|_2\sqrt{f(W)} + 2\ell^2\|W_1\|_{\mathrm{F}}^{2\ell-2}\|X\|_2^2. \tag{25}$$

Notice that this is the same upper bound as the one provided in (8).

**Lower bound for the loss value:** For lower bounding $f(W)$, we have

$$\|W_1\phi(W_2 \ldots W_\ell X)\|_{\mathrm{F}}^2 \geq \lambda_{\min}(W_1^\top W_1)\|\phi(W_2 \ldots W_\ell X)\|_{\mathrm{F}}^2$$

$$\geq hb^2\|W_2 \ldots W_\ell X\|_{\mathrm{F}}^2$$

$$\geq hb^2\lambda_{\min}(XX^\top)\frac{\|W_1\|_{\mathrm{F}}^{2\ell-2}}{d^{\ell-2}}. \tag{26}$$

The first inequality is obtained by standard inequalities in linear algebra, while the third is obtained by the same reasoning we used to obtain (11), together with the assumption that $\|W_1\|_{\mathrm{F}} = \|W_2\|_{\mathrm{F}} = \cdots = \|W_\ell\|_{\mathrm{F}}$. The second inequality comes from the fact that

$$\|\phi(S)\|_{\mathrm{F}}^2 \geq b^2\|S\|_{\mathrm{F}}^2,$$

for any matrix $S$. Indeed,

$$\|\phi(S)\|_{\mathrm{F}}^2 = \mathrm{Tr}(\phi(S)^\top \phi(S))$$

and, after denoting the entries of $S$ by $s_{ij}$, we have that the diagonal entries of $\phi(S)^\top \phi(S)$ are of the form

$$\sum_k b_{ik}^2 s_{ik}^2,$$

where

$$b_{ik} = \begin{cases} b, & \text{if } s_{ik} < 0, \\ 1, & \text{if } s_{ik} \geq 0. \end{cases}$$

In any case, we have

$$\sum_k b_{ik}^2 s_{ik}^2 \geq b^2 \sum_k s_{ik}^2 = b^2\,\mathrm{Tr}(S).$$

Now, we take the following cases:

- If $\|W_1\phi(W_2 \ldots W_\ell X)\|_{\mathrm{F}} \leq 2\|Y\|_{\mathrm{F}}$, then, by equation (26), we have

$$\|W_1\|_{\mathrm{F}}^{2\ell-2} \leq \frac{4d^{\ell-2}\|Y\|_{\mathrm{F}}^2}{hb^2\lambda_{\min}(XX^\top)}.$$

and

$$\|W_1\|_{\mathrm{F}}^{\ell-2} \leq \left( \frac{4d^{\ell-2}\|Y\|_{\mathrm{F}}^2}{hb^2\lambda_{\min}(XX^\top)} \right)^{\frac{\ell-2}{2\ell-2}}.$$

In this case, we have by equation (25) that

$$\|\nabla^2 f(W)\|_{\mathrm{F}} \leq \ell^2 \frac{8d^{\ell-2}\|Y\|_{\mathrm{F}}^2}{hb^2\lambda_{\min}(XX^\top)}\|X\|_2^2$$

$$+ 2(\ell^2 - \ell)\left( \frac{4d^{\ell-2}\|Y\|_{\mathrm{F}}^2}{hb^2\lambda_{\min}(XX^\top)} \right)^{\frac{\ell-2}{2\ell-2}}\|X\|_2\sqrt{f(W)}. \tag{27}$$

- If $\|W_1\phi(W_2\ldots W_\ell X)\|_{\mathrm{F}} > 2\|Y\|_{\mathrm{F}}$, then

$$\sqrt{f(W)} = \|W_1\phi(W_2\ldots W_\ell X) - Y\|_{\mathrm{F}} \geq \|W_1\phi(W_2\ldots W_\ell X)\|_{\mathrm{F}} - \|Y\|_{\mathrm{F}}$$

$$\geq \frac{\|W_1\phi(W_2\ldots W_\ell X)\|_{\mathrm{F}}}{2} \geq \sqrt{h}b\sqrt{\lambda_{\min}(XX^\top)}\frac{\|W_1\|_{\mathrm{F}}^{\ell-1}}{2d^{\frac{\ell-2}{2}}}.$$

The last inequality follows by inequality (26).

In this case, it holds

$$\|W_1\|_{\mathrm{F}}^{2\ell-2} \leq \frac{4}{hb^2\lambda_{\min}(XX^\top)}d^{\ell-2}f(W).$$

and

$$\|W_1\|_{\mathrm{F}}^{\ell-2} \leq \left(\frac{4}{hb^2\lambda_{\min}(XX^\top)}d^{\ell-2}f(W)\right)^{\frac{\ell-2}{2\ell-2}}.$$

By equation (25), we have

$$\|\nabla^2 f(W)\|_2 \leq \frac{8\ell^2}{hb^2\lambda_{\min}(XX^\top)}d^{\ell-2}f(W)\|X\|_2^2$$

$$+ 2(\ell^2 - \ell)\left(\frac{4}{hb^2\lambda_{\min}(XX^\top)}d^{\ell-2}\right)^{\frac{\ell-2}{2\ell-2}}f(W)^{\frac{2\ell-3}{2\ell-2}}\|X\|_2. \qquad (28)$$

Summing the right hand sides of inequalities (27) and (28)

$$\|\nabla^2 f(W)\|_2 \leq \frac{8\ell^2 d^{\ell-2}\|Y\|_{\mathrm{F}}^2}{hb^2\lambda_{\min}(XX^\top)}\|X\|_2^2 + 2(\ell^2-\ell)\left(\frac{4d^{\ell-2}\|Y\|_{\mathrm{F}}^2}{hb^2\lambda_{\min}(XX^\top)}\right)^{\frac{\ell-2}{2\ell-2}}\|X\|_2\sqrt{f(W)} +$$

$$\frac{8\ell^2}{hb^2\lambda_{\min}(XX^\top)}d^{\ell-2}\|X\|_2^2 f(W) + 2(\ell^2-\ell)\left(\frac{4}{hb^2\lambda_{\min}(XX^\top)}d^{\ell-2}\right)^{\frac{\ell-2}{2\ell-2}}f(W)^{\frac{2\ell-3}{2\ell-2}}\|X\|_2.$$

It holds $\frac{2\ell-3}{2\ell-2} \leq 1$ and we take the following cases:

If $f(W) < 1$, we have

$$\|\nabla^2 f(W)\|_2 \leq \frac{8\ell^2 d^{\ell-2}\|Y\|_{\mathrm{F}}^2}{hb^2\lambda_{\min}(XX^\top)}\|X\|_2^2 + 2(\ell^2-\ell)\left(\frac{4d^{\ell-2}\|Y\|_{\mathrm{F}}^2}{hb^2\lambda_{\min}(XX^\top)}\right)^{\frac{\ell-2}{2\ell-2}}\|X\|_2$$

$$+ 2(\ell^2-\ell)\left(\frac{4}{hb^2\lambda_{\min}(XX^\top)}d^{\ell-2}\right)^{\frac{\ell-2}{2\ell-2}}\|X\|_2 + \frac{8\ell^2}{hb^2\lambda_{\min}(XX^\top)}d^{\ell-2}\|X\|_2^2 f(W).$$

If $f(W) \geq 1$, then we have

$$\|\nabla^2 f(W)\|_2 \leq \frac{8\ell^2 d^{\ell-2}\|Y\|_{\mathrm{F}}^2}{hb^2\lambda_{\min}(XX^\top)}\|X\|_2^2 + \left(\frac{8\ell^2}{hb^2\lambda_{\min}(XX^\top)}d^{\ell-2}\|X\|_2^2 + \right.$$

$$2(\ell^2-\ell)\left(\frac{4d^{\ell-2}\|Y\|_{\mathrm{F}}^2}{hb^2\lambda_{\min}(XX^\top)}\right)^{\frac{\ell-2}{2\ell-2}}\|X\|_2 + 2(\ell^2-\ell)\left(\frac{4}{hb^2\lambda_{\min}(XX^\top)}d^{\ell-2}\right)^{\frac{\ell-2}{2\ell-2}}\|X\|_2\right)f(W).$$

Thus, $\|\nabla^2 f(W)\|_2$ is always upper bounded by the sum of the last two bounds. Incorporating $\ell^2 - \ell \leq \ell^2$, we derive

$$\|\nabla^2 f(W)\|_2 \le \ell^2 \left( \frac{16d^{\ell-2}\|Y\|_F^2}{hb^2\lambda_{\min}(XX^\top)}\|X\|_2^2 + 2\left(\frac{4d^{\ell-2}\|Y\|_F^2}{hb^2\lambda_{\min}(XX^\top)}\right)^{\frac{\ell-2}{2\ell-2}}\|X\|_2 \right.$$

$$\left. + 2\left(\frac{4}{hb^2\lambda_{\min}(XX^\top)}d^{\ell-2}\right)^{\frac{\ell-2}{2\ell-2}}\|X\|_2 \right)$$

$$+ \ell^2 \left( \frac{16}{hb^2\lambda_{\min}(XX^\top)}d^{\ell-2}\|X\|_2^2 + 2\left(\frac{4d^{\ell-2}\|Y\|_F^2}{hb^2\lambda_{\min}(XX^\top)}\right)^{\frac{\ell-2}{2\ell-2}}\|X\|_2 \right.$$

$$\left. + \left(\frac{4}{hb^2\lambda_{\min}(XX^\top)}d^{\ell-2}\right)^{\frac{\ell-2}{2\ell-2}}\|X\|_2 \right) f(W),$$

which is the desired result.

$\square$

**Proposition C.1.** *Let $f$ be defined as*

$$f(W) \equiv f(W_1,...,W_\ell) = \|Y - \underbrace{W_1\phi_1(W_2\phi(W_3\ldots\phi_{\ell-1}(W_\ell X)\ldots))}_{F}\|_F^2$$

*where $\phi_i$ is leaky-ReLU activation function with slopes 1 and $b_i$, i.e., $\phi_i(x) = \max\{b_i x, x\}$, $0 < b_i \le 1$, and matrices $Y, X, \{W_i\}_{i=1}^\ell$ defined as in Proposition 3.2. Assume that over the course of GD:*

- $\lambda_{\min}(W_i^\top W_i) \ge h_i > 0$, *for $i = 1,\ldots,\ell-1$.*
- *The layers $W_i$ are weakly balanced, i.e., $\|W_1\|_F = \ldots = \|W_\ell\|_F$.*

*Then, $f$ satisfies*

$$\|\nabla^2 f(W)\|_2 \le H_0 + H_1 f(W)^{\ell-1}(\le H_0 + 2^{\ell-2}H_1 f^* + 2^{\ell-2}H_1(f(W) - f^*)^{\ell-1}),$$

*where*

$$H_0 := 2\ell(\ell-1)\left(\frac{2\|Y\|_F}{\sqrt{\lambda_{\min}(XX^T)}\Pi_{i=1}^{\ell-1}\sqrt{h_i}b_i}\right)^{\ell-2}\|X\|_2 + 4\ell^2\left(\frac{2\|Y\|_F}{\sqrt{\lambda_{\min}(XX^T)}\Pi_{i=1}^{\ell-1}\sqrt{h_i}b_i}\right)^{2\ell-2}\|X\|_2^2 +$$

$$\frac{2\ell(\ell-1)4^{(\ell-2)/2}}{(\lambda_{\min}(XX^T)\Pi_{i=1}^{\ell-1}h_i b_i^2)^{(\ell-2)/2}}\|X\|_2 + \frac{2\ell^2 4^{(2\ell-2)/2}}{(\lambda_{\min}(XX^T)\Pi_{i=1}^{\ell-1}h_i b_i^2)^{(2\ell-2)/2}}\|X\|_2^2$$

*and*

$$H_1 := 2\ell(\ell-1)\left(\frac{2\|Y\|_F}{\sqrt{\lambda_{\min}(XX^T)}\Pi_{i=1}^{\ell-1}\sqrt{h_i}b_i}\right)^{\ell-2}\|X\|_2 + \frac{2\ell(\ell-1)4^{(\ell-2)/2}}{(\lambda_{\min}(XX^T)\Pi_{i=1}^{\ell-1}h_i b_i^2)^{(\ell-2)/2}}\|X\|_2$$

$$+ \frac{2\ell^2 4^{(2\ell-2)/2}}{(\lambda_{\min}(XX^T)\Pi_{i=1}^{\ell-1}h_i b_i^2)^{(2\ell-2)/2}}\|X\|_2^2.$$

*Proof.* We adopt again the notation

$$Z_\ell = W_\ell X$$
$$A_{\ell-1} = \phi_{\ell-1}(Z_\ell)$$
$$Z_{\ell-1} = W_{\ell-1}A_{\ell-1}$$
$$\vdots$$
$$Z_2 = W_2 A_2$$
$$A_1 = \phi_1(Z_2)$$
$$Z_1 = W_1 A_1 = F.$$

Similarly to the proof of Proposition 3.2, we can obtain the bound

$$\|\nabla^2 f(W)\|_2 \leq 2\ell(\ell-1)\|W_1\|_F^{\ell-2}\|X\|_2\sqrt{f(W)} + 2\ell^2\|W_1\|_F^{2\ell-2}\|X\|_2^2. \tag{29}$$

This is because the analysis of this part in the proof of Proposition 3.2 is valid for a general deep non-linear network.

We now move to a lower bound for the loss value. For $i = 1, \ldots, \ell-2$, we have

$$\|W_i A_i\|_F^2 \geq \lambda_{\min}(W_i^T W_i)\|A_i\|_F^2 \geq h_i\|\phi_i(W_{i+1}A_{i+1})\|_F^2 \geq h_i b_i^2\|W_{i+1}A_{i+1}\|_F^2$$

and by induction,

$$\begin{aligned}
\|W_1 A_1\|_F^2 &\geq \left(\Pi_{i=1}^{\ell-2} h_i b_i^2\right)\|W_{\ell-1}A_{\ell-1}\|_F^2 \\
&\geq \left(\Pi_{i=1}^{\ell-2} h_i b_i^2\right)\lambda_{\min}(W_{\ell-1}^T W_{\ell-1})\|A_{\ell-1}\|_F^2 \\
&= \left(\Pi_{i=1}^{\ell-2} h_i b_i^2\right)\lambda_{\min}(W_{\ell-1}^T W_{\ell-1})\|\phi_{\ell-1}(W_\ell X)\|_F^2 \\
&\geq \left(\Pi_{i=1}^{\ell-2} h_i b_i^2\right) h_{\ell-1} b_{\ell-1}^2 \lambda_{\min}(XX^T)\|W_\ell\|_F^2 \\
&= \left(\Pi_{i=1}^{\ell-2} h_i b_i^2\right) h_{\ell-1} b_{\ell-1}^2 \lambda_{\min}(XX^T)\|W_1\|_F^2 \\
&= \left(\Pi_{i=1}^{\ell-1} h_i b_i^2\right)\lambda_{\min}(XX^T)\|W_1\|_F^2. \tag{30}
\end{aligned}$$

We have repeatedly used the assumption that $\lambda_{\min}(W_i W_i^T) \geq h_i$ and that

$$\|\phi_i(S)\|_F^2 \geq b_i^2\|S\|_F^2,$$

for any matrix $S$, as we did in the proof of Proposition 3.2.

To derive inequality (30), we also used the weak balancedness assumption, that is, all $\|W_i\|_F$ have the same norm.

We proceed by considering the following cases:

- If $\|W_1 A_1\|_F \leq 2\|Y\|_F$, we have by inequality (30) that

$$\|W_1\|_F \leq \frac{2\|Y\|_F}{\sqrt{\lambda_{\min}(XX^T)}\Pi_{i=1}^{\ell-1}\sqrt{h_i}b_i},$$

  thus (by inequality (29))

$$\begin{aligned}
\|\nabla^2 f(W)\|_F &\leq 2\ell(\ell-1)\left(\frac{2\|Y\|_F}{\sqrt{\lambda_{\min}(XX^T)}\Pi_{i=1}^{\ell-1}\sqrt{h_i}b_i}\right)^{\ell-2}\|X\|_2\sqrt{f(W)} \\
&\quad + 2\ell^2\left(\frac{2\|Y\|_F}{\sqrt{\lambda_{\min}(XX^T)}\Pi_{i=1}^{\ell-1}\sqrt{h_i}b_i}\right)^{2\ell-2}\|X\|_2^2.
\end{aligned}$$

- If $\|W_1 A_1\|_F > 2\|Y\|_F$, we write

$$f(W) = \|Y - W_1 A_1\|_F^2 \geq (\|W_1 A_1\|_F - \|Y\|_F)^2 \geq \frac{\|W_1 A_1\|_F^2}{4}.$$

  By inequality (30), it holds

$$\|W_1\|_F^2 \leq \frac{4f(W)}{\lambda_{\min}(XX^T)\Pi_{i=1}^{\ell-1}h_i b_i^2}.$$

  Combining with inequality (29), we get

$$\begin{aligned}
\|\nabla^2 f(W)\|_2 &\leq \frac{2\ell(\ell-1)4^{(\ell-2)/2}}{(\lambda_{\min}(XX^T)\Pi_{i=1}^{\ell-1}h_i b_i^2)^{(\ell-2)/2}}(f(W))^{(\ell-2)/2}\|X\|_2\sqrt{f(W)} \\
&\quad + \frac{2\ell^2 4^{(2\ell-2)/2}}{(\lambda_{\min}(XX^T)\Pi_{i=1}^{\ell-1}h_i b_i^2)^{(2\ell-2)/2}}(f(W))^{(2\ell-2)/2}\|X\|_2^2.
\end{aligned}$$

Merging the two cases together, we get

$$\|\nabla^2 f(W)\|_2 \leq 2\ell(\ell-1)\left(\frac{2\|Y\|_F}{\sqrt{\lambda_{\min}(XX^T)}\Pi_{i=1}^{\ell-1}\sqrt{h_i}b_i}\right)^{\ell-2}\|X\|_2\sqrt{f(W)}$$

$$+ 2\ell^2\left(\frac{2\|Y\|_F}{\sqrt{\lambda_{\min}(XX^T)}\Pi_{i=1}^{\ell-1}\sqrt{h_i}b_i}\right)^{2\ell-2}\|X\|_2^2$$

$$+ \frac{2\ell(\ell-1)4^{(\ell-2)/2}}{(\lambda_{\min}(XX^T)\Pi_{i=1}^{\ell-1}h_ib_i^2)^{(\ell-2)/2}}(f(W))^{(\ell-1)/2}\|X\|_2$$

$$+ \frac{2\ell^2 4^{(2\ell-2)/2}}{(\lambda_{\min}(XX^T)\Pi_{i=1}^{\ell-1}h_ib_i^2)^{(2\ell-2)/2}}(f(W))^{(2\ell-2)/2}\|X\|_2^2.$$

We can write this in a more compact form, considering that if $f(W) \leq 1$, then

$$\|\nabla^2 f(W)\|_F \leq 2\ell(\ell-1)\left(\frac{2\|Y\|_F}{\sqrt{\lambda_{\min}(XX^T)}\Pi_{i=1}^{\ell-1}\sqrt{h_i}b_i}\right)^{\ell-2}\|X\|_2$$

$$+ 2\ell^2\left(\frac{2\|Y\|_F}{\sqrt{\lambda_{\min}(XX^T)}\Pi_{i=1}^{\ell-1}\sqrt{h_i}b_i}\right)^{2\ell-2}\|X\|_2^2$$

$$+ \frac{2\ell(\ell-1)4^{(\ell-2)/2}}{(\lambda_{\min}(XX^T)\Pi_{i=1}^{\ell-1}h_ib_i^2)^{(\ell-2)/2}}\|X\|_2$$

$$+ \frac{2\ell^2 4^{(2\ell-2)/2}}{(\lambda_{\min}(XX^T)\Pi_{i=1}^{\ell-1}h_ib_i^2)^{(2\ell-2)/2}}\|X\|_2^2$$

and if $f(W) > 1$, we have

$$\|\nabla^2 f(W)\|_2 \leq 2\ell^2\left(\frac{2\|Y\|_F}{\sqrt{\lambda_{\min}(XX^T)}\Pi_{i=1}^{\ell-1}\sqrt{h_i}b_i}\right)^{2\ell-2}\|X\|_2^2$$

$$+ \left(2\ell(\ell-1)\left(\frac{2\|Y\|_F}{\sqrt{\lambda_{\min}(XX^T)}\Pi_{i=1}^{\ell-1}\sqrt{h_i}b_i}\right)^{\ell-2}\|X\|_2\right.$$

$$+ \frac{2\ell(\ell-1)4^{(\ell-2)/2}}{(\lambda_{\min}(XX^T)\Pi_{i=1}^{\ell-1}h_ib_i^2)^{(\ell-2)/2}}\|X\|_2$$

$$\left.+ \frac{2\ell^2 4^{(2\ell-2)/2}}{(\lambda_{\min}(XX^T)\Pi_{i=1}^{\ell-1}h_ib_i^2)^{(2\ell-2)/2}}\|X\|_2^2\right)f(W)^{\ell-1}.$$

Summing the two expressions, we get that in any case

$$\|\nabla^2 f(W)\|_2 \leq$$

$$2\ell(\ell-1)\left(\frac{2\|Y\|_F}{\sqrt{\lambda_{\min}(XX^T)}\Pi_{i=1}^{\ell-1}\sqrt{h_i}b_i}\right)^{\ell-2}\|X\|_2 + 4\ell^2\left(\frac{2\|Y\|_F}{\sqrt{\lambda_{\min}(XX^T)}\Pi_{i=1}^{\ell-1}\sqrt{h_i}b_i}\right)^{2\ell-2}\|X\|_2^2$$

$$+ \frac{2\ell(\ell-1)4^{(\ell-2)/2}}{(\lambda_{\min}(XX^T)\Pi_{i=1}^{\ell-1}h_ib_i^2)^{(\ell-2)/2}}\|X\|_2 + \frac{2\ell^2 4^{(2\ell-2)/2}}{(\lambda_{\min}(XX^T)\Pi_{i=1}^{\ell-1}h_ib_i^2)^{(2\ell-2)/2}}\|X\|_2^2$$

$$+ \left(2\ell(\ell-1)\left(\frac{2\|Y\|_F}{\sqrt{\lambda_{\min}(XX^T)}\Pi_{i=1}^{\ell-1}\sqrt{h_i}b_i}\right)^{\ell-2}\|X\|_2 + \frac{2\ell(\ell-1)4^{(\ell-2)/2}}{(\lambda_{\min}(XX^T)\Pi_{i=1}^{\ell-1}h_ib_i^2)^{(\ell-2)/2}}\|X\|_2\right.$$

$$\left.+ \frac{2\ell^2 4^{(2\ell-2)/2}}{(\lambda_{\min}(XX^T)\Pi_{i=1}^{\ell-1}h_ib_i^2)^{(2\ell-2)/2}}\|X\|_2^2\right)f(W)^{\ell-1}.$$

This is the desired result.

$\square$

**Proposition 3.3.** *Consider a 2-layer neural network with MSE loss and L2 regularization:*

$$f(W) \equiv f(W_1, W_2) = \|Y - W_1\phi(W_2X)\|_{\mathrm{F}}^2 + \frac{\lambda_1}{2}\|W_1\|_{\mathrm{F}}^2 + \frac{\lambda_2}{2}\|W_2\|_{\mathrm{F}}^2,$$

*where $\phi$ is an activation function, such that $|\phi(x)| \leq C_1|x|, |\phi'(x)| \leq C_2$ and $|\phi''(x)| \leq C_3$ for all $x \in \mathbb{R}$, and matrices $Y, W_1, W_2$ are defined as before. Then, it holds*

$$\|\nabla^2 f(W)\|_2 \leq H_0 + H_1 f(W) \ \ (= H_0 + H_1 f^* + H_1(f(W) - f^*)),$$

*for $H_0$ and $H_1$ defined as in equations (31) and (32) respectively.*

$$H_0 := 2C_2\|X\|_2 + \lambda_1 + \lambda_2 \tag{31}$$

and

$$H_1 := \frac{4}{\lambda_1}(2C_2^2 + C_3 + 2C_1C_2)\|X\|_2^2 + \frac{8}{\lambda_2}(C_1^2 + C_1C_2)\|X\|_2^2 + 2C_3\|X\|_2^2 + 2C_2\|X\|_2. \tag{32}$$

*Proof.* We will recompute the Hessian of $L$ from scratch, since our calculation in the proof of Proposition 3.2 involves only getting an upper bound for its Frobenius norm and only in the case of piecewise linear activation functions and balanced weights. In the two-layer case, we can easily obtain an explicit form.

We denote $\bar{f}(W) := \|Y - W_1\phi(W_2X)\|_{\mathrm{F}}^2$. Then,

$$\|\nabla^2 f(W)\|_2 \leq \|\nabla^2 \bar{f}(W)\|_2 + (\lambda_1 + \lambda_2).$$

We proceed by computing an upper bound for $\|\nabla^2 \bar{f}(W)\|_{\mathrm{F}}$.

$\nabla^2 \bar{f}(W)$ is a block matrix of the form

$$\begin{bmatrix} \frac{\partial^2 \bar{f}}{\partial \mathrm{vec}(W_1)\mathrm{vec}(W_1)^\top} & \frac{\partial^2 \bar{f}}{\partial \mathrm{vec}(W_1)\mathrm{vec}(W_2)^\top} \\ \frac{\partial^2 \bar{f}}{\partial \mathrm{vec}(W_2)\mathrm{vec}(W_1)^\top} & \frac{\partial^2 \bar{f}}{\partial \mathrm{vec}(W_2)\mathrm{vec}(W_2)^\top} \end{bmatrix}.$$

For all computations, we work with a vectorized version of $\bar{f}$:

$$\bar{f}(W_1, W_2) = \|\mathrm{vec}(Y) - \mathrm{vec}(W_1\phi(W_2X))\|_{\mathrm{F}}^2.$$

Let us denote

$$R := \mathrm{vec}(Y) - \mathrm{vec}(W_1\phi(W_2X)) = \mathrm{vec}(Y) - (\phi(W_2X)^\top \otimes I_c)\mathrm{vec}(W_1).$$

For the second inequality, we used a classic property between vectorization and the Kronecker product.

The derivative with respect to $\mathrm{vec}(W_1)$ is

$$\frac{\partial \bar{f}}{\partial \mathrm{vec}(W_1)} = \frac{\partial \bar{f}}{\partial R} \cdot \frac{\partial R}{\partial \mathrm{vec}(W_1)} = -2R^\top(\phi(W_2X)^\top \otimes I_c) = -2R^\top(\phi(W_2X)^\top \otimes I_c).$$

Transposing in order to bring the vector in column form, we get

$$\frac{\partial \bar{f}}{\partial \mathrm{vec}(W_1)} = 2(\phi(W_2X) \otimes I_c)R = -2\mathrm{vec}((Y - W_1\phi(W_2X))\phi(W_2X)^\top). \tag{33}$$

The gradient with respect to $W_2$ is similarly

$$\frac{\partial \bar{f}}{\partial \mathrm{vec}(W_2)} = \frac{\partial \bar{f}}{\partial R} \cdot \frac{\partial R}{\partial \mathrm{vec}(W_2)}.$$

$\frac{\partial \bar{f}}{\partial R}$ is again $2R^\top$. In order to deal with $\frac{\partial R}{\partial \text{vec}(W_2)}$, we write

$$R = \text{vec}(Y) - (I_m \otimes W_1)\text{vec}(\phi(W_2 X)).$$

Thus,

$$\frac{\partial R}{\partial \text{vec}(W_2)} = -(I_m \otimes W_1)\frac{\partial \text{vec}(\phi(W_2 X))}{\partial \text{vec}(W_2)} = -(I_m \otimes W_1)\frac{\partial \text{vec}(\phi(W_2 X))}{\partial \text{vec}(W_2 X)}\frac{\partial \text{vec}(W_2 X)}{\partial \text{vec}(W_2)}$$

$\frac{\partial \text{vec}(\phi(W_2 X))}{\partial \text{vec}(W_2)}$ is the diagonal matrix $\text{diag}(\text{vec}(\phi'(W_2 X))$.

Since $\text{vec}(W_2 X) = (X^\top \otimes I_{n_1})\text{vec}(W_2)$, the gradient $\frac{\partial \text{vec}(W_2 X)}{\partial \text{vec}(W_2)}$ is

$$\frac{\partial \text{vec}(W_2 X)}{\partial \text{vec}(W_2)} = X^\top \otimes I_{n_1}.$$

Putting it all together, we have

$$\frac{\partial \bar{f}}{\partial \text{vec}(W_2)} = -2R^\top(I_m \otimes W_1)\text{diag}(\text{vec}(\phi'(W_2 X))(X^\top \otimes I_{n_1}). \tag{34}$$

Writing that again as column vector yields

$$-2(X \otimes I_{n_1})\text{diag}(\text{vec}(\phi'(W_2 X)))(I_m \otimes W_1^\top)R.$$

After some modifications, we can write

$$\text{diag}(\text{vec}(\phi'(W_2 X)))(I_m \otimes W_1^\top)R =$$
$$\text{diag}(\text{vec}(\phi'(W_2 X)))\text{vec}(W_1^\top(Y - W_1\phi(W_2 X)))) =$$
$$\text{vec}((W_1^\top(Y - W_1\phi(W_2 X)) \odot \phi'(W_2 X)).$$

where $\odot$ is the Hadamard product.

This means that we can write the previous gradient as

$$-2\text{vec}(((W_1^\top(Y - W_1\phi(W_2 X))) \odot \phi'(W_2 X))X^\top).$$

For the first block, we differentiate $\frac{\partial \bar{f}}{\partial \text{vec}(W_1)}$ with respect to $\partial \text{vec}(W_1)^\top$. Since

$$\frac{\partial \bar{f}}{\partial \text{vec}(W_1)} = -2\text{vec}((Y - W_1\phi(W_2 X))\phi(W_2 X)^\top) = -2(\phi(W_2 X) \otimes I_c)\text{vec}(Y - W_1\phi(W_2 X)),$$

we have

$$\frac{\partial^2 \bar{f}}{\partial \text{vec}(W_1)\text{vec}(W_1)^\top} = -2(\phi(W_2 X) \otimes I_c)\frac{\partial \text{vec}(Y - W_1\phi(W_2 X))}{\partial \text{vec}(W_1)^\top}$$

$$= 2(\phi(W_2 X) \otimes I_c)(\phi(W_2 X)^\top \otimes I_c)\frac{\partial \text{vec}(W_1)}{\partial \text{vec}(W_1)^\top}$$

$$= 2(\phi(W_2 X)\phi(W_2 X)^\top \otimes I_c).$$

For the off-diagonal blocks, it suffices to compute only one, as they are symmetric to each other. We use the product rule (see Magnus (1985), Theorem 9)

$$\frac{\partial \text{vec}(A(W)B(W))}{\partial \text{vec}(W)^\top} = (B(W)^\top \otimes I)\frac{\partial \text{vec}(A(W))}{\partial \text{vec}(W)^\top} + (I \otimes A(W))\frac{\partial \text{vec}(B(W))}{\partial \text{vec}(W)^\top}.$$

We have

$$\frac{\partial}{\partial \text{vec}(W_2)^\top}\frac{\partial \bar{f}}{\partial \text{vec}(W_1)} = -2(\phi(W_2 X) \otimes I_c)\frac{\partial \text{vec}(Y - W_1\phi(W_2 X))}{\partial \text{vec}(W_2)^\top}$$

$$-2(I_{n_1} \otimes (Y - W_1\phi(W_2 X)))\frac{\partial \text{vec}(\phi(W_2 X)^\top)}{\partial \text{vec}(W_2)^\top}.$$

In order to proceed, we need to write $\text{vec}(\phi(W_2X)^\top)$ in terms of $\text{vec}(\phi(W_2X))$, and this can be done formally using the so-called commutation matrix:

$$\text{vec}(\phi(W_2X)^\top) = K_{n_1m}\text{vec}(\phi(W_2X)).$$

For the first partial derivative in the sum, we have

$$\frac{\partial\text{vec}(Y - W_1\phi(W_2X))}{\partial\text{vec}(W_2)^\top} = -\frac{\partial\text{vec}(W_1\phi(W_2X))}{\partial\text{vec}(W_2)^\top} = -(I_m \otimes W_1)\frac{\partial\text{vec}(\phi(W_2X))}{\partial\text{vec}(W_2)^\top}$$

$$= -(I_m \otimes W_1)\text{diag}(\text{vec}(\phi'(W_2X)))\frac{\partial\text{vec}(W_2X)}{\partial\text{vec}(W_2)^\top}$$

$$= -(I_m \otimes W_1)\text{diag}(\text{vec}(\phi'(W_2X)))(X^\top \otimes I_{n_1}).$$

As it is evident in the previous calculation

$$\frac{\partial\text{vec}(\phi(W_2X))}{\partial\text{vec}(W_2)^\top} = \text{diag}(\text{vec}(\phi'(W_2X)))(X^\top \otimes I_{n_1}).$$

Putting it all together, we get

$$\frac{\partial^2\bar{f}}{\partial\text{vec}(W_1)\text{vec}(W_2)^\top} =$$

$$2(\phi(W_2X) \otimes W_1)\text{diag}(\text{vec}(\phi'(W_2X)))(X^\top \otimes I_{n_1})$$

$$-2(I_{n_1} \otimes (Y - W_1\phi(W_2X)))K_{n_1m}\text{diag}(\text{vec}(\phi'(W_2X)))(X^\top \otimes I_{n_1}) =$$

$$2(\phi(W_2X) \otimes W_1 + (I_{n_1} \otimes (W_1\phi(W_2X) - Y))K_{n_1m})\text{diag}(\text{vec}(\phi'(W_2X)))(X^\top \otimes I_{n_1}).$$

We also have

$$\frac{\partial^2\bar{f}}{\partial\text{vec}(W_2)\text{vec}(W_1)^\top} = \left(\frac{\partial^2\bar{f}}{\partial\text{vec}(W_1)\text{vec}(W_2)^\top}\right)^\top.$$

For the second derivative of $L$ with respect to $W_2$, we remind that

$$\frac{\partial\bar{f}}{\partial\text{vec}(W_2)} = -2(X \otimes I_{n_1})\text{diag}(\text{vec}(\phi'(W_2X)))(I_m \otimes W_1^\top)R.$$

Differentiating that with respect to $\text{vec}(W_2)^\top$ involves a product rule, as $W_2$ appears in $\text{diag}(\text{vec}(\phi'(W_2X)))$ and in $R$. It is more convenient to bring $\frac{\partial\bar{f}}{\partial\text{vec}(W_2)}$ back in fully vectorized form as:

$$\frac{\partial\bar{f}}{\partial\text{vec}(W_2)} = -2\text{vec}(((W_1^\top(Y - W_1\phi(W_2X))) \odot \phi'(W_2X))X^\top).$$

We have

$$-2\frac{\partial\text{vec}(((W_1^\top(Y - W_1\phi(W_2X))) \odot \phi'(W_2X))X^\top)}{\partial\text{vec}(W_2)^\top} =$$

$$-2(X \otimes I_{n_1})\left(\frac{\partial\text{vec}(W_1^\top(Y - W_1\phi(W_2X)) \odot \phi'(W_2X))}{\partial\text{vec}(W_2)^\top}\right).$$

Now we can use the product rule for the Hadamard product, see Magnus (1985) (Theorem 10):

$$\frac{\partial\text{vec}((W_1^\top(Y - W_1\phi(W_2X))) \odot \phi'(W_2X))}{\partial\text{vec}(W_2)^\top} =$$

$$\text{diag}(\text{vec}(\phi'(W_2X))\frac{\partial\text{vec}(W_1^\top(Y - W_1\phi(W_2X)))}{\partial\text{vec}(W_2)^\top} + \text{diag}(\text{vec}(W_1^\top(Y - W_1\phi(W_2X))))\frac{\partial\phi'(W_2X)}{\partial\text{vec}(W_2)^\top}.$$

For the first term of the last sum, we have by previous calculations that

$$\frac{\partial \text{vec}(W_1^\top(Y - W_1\phi(W_2X)))}{\partial \text{vec}(W_2)^\top} = -(I_m \otimes W_1^\top W_1)\text{diag}(\text{vec}(\phi'(W_2X)))(X^\top \otimes I_{n_1}).$$

For the second term of the last sum, we have

$$\frac{\partial \phi'(W_2X)}{\partial \text{vec}(W_2)^\top} = \text{diag}(\text{vec}(\phi''(W_2X)))(X^\top \otimes I_{n_1}).$$

In total, we have

$$\frac{\partial^2 \bar{f}}{\partial \text{vec}(W_2)\text{vec}(W_2)^\top} =$$

$$2(X \otimes I_{n_1})\text{diag}(\text{vec}(\phi'(W_2X)))(I_m \otimes W_1^\top W_1)\text{diag}(\text{vec}(\phi'(W_2X)))(X^\top \otimes I_{n_1})$$

$$-2(X \otimes I_{n_1})\text{diag}(\text{vec}(W_1^\top(Y - W_1\phi(W_2X))))\text{diag}(\text{vec}(\phi''(W_2X)))(X^\top \otimes I_{n_1}). \tag{35}$$

This completes the calculation of all four blocks of the Hessian.

We can now upper bound the spectral norm of the Hessian as

$$\|\nabla^2\bar{f}(W)\|_2 \le \left\|\frac{\partial^2\bar{f}}{\partial\text{vec}(W_1)\text{vec}(W_1)^\top}\right\|_2 + 2\left\|\frac{\partial^2\bar{f}}{\partial\text{vec}(W_1)\text{vec}(W_2)^\top}\right\|_2 + \left\|\frac{\partial^2\bar{f}}{\partial\text{vec}(W_2)\text{vec}(W_2)^\top}\right\|_2.$$

This is a special case of inequality (7).

It holds

$$\left\|\frac{\partial^2\bar{f}}{\partial\text{vec}(W_1)\text{vec}(W_1)^\top}\right\|_2 \le 2\|\phi(W_2X)\phi(W_2X)^\top\|_2^2$$

$$\le 2\|\phi(W_2X)\phi(W_2X)^\top\|_F^2$$

$$= 2C_1^2\|W_2\|_F^2\|X\|_2^2,$$

$$\left\|\frac{\partial^2\bar{f}}{\partial\text{vec}(W_1)\text{vec}(W_2)^\top}\right\|_2 \le 2(\|\phi(W_2X)\|_2\|W_1\|_2 + \|W_1\phi(W_2X) - Y\|_2)C_2\|X\|_2$$

$$\le 2(\|\phi(W_2X)\|_F\|W_1\|_F + \|W_1\phi(W_2X) - Y\|_F)C_2\|X\|_2$$

$$\le 2C_2(C_1\|W_1\|_F\|W_2\|_F\|X\|_2 + \|W_1\phi(W_2X) - Y\|_F)\|X\|_2$$

$$\le 2C_2(C_1\|W_1\|_F^2\|X\|_2 + C_1\|W_2\|_F^2\|X\|_2 + \|W_1\phi(W_2X) - Y\|_F)\|X\|_2$$

and

$$\left\|\frac{\partial^2\bar{f}}{\partial\text{vec}(W_2)\text{vec}(W_2)^\top}\right\|_2 \le 2\|X\|_2^2C_2^2\|W_1^\top W_1\|_2 + 2\|X\|_2^2C_3\|W_1^\top(Y - W_1\phi(W_2X))\|_2$$

$$\le 2\|X\|_2^2C_2^2\|W_1^\top W_1\|_F + 2\|X\|_2^2C_3\|W_1^\top(Y - W_1\phi(W_2X))\|_F$$

$$\le 2\|X\|_2^2C_2^2\|W_1\|_F^2 + 2\|X\|_2^2C_3\|W_1\|_F\|Y - W_1\phi(W_2X)\|_F$$

$$\le 2\|X\|_2^2C_2^2\|W_1\|_F^2 + \|X\|_2^2C_3\|W_1\|_F^2 + \|X\|_2^2C_3\|Y - W_1\phi(W_2X)\|_F^2.$$

Overall, we have

$$\|\nabla^2\bar{f}(W)\|_2 \le (2C_2^2 + C_3 + 4C_1C_2)\|X\|_2^2\|W_1\|_F^2 + 2(C_1^2 + 2C_1C_2)\|X\|_2^2\|W_2\|_F^2$$

$$+ 4C_2\|X\|_2\|W_1\phi(W_2X) - Y\|_F + C_3\|X\|_2^2\|Y - W_1\phi(W_2X)\|_F^2.$$

It is easy to verify that

$$\|\nabla^2\bar{f}(W)\|_2 \le \left(\frac{2}{\lambda_1}(2C_2^2 + C_3 + 4C_1C_2)\|X\|_2^2 + \frac{4}{\lambda_2}(C_1^2 + 2C_1C_2)\|X\|_2^2 + C_3\|X\|_2^2\right)f(W)$$

$$+ 4C_2\|X\|_2\sqrt{f(W)}.$$

This is because

$$\|W_1\|_F^2 \le \frac{2}{\lambda_1} f(W),$$

$$\|W_2\|_F^2 \le \frac{2}{\lambda_2} f(W)$$

and

$$\bar{f}(W) \le f(W).$$

In total, we get

$$\|\nabla^2 f(W)\|_2 \le \left( \frac{2}{\lambda_1}(2C_2^2 + C_3 + 4C_1C_2)\|X\|_2^2 + \frac{4}{\lambda_2}(C_1^2 + 2C_1C_2)\|X\|_2^2 + C_3\|X\|_2^2 \right) f(W)$$

$$+ 4C_2\|X\|_2\sqrt{f(W)} + (\lambda_1 + \lambda_2)$$

As usual, we can take the cases $f(W) < 1$ and $f(W) \ge 1$, sum the two right hand sides of the obtained inequalities and we derive that

$$\|\nabla^2 f(W)\|_2 \le H_0 + H_1 f(W)$$

where

$$H_0 = 4C_2\|X\|_2 + 2(\lambda_1 + \lambda_2)$$

and

$$H_1 = \frac{4}{\lambda_1}(2C_2^2 + C_3 + 4C_1C_2)\|X\|_2^2 + \frac{8}{\lambda_2}(C_1^2 + 2C_1C_2)\|X\|_2^2 + 2C_3\|X\|_2^2 + 4C_2\|X\|_2.$$

$\square$

**Proposition 3.4.** *Consider a 2-layer non-linear model with cross-entropy loss and L2 regularization:*

$$f(W) \equiv f(W_1, W_2) = -Y \log(P)^\top - (\mathbb{1} - Y) \log(\mathbb{1} - P)^\top + \frac{\lambda_1}{2}\|W_1\|_F^2 + \frac{\lambda_2}{2}\|W_2\|_F^2,$$

*where $Y \in \mathbb{R}^{1 \times m}$ are true labels, and $P = \sigma(W_1\phi(W_2 X))$ is the output of the model with the activation function $\phi$ such that $|\phi(x)| \le C_1|x|, |\phi'(x)| \le C_2$ and $|\phi''(x)| \le C_3$ for all $x \in \mathbb{R}$, sigmoid function $\sigma$, and weight matrices $W_1 \in \mathbb{R}^{1 \times n_1}, W_2 \in \mathbb{R}^{n_1 \times d}$. Then, it holds*

$$\|\nabla^2 f(W)\|_2 \le H_0 + H_1 f(W) \ \ (= H_0 + H_1 f^* + H_1(f(W) - f^*))$$

*for $H_0$ and $H_1$ defined as in equations (36) and (37) respectively.*

$$H_0 := \lambda_1 + \lambda_2 \tag{36}$$

and

$$H_1 := \frac{2}{\lambda_1}(C_2^2 + C_3 + 2C_1C_2)\|X\|_2^2 + \frac{2}{\lambda_2}(C_1^2 + 2C_1C_2)\|X\|_2^2 + 2C_2\|X\|_2 + C_3\|X\|_2^2. \tag{37}$$

*Proof.* We start by calculating the gradients and Hessians of $f$. The Hessian of the regularization part is just $(\lambda_1 + \lambda_2)I$. We denote the main part of the loss as

$$\bar{f}(W) = -Y \log(P)^\top - (\mathbb{1} - Y) \log(\mathbb{1} - P)^\top.$$

Again, it holds

$$\|\nabla^2 f(W)\|_2 \le \|\nabla^2 \bar{f}(W)\|_2 + (\lambda_1 + \lambda_2).$$

Some useful notation is

$$A := W_2 X$$
$$H := \phi(A)$$
$$Z := W_1 H$$
$$P := \sigma(Z).$$

The gradient of $\bar{L}$ with respect to $\mathrm{vec}(W_1)$ is

$$\frac{\partial \bar{f}}{\partial Z} \cdot \frac{\partial Z}{\partial \mathrm{vec}(W_1)}.$$

It holds

$$\frac{\partial \bar{f}}{\partial P} = -Y \odot \frac{1}{P} + (\mathbb{1} - Y) \odot \frac{1}{\mathbb{1} - P}$$

where 1/vector is used to denote entry-wise inversion.

We also have

$$\frac{\partial P}{\partial Z} = \sigma'(Z) = P \odot (\mathbb{1} - P).$$

Thus,

$$\frac{\partial \bar{f}}{\partial Z} = \frac{\partial \bar{f}}{\partial P} \odot \frac{\partial P}{\partial Z} = P - Y.$$

We denote the vectorized form of this term by $R$ since it plays the role of a residual. Since $P - Y$ is a row vector, its vectorized form is just its transpose, however, we will often keep the standard form $R = \mathrm{vec}(P - Y)$ to ensure compatibility with previous calculations.

It holds

$$\frac{\partial \bar{f}}{\partial \mathrm{vec}(W_1)} = \frac{\partial \bar{f}}{\partial Z} \frac{\partial Z}{\partial \mathrm{vec}(W_1)} = R^\top H^\top = R^\top \phi(W_2 X)^\top.$$

This is a row vector, thus we transpose it to bring it to column form:

$$\frac{\partial \bar{f}}{\partial \mathrm{vec}(W_1)} = HR = \mathrm{vec}((P - Y)H^\top) = \mathrm{vec}((P - Y)\phi(W_2 X)^\top)$$

For the partial derivative with respect to $\mathrm{vec}(W_2)$, we have

$$\frac{\partial \bar{f}}{\partial \mathrm{vec}(W_2)} = \frac{\partial \bar{f}}{\partial Z} \cdot \frac{\partial Z}{\partial \mathrm{vec}(W_2)} = R^\top \frac{\partial Z}{\partial \mathrm{vec}(W_2)}$$

and

$$\frac{\partial R}{\partial \mathrm{vec}(W_2)} = -(I_m \otimes W_1)\frac{\partial \mathrm{vec}(\phi(W_2 X))}{\partial \mathrm{vec}(W_2)} = -(I_m \otimes W_1)\frac{\partial \mathrm{vec}(\phi(W_2 X))}{\partial \mathrm{vec}(W_2 X)}\frac{\partial \mathrm{vec}(W_2 X)}{\partial \mathrm{vec}(W_2)}$$

$\frac{\partial \mathrm{vec}(\phi(W_2 X))}{\partial \mathrm{vec}(W_2)}$ is the diagonal matrix $\mathrm{diag}(\mathrm{vec}(\phi'(W_2 X)))$.

Since $\mathrm{vec}(W_2 X) = (X^\top \otimes I_{n_1})\mathrm{vec}(W_2)$, the gradient $\frac{\partial \mathrm{vec}(W_2 X)}{\partial \mathrm{vec}(W_2)}$ is

$$\frac{\partial \mathrm{vec}(W_2 X)}{\partial \mathrm{vec}(W_2)} = X^\top \otimes I_{n_1}.$$

Putting it all together, we have

$$\frac{\partial f}{\partial \mathrm{vec}(W_2)} = R^\top (I_m \otimes W_1)\mathrm{diag}(\mathrm{vec}(\phi'(W_2 X)))(X^\top \otimes I_{n_1}).$$

Writing that again as column vector yields

$$(X \otimes I_{n_1})\mathrm{diag}(\mathrm{vec}(\phi'(W_2 X)))(I_m \otimes W_1^\top)R.$$

After some modifications, we can write

$$\mathrm{diag}(\mathrm{vec}(\phi'(W_2 X)))(I_m \otimes W_1^\top)R =$$
$$\mathrm{diag}(\mathrm{vec}(\phi'(W_2 X)))\mathrm{vec}(W_1^\top(P - Y)) =$$
$$\mathrm{vec}(W_1^\top(P - Y) \odot \phi'(W_2 X)).$$

where $\odot$ is the Hadamard product.

This means that we can write the previous gradient as

$$-2\text{vec}(((W_1^\top(P-Y)) \odot \phi'(W_2 X))X^\top).$$

We now move to the calculation of the Hessian.

For the first block, we have

$$\frac{\partial^2 \bar{f}}{\partial\text{vec}(W_1)\text{vec}(W_1)^\top} = \phi(W_2 X)\frac{\partial R}{\partial\text{vec}(W_1)^\top}$$

$$= \phi(W_2 X)\frac{\partial\text{vec}(P-Y)}{\partial\text{vec}(W_1)^\top}$$

$$= \phi(W_2 X)\text{diag}(P \odot (\mathbb{1}-P))\phi(W_2 X)^\top.$$

For the off-diagonal blocks, it suffices to compute one of them, as they are symmetric.

We use the product rule (see Magnus (1985), Theorem 9)

$$\frac{\partial\text{vec}(A(W)B(W))}{\partial\text{vec}(W)^\top} = (B(W)^\top \otimes I)\frac{\partial\text{vec}(A(W))}{\partial\text{vec}(W)^\top} + (I \otimes A(W))\frac{\partial\text{vec}(B(W))}{\partial\text{vec}(W)^\top}.$$

We have

$$\frac{\partial}{\partial\text{vec}(W_2)^\top}\frac{\partial\bar{f}}{\partial\text{vec}(W_1)} = (\phi(W_2 X) \otimes I_1)\frac{\partial\text{vec}(P-Y)}{\partial\text{vec}(W_2)^\top}$$

$$+ (I_{n_1} \otimes (P-Y))\frac{\partial\text{vec}(\phi(W_2 X)^\top)}{\partial\text{vec}(W_2)^\top}.$$

In order to proceed, we need to write $\text{vec}(\phi(W_2 X)^\top)$ in terms of $\text{vec}(\phi(W_2 X))$, and this can be done formally using the so-called commutation matrix:

$$\text{vec}(\phi(W_2 X)^\top) = K_{n_1 m}\text{vec}(\phi(W_2 X)).$$

For the first partial derivative in the sum, we have

$$\frac{\partial\text{vec}(P-Y)}{\partial\text{vec}(W_2)^\top} = \frac{\partial\text{vec}(P)}{\partial\text{vec}(Z)}\frac{\partial\text{vec}(Z)}{\partial\text{vec}(W_2)^\top}$$

$$= \text{diag}(P \odot (\mathbb{1}-P))\frac{\partial\text{vec}(W_1\phi(W_2 X))}{\partial\text{vec}(W_2)^\top}$$

$$= \text{diag}(P \odot (\mathbb{1}-P))(I_m \otimes W_1)\frac{\partial\text{vec}(\phi(W_2 X))}{\partial\text{vec}(W_2)^\top}$$

$$= \text{diag}(P \odot (\mathbb{1}-P))(I_m \otimes W_1)\text{diag}(\text{vec}(\phi'(W_2 X)))\frac{\partial\text{vec}(W_2 X)}{\partial\text{vec}(W_2)^\top}$$

$$= \text{diag}(P \odot (\mathbb{1}-P))(I_m \otimes W_1)\text{diag}(\text{vec}(\phi'(W_2 X)))(X^\top \otimes I_{n_1}).$$

As it is evident in the previous calculation

$$\frac{\partial\text{vec}(\phi(W_2 X))}{\partial\text{vec}(W_2)^\top} = \text{diag}(\text{vec}(\phi'(W_2 X)))(X^\top \otimes I_{n_1}).$$

Putting it all together, we get

$$\frac{\partial^2 \bar{f}}{\partial\text{vec}(W_1)\text{vec}(W_2)^\top} = \phi(W_2 X)\text{diag}(P \odot (\mathbb{1}-P)(I_m \otimes W_1)\text{diag}(\text{vec}(\phi'(W_2 X)))(X^\top \otimes I_{n_1})$$

$$+ (I_{n_1} \otimes (P-Y))K_{n_1 m}\text{diag}(\text{vec}(\phi'(W_2 X)))(X^\top \otimes I_{n_1})$$

$$= (\phi(W_2 X)\text{diag}(P \odot (\mathbb{1}-P))(I_m \otimes W_1)$$

$$+ (I_{n_1} \otimes (P-Y)K_{n_1 m}))\text{diag}(\text{vec}(\phi'(W_2 X)))(X^\top \otimes I_{n_1}).$$

We conclude with the calculation of the last block. To differentiate $\text{vec}(((W_1^\top R) \odot \phi'(W_2 X))X^\top)$, we can use the product rule for the Hadamard product, see Magnus (1985) (Theorem 10):

$$\frac{\partial \text{vec}((W_1^\top R) \odot \phi'(W_2 X))}{\partial \text{vec}(W_2)^\top} = \text{diag}(\text{vec}(\phi'(W_2 X))\frac{\partial \text{vec}(W_1^\top R)}{\partial \text{vec}(W_2)^\top} + \text{diag}(\text{vec}(W_1^\top R))\frac{\partial \phi'(W_2 X)}{\partial \text{vec}(W_2)^\top}.$$

For the first term of the last sum, we have by previous calculations that

$$\frac{\partial \text{vec}(W_1^\top R)}{\partial \text{vec}(W_2)^\top} = (I_m \otimes W_1^\top)\text{diag}(P \odot (\mathbb{1} - P))(I_m \otimes W_1)\text{diag}(\text{vec}(\phi'(W_2 X)))(X^\top \otimes I_{n_1}).$$

For the second term of the last sum, we have

$$\frac{\partial \phi'(W_2 X)}{\partial \text{vec}(W_2)^\top} = \text{diag}(\text{vec}(\phi''(W_2 X)))(X^\top \otimes I_{n_1}).$$

In total, we have

$$\frac{\partial^2 \bar{f}}{\partial \text{vec}(W_2)\text{vec}(W_2)^\top} = (X \otimes I_{n_1})\text{diag}(\text{vec}(\phi'(W_2 X)))(I_m \otimes W_1^\top)\text{diag}(P \odot (\mathbb{1} - P))$$
$$(I_m \otimes W_1)\text{diag}(\text{vec}(\phi'(W_2 X)))(X^\top \otimes I_{n_1})$$
$$+ (X \otimes I_{n_1})\text{diag}(\text{vec}(W_1^\top R))\text{diag}(\text{vec}(\phi''(W_2 X)))(X^\top \otimes I_{n_1}).$$

This completes the calculation of all four blocks of the Hessian of $\bar{f}$.

To upper bound $\|\nabla^2 \bar{f}(W)\|_2$, we can write

$$\|\nabla^2 \bar{f}(W)\|_2 \leq \left\|\frac{\partial^2 \bar{f}}{\partial \text{vec}(W_1)\text{vec}(W_1)^\top}\right\|_2 + 2\left\|\frac{\partial^2 \bar{f}}{\partial \text{vec}(W_1)\text{vec}(W_2)^\top}\right\|_2 + \left\|\frac{\partial^2 \bar{f}}{\partial \text{vec}(W_2)\text{vec}(W_2)^\top}\right\|_2.$$

It holds

$$\left\|\frac{\partial^2 \bar{f}}{\partial \text{vec}(W_1)\text{vec}(W_1)^\top}\right\|_2 \leq \|\text{diag}(P \odot (\mathbb{1} - P))\|_2^2 \|\phi(W_2 X)\phi(W_2 X)^\top\|_2^2$$
$$\leq \|\text{diag}(P \odot (\mathbb{1} - P))\|_2^2 \|\phi(W_2 X)\phi(W_2 X)^\top\|_F^2 \leq C_1^2 \|W_2\|_F^2 \|X\|_2^2,$$

since all entries of $P \odot (\mathbb{1} - P)$ are upper bounded by 1 in absolute value.

For the off-diagonal blocks, it holds

$$\left\|\frac{\partial^2 \bar{f}}{\partial \text{vec}(W_1)\text{vec}(W_2)^\top}\right\|_2 \leq (\|\phi(W_2 X)\|_2 \|W_1\|_2 + \|P - Y\|_2)C_2 \|X\|_2$$
$$\leq C_2(C_1 \|W_1\|_F \|W_2\|_F \|X\|_2 + \|P - Y\|_F)\|X\|_2$$
$$\leq C_2(C_1 \|W_1\|_F^2 \|X\|_2 + C_1 \|W_2\|_F^2 \|X\|_2 + \bar{f}(W))\|X\|_2$$

and

$$\left\|\frac{\partial^2 \bar{f}}{\partial \text{vec}(W_2)\text{vec}(W_2)^\top}\right\|_2 \leq \|X\|_2^2 C_2^2 \|W_1^\top\|_2 \|W_1\|_2 + \|X\|_2^2 C_3 \|W_1^\top(P - Y)\|_2$$
$$\leq \|X\|_2^2 C_2^2 \|W_1\|_F^2 + \|X\|_2^2 C_3 \|W_1\|_F \|P - Y\|_F$$
$$\leq \|X\|_2^2 C_2^2 \|W_1\|_F^2 + \|X\|_2^2 C_3 \|W_1\|_F^2 + \|X\|_2^2 C_3 \|P - Y\|_F^2$$
$$\leq \|X\|_2^2 C_2^2 \|W_1\|_F^2 + \|X\|_2^2 C_3 \|W_1\|_F^2 + \|X\|_2^2 C_3 \bar{f}(W).$$

In the two previous bounds, we have used that

$$\|P - Y\|_F, \|P - Y\|_F^2 \leq \bar{f}(W)$$

which follow from simple inequalities between the logarithm and linear functions in the domain $[0, 1]$.

Putting it all together, we have

$\|\nabla^2 \bar{f}(W)\|_2 \leq$
$(C_2^2 + C_3 + 2C_1C_2)\|X\|_2^2\|W_1\|_F^2 + (C_1^2 + 2C_1C_2)\|X\|_2^2\|W_2\|_2^2 + (2C_2\|X\|_2 + C_3\|X\|_2^2)\bar{f}(W).$

It holds $\bar{f}(W) \leq f(W)$ (since the regularization part is nonnegative), thus

$$\begin{aligned}\|\nabla^2 \bar{f}(W)\|_2 \leq\ &(C_2^2 + C_3 + 2C_1C_2)\|X\|_2^2\|W_1\|_F^2 \\ &+ (C_1^2 + 2C_1C_2)\|X\|_2^2\|W_2\|_F^2 + (2C_2\|X\|_2 + C_3\|X\|_2^2)\bar{f}(W).\end{aligned}$$

It is now easy to see that,

$$\begin{aligned}\|\nabla^2 f(W)\|_2 \leq\ &\left(\frac{2}{\lambda_1}(C_2^2 + C_3 + 2C_1C_2)\|X\|_2^2 + \frac{2}{\lambda_2}(C_1^2 + 2C_1C_2)\|X\|_2^2\right. \\ &\left. + 2C_2\|X\|_2 + C_3\|X\|_2^2\right)f(W) + (\lambda_1 + \lambda_2).\end{aligned}$$

This is the desired result.

$\square$

## D    MISSING PROOFS FROM SECTION 3.1.2

We begin by establishing an $(H_0, H_1)$ condition for a one-layer transformer trained on a single task, considering both the MSE and cross-entropy losses. Passing to the in-context loss then follows from arguments common to both cases.

We briefly recall the specifics of the model applied to an initial matrix

$$Z_0 = \begin{bmatrix} x_1 & x_2 & \ldots & x_n & x_{\text{query}} \\ y_1 & y_2 & \ldots & y_n & 0 \end{bmatrix} \in \mathbb{R}^{(d+1)\times(n+1)}.$$

The model is defined by $Z_1 = Z_0 + \frac{1}{n}PZ_0M \cdot \phi(Z_0^T Q Z_0)$, where the trainable parameters are $P$ and $Q$. The output of the model is

$$\hat{y} = [Z_1]_{(d+1),(n+1)}.$$

Below, we show an $(H_0, H_1)$ property for the cost function based on the MSE loss on one task.

**Proposition D.1.** *Consider the aforementioned $1$-layer transformer model with a regularized loss:*

$$f(P, Q) = (\hat{y} - y_{true})^2 + \frac{\lambda_P}{2}\|P\|_F^2 + \frac{\lambda_Q}{2}\|Q\|_F^2.$$

*Assume that the input $Z$ is bounded, and the activation function and its derivatives are globally bounded:*

- $\|Z\|_2 \leq C_Z$

- $\|\phi(x)\| \leq C_1\|x\|$

- $\|\phi'(x)\| \leq C_2$

- $\|\phi''(x)\| \leq C_3$.

*Then, it holds*

$$\|\nabla^2 f(P,Q)\|_2 \le H_0 + H_1 f(P,Q),$$

*where*

$$H_0 = \frac{2}{n} C_Z^3 C_2 + \lambda_P + \lambda_Q$$

*and*

$$H_1 = \frac{2}{\lambda_P} \left( \frac{2}{n^2} C_Z^6 C_1 C_2 + \frac{2}{n^2} C_Z^6 C_1^2 + \frac{2}{n} C_Z^5 C_2 \right) + \frac{2}{\lambda_Q} \left( \frac{2}{n} C_Z^6 C_1^2 + \frac{2}{n^2} C_Z^6 C_1 C_1 \right) + \frac{2}{n} C_Z^5 C_2 + \frac{2}{n} C_Z^3 C_1.$$

*Proof.* To compute the Hessian, we first vectorize the parameters and the model. For simplicity, we set $k = d + 1$ and $m = n + 1$.

- Parameters: $p = \text{vec}(P) \in \mathbb{R}^{k^2 \times 1}$ and $q = \text{vec}(Q) \in \mathbb{R}^{k^2 \times 1}$.

- Constants: Let $V = ZM \in \mathbb{R}^{k \times m}$ and $E = e_k e_m^T \in \mathbb{R}^{k \times m}$.

- Intermediates: $S(Q) = Z^T QZ$ and $A(Q) = \phi(S(Q))$.

Using the trace identity $e_k^T X e_m = \text{Tr}(e_m e_k^T X) = \text{Tr}(E^T X)$, we rewrite $\hat{y}$:

$$\hat{y} = \frac{1}{n} \text{Tr}(E^T P V A(Q)) = \frac{1}{n} \text{Tr}((V A(Q) E^T) P).$$

Using $\text{Tr}(\mathbf{A}^T \mathbf{B}) = \text{vec}(\mathbf{A})^T \text{vec}(\mathbf{B})$, we get:

$$\hat{y} = \frac{1}{n} \text{vec}((V A(Q) E^T)^T)^T \text{vec}(P) = \frac{1}{n} \text{vec}(E A(Q)^T V^T)^T p.$$

Let $b(q) = \text{vec}(E A(Q)^T V^T)$. The prediction is linear in $p$:

$$\hat{y}(p,q) = \frac{1}{n} b(q)^T p.$$

We now expand $b(q)$ using the Kronecker product ($\otimes$) and commutation matrix ($K$):

1. $s(q) = \text{vec}(S(Q)) = \text{vec}(Z^T Q Z) = (Z^T \otimes Z^T) q$. Let $J_S = (Z^T \otimes Z^T)$.

2. $a(q) = \text{vec}(A(Q)) = \phi(s(q)) = \phi(J_S q)$.

3. $b(q) = \text{vec}(E A(Q)^T V^T) = (V \otimes E) \text{vec}(A(Q)^T)$.

4. $\text{vec}(A(Q)^T) = K_{m,m} \text{vec}(A(Q)) = K_{m,m} a(q)$.

Combining these, we define the constant matrix $J_A$:

$$b(q) = (V \otimes E) K_{m,m} a(q) = \underbrace{(V \otimes E) K_{m,m}}_{J_A} \phi(J_S q).$$

Our final vectorized prediction and loss are:

$$\hat{y}(p,q) = \frac{1}{n} (J_A \phi(J_S q))^T p \quad \text{and} \quad f(p,q) = \left( \frac{1}{n} b(q)^T p - y_{\text{true}} \right)^2.$$

Finally, we define the diagonal matrices of derivatives:

$$D'(q) = \text{diag}(\text{vec}(\phi'(J_S q))) \quad \text{and} \quad D''(q) = \text{diag}(\text{vec}(\phi''(J_S q))).$$

The gradient of the loss is $\nabla f = 2r \cdot \nabla \hat{y}$. We first compute $\nabla \hat{y}$.

$$\nabla_p \hat{y} = \frac{\partial \hat{y}}{\partial p} = \frac{\partial}{\partial p} \left( \frac{1}{n} b(q)^T p \right) = \frac{1}{n} b(q) = \frac{1}{n} J_A \phi(J_S q).$$

The loss gradient w.r.t $p$ is:

$$\nabla_p f = \frac{2r}{n} J_A \phi(J_S q).$$

We also have

$$\nabla_q \hat{y} = \frac{\partial \hat{y}}{\partial q} = \frac{\partial}{\partial q}\left(\frac{1}{n}p^T b(q)\right) = \frac{1}{n}\left(\frac{\partial b(q)}{\partial q}\right)^T p.$$

We need the Jacobian of $b(q)$, $\frac{\partial b(q)}{\partial q^T}$.

$$\frac{\partial b(q)}{\partial q^T} = \frac{\partial(J_A \phi(J_S q))}{\partial q^T} = J_A \frac{\partial(\phi(J_S q))}{\partial q^T} = J_A D'(q)\frac{\partial(J_S q)}{\partial q^T} = J_A D'(q)J_S$$

Plugging this back in (and transposing):

$$\nabla_q \hat{y} = \frac{1}{n}(J_A D'(q)J_S)^T p = \frac{1}{n}J_S^T D'(q)^T J_A^T p = \frac{1}{n}J_S^T D'(q)J_A^T p.$$

The loss gradient w.r.t $q$ is:

$$\nabla_q f = \frac{2r}{n} J_S^T D'(q) J_A^T p.$$

The Hessian $H_f$ of the loss $f = r^2$ is found using the chain rule and product rule:

$$H_f = \nabla(\nabla f)^T = \nabla(2r \cdot \nabla \hat{y})^T = 2(\nabla r)(\nabla \hat{y})^T + 2r\nabla(\nabla \hat{y})^T.$$

Since $\nabla r = \nabla(\hat{y} - y_{\text{true}}) = \nabla \hat{y}$, this simplifies to:

$$H_f = 2(\nabla \hat{y})(\nabla \hat{y})^T + 2r H_{\hat{y}}$$

where $H_{\hat{y}} = \nabla(\nabla \hat{y})^T$ is the Hessian of the prediction. We will compute the four blocks of $H_f$ by first finding the four blocks of $H_{\hat{y}}$.

$H_{\hat{y},pp}$: We differentiate $\nabla_p \hat{y}$ w.r.t $p^T$:

$$H_{\hat{y},pp} = \frac{\partial}{\partial p^T}(\nabla_p \hat{y}) = \frac{\partial}{\partial p^T}\left(\frac{1}{n}b(q)\right) = \mathbf{0}.$$

since $b(q)$ does not depend on $p$.

$H_{\hat{y},pq}$: We differentiate $\nabla_p \hat{y}$ w.r.t $q^T$:

$$H_{\hat{y},pq} = \frac{\partial}{\partial q^T}(\nabla_p \hat{y}) = \frac{\partial}{\partial q^T}\left(\frac{1}{n}b(q)\right) = \frac{1}{n}\frac{\partial b(q)}{\partial q^T}.$$

We have already computed $\frac{\partial b(q)}{\partial q^T}$:

$$H_{\hat{y},pq} = \frac{1}{n}J_A D'(q)J_S.$$

$H_{\hat{y},qp}$: This block is the transpose of the previous one:

$$H_{\hat{y},qp} = H_{\hat{y},pq}^T = \left(\frac{1}{n}J_A D'(q)J_S\right)^T = \frac{1}{n}J_S^T D'(q)J_A^T.$$

$H_{\hat{y},qq}$: We differentiate $\nabla_q \hat{y}$ w.r.t $q^T$:

$$H_{\hat{y},qq} = \frac{\partial}{\partial q^T}(\nabla_q \hat{y}) = \frac{\partial}{\partial q^T}\left(\frac{1}{n}J_S^T D'(q)J_A^T p\right).$$

Let $v = J_A^T p$ be a constant vector. We need to find $\frac{1}{n}J_S^T\left(\frac{\partial(D'(q)v)}{\partial q^T}\right)$. Using the rule for differentiating a diagonal matrix $\frac{\partial(\text{diag}(X)v)}{\partial y^T} = \text{diag}(v)\frac{\partial X}{\partial y^T}$, we have:

$$\frac{\partial(D'(q)v)}{\partial q^T} = \frac{\partial(\text{diag}(\text{vec}(\phi'(J_S q)))v)}{\partial q^T} = \text{diag}(v)\frac{\partial(\text{vec}(\phi'(J_S q)))}{\partial q^T}.$$

The derivative of the vector $\text{vec}(\phi'(J_S q)$ is $D''(q)J_S$.

$$\frac{\partial(D'(q)v)}{\partial q^T} = \text{diag}(v)D''(q)J_S = \text{diag}(J_A^T p)D''(q)J_S.$$

Plugging this back into the expression for $H_{\hat{y},qq}$:

$$H_{\hat{y},qq} = \frac{1}{n}J_S^T \text{diag}(J_A^T p)D''(q)J_S.$$

We now put together the four blocks of $H_f = 2(\nabla \hat{y})(\nabla \hat{y})^T + 2rH_{\hat{y}}$.

$$H_{pp} = 2(\nabla_p \hat{y})(\nabla_p \hat{y})^T + 2rH_{\hat{y},pp} = 2\left(\frac{1}{n}b(q)\right)\left(\frac{1}{n}b(q)\right)^T + \mathbf{0}.$$

$$H_{pp} = \frac{2}{n^2}(J_A \phi(J_S q))(J_A \phi(J_S q))^T$$

$$H_{qq} = 2(\nabla_q \hat{y})(\nabla_q \hat{y})^T + 2rH_{\hat{y},qq}$$

$$H_{qq} = \frac{2}{n^2}\left(\frac{1}{n}J_S^T D'(q)J_A^T p\right)\left(\frac{1}{n}J_S^T D'(q)J_A^T p\right)^T + \frac{2r}{n}\left[J_S^T \text{diag}(J_A^T p)D''(q)J_S\right]$$

$$H_{pq} = 2(\nabla_p \hat{y})(\nabla_q \hat{y})^T + 2rH_{\hat{y},pq}$$

$$H_{pq} = 2\left(\frac{1}{n}J_A \phi(J_S q)\right)\left(\frac{1}{n}J_S^T D'(q)J_A^T p\right)^T + 2r\left(\frac{1}{n}J_A D'(q)J_S\right)$$

$$H_{pq} = \frac{2}{n^2}(J_A \phi(J_S q))(p^T J_A D'(q)J_S) + \frac{2r}{n}(J_A D'(q)J_S)$$

$$H_{qp} = 2(\nabla_q \hat{y})(\nabla_p \hat{y})^T + 2rH_{\hat{y},qp} = H_{pq}^T$$

$$H_{qp} = \frac{2}{n^2}(J_S^T D'(q)J_A^T p)(J_A \phi(J_S q))^T + \frac{2r}{n}(J_S^T D'(q)J_A^T).$$

Now we pass in bounding the regulatized loss. We use the notation $f$ for the regulatized and $\bar{f}$ unregularized loss:

$$\bar{f}(p,q) = (\hat{y} - y_{true})^2 = r^2$$

$$f(p,q) = \bar{f}(p,q) + \frac{\lambda_P}{2}\|p\|_2^2 + \frac{\lambda_Q}{2}\|q\|_2^2.$$

The Hessian of the regularized loss $f$ is

$$\nabla^2 f(p,q) = \nabla^2 \bar{f}(p,q) + \begin{bmatrix} \lambda_P I & 0 \\ 0 & \lambda_Q I \end{bmatrix}.$$

Using the triangle inequality for the spectral norm ($\|\cdot\|_2$):

$$\|\nabla^2 f(p,q)\|_2 \le \|\nabla^2 \bar{f}(p,q)\|_2 + \left\|\begin{bmatrix} \lambda_P I & 0 \\ 0 & \lambda_Q I \end{bmatrix}\right\|_2 \le \|\nabla^2 \bar{f}(p,q)\|_2 + (\lambda_P + \lambda_Q).$$

We proceed by computing an upper bound for $\|\nabla^2 \bar{f}(p,q)\|_2$.

From our previous calculations, we have the four blocks:

$$H_{pp} = \frac{2}{n^2}(J_A \phi(J_S q))(J_A \phi(J_S q))^T$$

$$H_{qq} = \frac{2}{n^2}(J_S^T D'(q)J_A^T p)(J_S^T D'(q)J_A^T p)^T + \frac{2r}{n}\left[J_S^T \text{diag}(J_A^T p)D''(q)J_S\right]$$

$$H_{pq} = \frac{2}{n^2}(J_A \phi(J_S q))(p^T J_A D'(q)J_S) + \frac{2r}{n}(J_A D'(q)J_S)$$

$$H_{qp} = H_{pq}^T$$

We now upper bound the spectral norm of the full Hessian by the sum of the norms of its blocks:

$$\|\nabla^2 \bar{f}(p,q)\|_2 \le \|H_{pp}\|_2 + 2\|H_{pq}\|_2 + \|H_{qq}\|_2.$$

Since $\|Z\|_2 \le C_Z$, it holds $\|J_S\|_2 \le C_Z^2$ and $\|J_A\|_2 \le C_Z$.

**Bounding $H_{pp}$:**   $H_{pp}$ is a rank-1 matrix.

$$\|H_{pp}\|_2 = \frac{2}{n^2}\|J_A\phi(J_Sq)\|_2^2 \leq \frac{2}{n^2}\|J_A\|_2^2\|\phi(J_Sq)\|_2^2 \leq \frac{2}{n}C_Z^6C_1^2\|q\|_2^2.$$

**Bounding $H_{pq}$:**   We use the triangle inequality.

$$\|H_{pq}\|_2 \leq \left\|\frac{2}{n^2}(J_A\phi(J_Sq))(p^TJ_AD'(q)J_S)\right\|_2 + \left\|\frac{2r}{n}(J_AD'(q)J_S)\right\|_2$$

$$\leq \frac{2}{n^2}\|J_A\phi(J_Sq)\|_2\cdot\|p^TJ_AD'(q)J_S\|_2 + \frac{2|r|}{n}\|J_AD'(q)J_S\|_2$$

$$\leq \frac{2}{n^2}(C_Z^3C_1\|q\|_2)\cdot(\|p\|_2\|J_A\|_2\|D'(q)\|_2\|J_S\|_2) + \frac{2|r|}{n}(\|J_A\|_2\|D'(q)\|_2\|J_S\|_2)$$

$$\leq \frac{2}{n^2}(C_Z^3C_1\|q\|_2)\cdot(\|p\|_2C_2C_Z^3) + \frac{2|r|}{n}C_2C_Z^3$$

$$= \frac{2}{n^2}C_Z^6C_1C_2\|p\|_2\|q\|_2 + \frac{2}{n}C_2C_Z^3|r|.$$

**Bounding $H_{qq}$:**   We use the triangle inequality.

$$\|H_{qq}\|_2 \leq \frac{2}{n^2}\|J_S^TD'(q)J_A^Tp\|_2^2 + \frac{2|r|}{n}\|J_S^T\text{diag}(J_A^Tp)D''(q)J_S\|_2$$

$$\leq \frac{2}{n^2}(\|J_S\|_2\|D'(q)\|_2\|J_A\|_2\|p\|_2)^2 + \frac{2|r|}{n}(\|J_S\|_2^2\|\text{diag}(J_A^Tp)\|_2\|D''(q)\|_2)$$

$$\leq \frac{2}{n^2}(C_2C_Z^3\|p\|_2)^2 + \frac{2|r|}{n}((C_Z^2)^2\cdot(\|J_A\|_2\|p\|_2)\cdot C_3)$$

$$= \frac{2}{n^2}C_2^2C_Z^6\|p\|_2^2 + \frac{2}{n}C_Z^5C_3|r|\|p\|_2.$$

**Overall Bound for $\bar{f}$:**

$$\|\nabla^2\bar{f}\|_2 \leq \|H_{pp}\|_2 + 2\|H_{pq}\|_2 + \|H_{qq}\|_2$$

$$\leq \underbrace{\frac{2}{n}C_Z^6C_1^2\|q\|_2^2}_{K_1} + \underbrace{\frac{2}{n^2}C_Z^6C_1C_2\|p\|_2\|q\|_2}_{K_2} + \underbrace{\frac{2}{n}C_2C_Z^3|r|}_{K_3} + \underbrace{\frac{2}{n^2}C_2^2C_Z^6\|p\|_2^2}_{K_4} + \underbrace{\frac{2}{n}C_Z^5C_3|r|\|p\|_2}_{K_5}.$$

Using $2ab \leq a^2 + b^2$ and $ab \leq \frac{1}{2}(a^2 + b^2)$:

$$\|\nabla^2\bar{f}\|_2 \leq K_1\|q\|_2^2 + K_2(\|p\|_2^2 + \|q\|_2^2) + K_3|r| + K_4\|p\|_2^2 + K_5(r^2 + \|p\|_2^2)$$

$$= (K_2 + K_4 + K_5)\|p\|_2^2 + (K_1 + K_2)\|q\|_2^2 + K_3|r| + K_5r^2$$

$$= (K_2 + K_4 + K_5)\|p\|_2^2 + (K_1 + K_2)\|q\|_2^2 + K_3\sqrt{\bar{f}} + K_5f.$$

Now, we relate this to the regularized loss $f(p, q)$:

$$\|p\|_2^2 \leq \frac{2}{\lambda_P}f(p, q)$$

$$\|q\|_2^2 \leq \frac{2}{\lambda_Q}f(p, q)$$

$$\bar{f} \leq f(p, q) \quad \text{and} \quad \sqrt{\bar{f}} \leq \sqrt{f(p, q)}.$$

Substituting these in:

$$\|\nabla^2\bar{f}\|_2 \leq (K_2 + K_4 + K_5)\left(\frac{2}{\lambda_P}f\right) + (K_1 + K_2)\left(\frac{2}{\lambda_Q}f\right) + K_3\sqrt{f} + K_5f$$

$$= \underbrace{\left(\frac{2(K_2 + K_4 + K_5)}{\lambda_P} + \frac{2(K_1 + K_2)}{\lambda_Q} + K_5\right)}_{A}f + \underbrace{K_3}_{B}\sqrt{f}.$$

Finally, we bound the full Hessian $\|\nabla^2 f\|_2$:

$$\|\nabla^2 f(p,q)\|_2 \leq \|\nabla^2 \bar{f}(p,q)\|_2 + (\lambda_P + \lambda_Q)$$
$$\leq A f(p,q) + B\sqrt{f(p,q)} + (\lambda_P + \lambda_Q).$$

We can handle $\sqrt{f}$ by noting that $B\sqrt{f} \leq B(1 + f)$.

$$\|\nabla^2 f(p,q)\|_2 \leq A f(p,q) + B(1 + f(p,q)) + (\lambda_P + \lambda_Q)$$
$$= (A + B)f(p,q) + (B + \lambda_P + \lambda_Q).$$

This is of the form $\|\nabla^2 f(W)\|_2 \leq H_0 + H_1 f(W)$, where

$$H_1 = A + B$$
$$H_0 = B + \lambda_P + \lambda_Q.$$

This completes the proof. For recovering the specific expressions for $H_0$ and $H_1$ it suffices to substitute the expressions for $A$ and $B$. $\qquad\square$

We now examine the cross-entropy case. The underlying transformer model architecture stays unchanged, but the single scalar prediction (previously $\hat{y}$) is now the logit, which we will call $z$:

$$z(P,Q) = \frac{1}{n} e_k^T \left( P(ZM)\phi(Z^T Q Z) \right) e_m.$$

For a binary classification problem ($y_{\text{true}} \in \{0,1\}$), we apply a sigmoid function $\sigma(\cdot)$ to the logit $z$ to get the probability $p_1$:

$$p_1 = \sigma(z) = \frac{1}{1 + e^{-z}}.$$

The unregularized loss $\bar{f}$ is the binary cross-entropy (BCE):

$$\bar{f}(P,Q) = -(y_{\text{true}} \log(p_1) + (1 - y_{\text{true}}) \log(1 - p_1)).$$

**Proposition D.2.** *Consider the 1-layer transformer with the regularized cross-entropy loss $f(P,Q)$:*

$$f(P,Q) = -(y_{true} \log(p_1) + (1 - y_{true}) \log(1 - p_1)) + \frac{\lambda_P}{2}\|P\|_F^2 + \frac{\lambda_Q}{2}\|Q\|_F^2.$$

*Under the same assumptions as in Proposition D.1, it holds that*

$$\|\nabla^2 f(P,Q)\|_2 \leq H_0 + H_1 f(P,Q),$$

*where*

$$H_0 := \lambda_P + \lambda_Q$$

*and*

$$H_1 := \frac{2}{\lambda_p}\left(\frac{1}{n^2}C_1^2 C_Z^6 + \frac{1}{n^2}C_1^2 C_Z^6 + \frac{1}{n}C_2 C_Z^5\right) + \frac{2}{\lambda_q}\left(\frac{1}{n^2}C_1^2 C_Z^6 + \frac{1}{n^2}C_1 C_2 C_Z^6\right) + \frac{C_2 C_Z^3}{n} + \frac{C_2 C_Z^5}{n}.$$

*Proof.* We start as previously, by calculating the Hessian of $\bar{f}$. We use the chain rule for calculating the gradient.

$$\frac{\partial \bar{f}}{\partial z} = \frac{\partial \bar{f}}{\partial p_1} \cdot \frac{\partial p_1}{\partial z}$$

$$\frac{\partial \bar{f}}{\partial p_1} = -\frac{y_{\text{true}}}{p_1} + \frac{1 - y_{\text{true}}}{1 - p_1}$$

$$\frac{\partial p_1}{\partial z} = \sigma'(z) = \sigma(z)(1 - \sigma(z)) = p_1(1 - p_1).$$

Combining them, we get

$$\frac{\partial \bar{f}}{\partial z} = \left(-\frac{y_{\text{true}}}{p_1} + \frac{1 - y_{\text{true}}}{1 - p_1}\right) p_1(1 - p_1) = -y_{\text{true}}(1 - p_1) + (1 - y_{\text{true}})p_1 = p_1 - y_{\text{true}}.$$

This is the classic, simple residual for cross-entropy. We denote $r := p - y_{\text{true}}$. Similarly as in the proof of Proposition 3.4, we have

$$|r|, r^2 \leq \bar{f}.$$

The gradients of $\bar{f}$ are now simple:

$$\nabla_p \bar{f} = \frac{\partial \bar{f}}{\partial z}\frac{\partial z}{\partial p} = r \cdot \nabla_p z \qquad \text{and} \qquad \nabla_q \bar{f} = \frac{\partial \bar{f}}{\partial z}\frac{\partial z}{\partial q} = r \cdot \nabla_q z$$

where $\nabla_p z$ and $\nabla_q z$ are the same as $\nabla_p \hat{y}$ and $\nabla_q \hat{y}$ from the previous derivation for the MSE loss.

We use the product rule to find the Hessian $H_{\bar{f}} = \nabla(\nabla \bar{f})^T = \nabla(r \cdot \nabla z)^T$:

$$H_{\bar{f}} = (\nabla r)(\nabla z)^T + r \cdot H_z$$

where $H_z = \nabla(\nabla z)^T$ is the Hessian of the prediction (identical to $H_{\hat{y}}$ in the MSE case). We need $\nabla r$:

$$\nabla r = \nabla(p_1 - y_{\text{true}}) = \nabla p_1 = \frac{\partial p_1}{\partial z}\nabla z = \sigma'(z)\nabla z.$$

Let $\sigma' = \sigma'(z) = p_1(1 - p_1)$. This scalar can be bounded as:

$$0 \leq \sigma' \leq 1.$$

Substituting this in, the Hessian for the unregularized loss is:

$$H_{\bar{f}} = (\sigma' \nabla z)(\nabla z)^T + rH_z = \sigma'(\nabla z)(\nabla z)^T + rH_z.$$

The four blocks are assembled from the components $\nabla z$ and $H_z$ (which we already calculated), scaled by the bounded scalars $\sigma' \leq 1$ and $|r| \leq \bar{f}$.

$$H_{pp} = \sigma'(\nabla_p z)(\nabla_p z)^T + rH_{z,pp} = \sigma'(\nabla_p z)(\nabla_p z)^T + \mathbf{0}$$
$$H_{qq} = \sigma'(\nabla_q z)(\nabla_q z)^T + rH_{z,qq}$$
$$H_{pq} = \sigma'(\nabla_p z)(\nabla_q z)^T + rH_{z,pq}$$
$$H_{qp} = H_{pq}^T$$

We now prove the bound for the regularized loss $f = \bar{f} + \frac{\lambda_P}{2}\|p\|_2^2 + \frac{\lambda_Q}{2}\|q\|_2^2$.

The regularized Hessian is $H_f = H_{\bar{f}} + \text{diag}(\lambda_P I, \lambda_Q I)$. Thus,

$$\|\nabla^2 f(p, q)\|_2 \leq \|\nabla^2 \bar{f}(p, q)\|_2 + (\lambda_P + \lambda_Q).$$

We bound $\|\nabla^2 \bar{f}\|_2 \leq \|H_{pp}\|_2 + 2\|H_{pq}\|_2 + \|H_{qq}\|_2$.

We use the norm bounds on $\nabla z$ and $H_z$ from the proof of Proposition D.1 (with $C_Z = \|Z\|_2$):

- $\|\nabla_p z\|_2 \leq \frac{1}{n}C_1 C_Z^3 \|q\|_2$

- $\|\nabla_q z\|_2 \leq \frac{1}{n}C_2 C_Z^3 \|p\|_2$

- $\|H_{z,pq}\|_2 \leq \frac{1}{n}C_2 C_Z^3$

- $\|H_{z,qq}\|_2 \leq \frac{1}{n}C_3 C_Z^5 \|p\|_2$

**Bounding $H_{pp}$:**

$$\|H_{pp}\|_2 = \|\sigma'(\nabla_p z)(\nabla_p z)^T\|_2 \leq \sigma'\|\nabla_p z\|_2^2 \leq \left(\frac{1}{n}C_1 C_Z^3 \|q\|_2\right)^2$$

$$\|H_{pp}\|_2 \leq \frac{1}{n^2}C_1^2 C_Z^6 \|q\|_2^2$$

**Bounding $H_{pq}$:**

$$\|H_{pq}\|_2 \leq \|\sigma'(\nabla_p z)(\nabla_q z)^T\|_2 + \|r H_{z,pq}\|_2 \leq \sigma'\|\nabla_p z\|_2\|\nabla_q z\|_2 + |r|\|H_{z,pq}\|_2.$$

Using $\sigma' \leq 1$ and $|r| \leq \bar{f}$:

$$\|H_{pq}\|_2 \leq \left(\frac{C_1 C_Z^3}{n}\|q\|_2\right)\left(\frac{C_2 C_Z^3}{n}\|p\|_2\right) + \frac{C_2 C_Z^3}{n}\bar{f}$$

$$\|H_{pq}\|_2 \leq \frac{1}{n^2} C_1 C_2 C_Z^6 \|p\|_2 \|q\|_2 + \frac{C_2 C_Z^3}{n}\bar{f}$$

**Bounding $H_{qq}$:**

$$\|H_{qq}\|_2 \leq \|\sigma'(\nabla_q z)(\nabla_q z)^T\|_2 + \|r H_{z,qq}\|_2 \leq \sigma'\|\nabla_q z\|_2^2 + |r|\|H_{z,qq}\|_2$$

$$\|H_{qq}\|_2 \leq \left(\frac{C_1 C_Z^3}{n}\|p\|_2\right)^2 + \left(\frac{C_2 C_Z^5}{n}\|p\|_2\right)\sqrt{\bar{f}}$$

$$\|H_{qq}\|_2 \leq \frac{1}{n^2} C_1^2 C_Z^6 \|p\|_2^2 + \frac{C_2 C_Z^5}{n}\|p\|_2\sqrt{\bar{f}}$$

**Overall Bound for $\bar{f}$:** Combining these terms, $\|\nabla^2 \bar{f}\|_2$ is bounded by a quadratic polynomial in $\|p\|_2$ and $\|q\|_2$, with coefficients that depend on $C_Z$.

$$\|\nabla^2 \bar{f}\|_2 \leq \|H_{pp}\|_2 + 2\|H_{pq}\|_2 + \|H_{qq}\|_2$$

$$\leq K_1\|q\|_2^2 + K_2\|p\|_2\|q\|_2 + K_3\bar{f} + K_4\|p\|_2^2 + K_5\|p\|_2\sqrt{\bar{f}}$$

where $K_1, \ldots, K_5$ are constants depending on $n, C_Z, C_1, C_2, C_3$. Using $2ab \leq a^2 + b^2$ and $a \leq \frac{1}{2}(1 + a^2)$, we can simplify this to:

$$\|\nabla^2 \bar{f}\|_2 \leq A_p\|p\|_2^2 + A_q\|q\|_2^2 + B\bar{f}$$

where $A_p$, $A_q$, and $B$ are new positive constants that depend on the $K_i$.

**Final Step:** We now relate this to the regularized loss $f$. We use that

$$\|p\|_2^2 \leq \frac{2}{\lambda_P}f(p, q)$$

$$\|q\|_2^2 \leq \frac{2}{\lambda_Q}f(p, q).$$

Substituting into our bound for $\bar{f}$:

$$\|\nabla^2 \bar{f}\|_2 \leq A_p\left(\frac{2}{\lambda_P}f\right) + A_q\left(\frac{2}{\lambda_Q}f\right) + B.$$

Now, we bound the full regularized Hessian $H_f$:

$$\|\nabla^2 f\|_2 \leq \|\nabla^2 \bar{f}\|_2 + (\lambda_P + \lambda_Q) \leq \left(\frac{2A_p}{\lambda_P} + \frac{2A_q}{\lambda_Q} + B\right)f + (\lambda_P + \lambda_Q).$$

This is of the form $H_0 + H_1 f$, where:

$$H_1 = \frac{2A_p}{\lambda_P} + \frac{2A_q}{\lambda_Q} + B$$

$$H_0 = \lambda_P + \lambda_Q.$$

This completes the proof. For getting the exact expressions in the statement, we can just substitute the values of $A_p, A_q$ and $B$. $\qquad\square$

We close the discussion of this section with out main result, i.e. proving an $(H_0, H_1)$ condition for the *in-context* loss function over a distribution of tasks. Before we get to its proof, we present some useful lemmas from the theory of random matrices.

**Lemma 1.** *A sub-Gaussian random variable is also sub-exponential.*

**Lemma 2.** *For a random matrix $Z$ whose entries are independent and sub-exponential (with zero mean), its spectral norm $\|Z\|_2$ exhibits sub-exponential tails. This means the probability of the norm being large decays very quickly.*

$$\mathbb{P}(\|Z\|_2 > t) \leq C \exp(-ct) \quad \text{for large } t,$$

*for constants $C, c$.*

**Lemma 3** (Finite Moments). *A random variable $X$ with sub-exponential tails (like $\|Z_j\|_2$) has finite moments:*

$$\mathbb{E}[|X|^k] < \infty \quad \text{for all } k \geq 1.$$

For a rigorous treatment of these topics, see Vershynin (2018).

Now, we are ready for the main proof.

**Proposition 3.5.** *Consider the transformer model as described before, for in-context learning in continuous variables with MSE loss, or in binary variables with CE loss. $\phi$ is assumed to satisfy the conditions of Proposition 3.4 and $f_j(P, Q)$ is the L2 regularized loss corresponding to the j-th prompt. Consider the regularized in-context loss function*

$$f(P, Q) = \mathbb{E}_j[f_j(P, Q)].$$

*Assume that $D_x$ is sub-gaussian and the distribution of $y = h(x)$ is sub-exponential. Then, it holds*

$$\|\nabla^2 f(P, Q)\|_2 \leq H_0 + H_1 f(P, Q)(= H_0 + H_1 f^* + H_1(f(P, Q) - f^*))$$

*for some positive and finite constants $H_0$ and $H_1$.*

*Proof.* For simplicity, we denote $\theta = (P, Q)$.

By standard regularity conditions, the Hessian of the expected loss is the expectation of the individual Hessians:

$$\nabla^2 f(\theta) = \nabla^2 \mathbb{E}_A[f_A(\theta)] = \mathbb{E}_j[\nabla^2 f_j(\theta)].$$

We begin by applying Jensen's Inequality, as the spectral norm $\|\cdot\|_2$ is a convex function.

$$\|\nabla^2 f(\theta)\|_2 = \|\mathbb{E}_j[\nabla^2 f_j(\theta)]\|_2 \leq \mathbb{E}_j[\|\nabla^2 f_j(\theta)\|_2].$$

Substituting the per-task bound from Proposition D.1 or D.2, we get

$$\mathbb{E}_j[\|\nabla^2 f_i(\theta)\|_2] \leq \mathbb{E}_j[H_{0,j} + H_{1,j} f_j(\theta)].$$

Using the linearity of expectation, we split the term:

$$\mathbb{E}_i[H_{0,j} + H_{1,j} f_j(\theta)] = \mathbb{E}_i[H_{0,j}] + \mathbb{E}_j[H_{1,j} f_j(\theta)].$$

The second term $\mathbb{E}_i[H_{1,j} f_A(\theta)]$ is problematic, as $H_{1,j}$ and $f_j(\theta)$ are dependent (both depend on the task $j$). We bound this product using the Cauchy-Schwarz Inequality:

$$\mathbb{E}_A[H_{1,j} f_j(\theta)] \leq \sqrt{\mathbb{E}_j[H_{1,j}^2] \cdot \mathbb{E}_j[f_j(\theta)^2]}.$$

This gives us the bound:

$$\|\nabla^2 f(\theta)\|_2 \leq \mathbb{E}_j[H_{0,j}] + \sqrt{\mathbb{E}_j[H_{1,j}^2] \cdot \mathbb{E}_j[f_j(\theta)^2]}.$$

Let's define two constants, $H_A$ and $H_B$:

$$H_A = \mathbb{E}_i[H_{0,j}]$$

$$H_B = \sqrt{\mathbb{E}_j[H_{1,j}^2]}.$$

From Lemma 3, all moments of $C_{Z,j} = \|Z_j\|_2$ are finite. Since $H_{0,j}$ and $H_{1,j}$ are polynomials in $C_{Z,j}$, their moments (e.g., $\mathbb{E}[H_{0,j}]$ and $\mathbb{E}[H_{1,j}^2]$) are also finite. Thus, $H_A$ and $H_B$ are finite constants.

Now, let's analyze the $\mathbb{E}_j[f_j(\theta)^2]$ term using the definition of variance:

$$\mathbb{E}_j[f_j(\theta)^2] = \text{Var}_j(f_j(\theta)) + (\mathbb{E}_j[f_j(\theta)])^2.$$

By definition, $f(\theta) = \mathbb{E}_j[f_j(\theta)]$. Let $V_f := \text{Var}_j(f_j(\theta))$, thus

$$\mathbb{E}_j[f_j(\theta)^2] = V_f + f(\theta)^2. \tag{38}$$

The variance $V_f$ is finite. Indeed, $V_f$ is finite if and only if the second moment of the unregularized loss, $\mathbb{E}_j[\bar{f}_j(\theta)^2]$, is finite. This is because the regularization term $\frac{\lambda}{2}\|\theta\|^2$ is constant with respect to the expectation $\mathbb{E}_j$.

We need to prove $\mathbb{E}_j[\bar{f}_j(\theta)^2] < \infty$ for both the MSE anc CE losses. This requires demonstrating that the highest necessary moment of the model's random components ($\hat{y}_j$ and $y_{\text{true},j}$) is finite.

Let $z$ be the model's output logit ($z = \hat{y}$ for both cases).

**Case 1: Mean Squared Error (MSE)**

$$\bar{f}_{\text{MSE}} = (z - y_{\text{true},j})^2.$$

The highest moment we require is the second moment of $\bar{f}_{\text{MSE}}$, which is $\mathbb{E}[\bar{f}_{\text{MSE}}^2] = \mathbb{E}[(z - y_{\text{true},j})^4]$. By the AM-GM inequality, this is bounded by $8(\mathbb{E}[z^4] + \mathbb{E}[y_{\text{true},j}^4])$.

**Case 2: Cross-Entropy (CE)**

$$\bar{f}_{\text{CE}} = -(y_{\text{true},j}\log(\sigma(z)) + (1 - y_{\text{true},j})\log(1 - \sigma(z))).$$

Since $\bar{f}_{\text{CE}}$ is bounded by a linear function of $|z|$ (i.e., $\bar{f}_{\text{CE}} \leq |z| + C_L$), we have $\bar{f}_{\text{CE}}^2 \leq C'(|z|^2 + C'')$. Therefore, the highest moment required is $\mathbb{E}[z^2]$. To unify with MSE, we show that $\mathbb{E}[\bar{f}_{\text{CE}}^2]$ is bounded by $\mathbb{E}[z^4] + \mathbb{E}[y_{\text{true},j}^4]$.

We require that $\mathbb{E}_i[z^4] < \infty$ and $\mathbb{E}_i[y_{\text{true},i}^4] < \infty$.

**Finiteness of $\mathbb{E}_i[y_{\text{true},i}^4]$:** The label $y_{\text{true},i}$ is drawn from a sub-exponential distribution (Assumption). By Lemma 3, all moments of $y_{\text{true},i}$ are finite. Thus, $\mathbb{E}[y_{\text{true},i}^4] < \infty$.

**Finiteness of $\mathbb{E}_i[z^4]$:** The logit $z$ is the output of the model, which is a complex function of the random data matrix $Z_j$.

$$z \leq C_\theta \cdot \text{poly}(\|Z_j\|_2)$$

Therefore, $\mathbb{E}[z^4]$ is bounded by a sum of moments of the spectral norm $\|Z_j\|_2$, such as $\mathbb{E}[\|Z_j\|_2^k]$.

Since the inputs $x$ are sub-Gaussian and the outputs $y$ are sub-exponential, we can combine Lemmas 1 and 3 and conclude that $\mathbb{E}[\|Z_j\|_2^k] < \infty$ for all $k \geq 1$.

Consequently, $\mathbb{E}_j[z^4]$ is finite.

Since $\mathbb{E}_j[\bar{f}_j(\theta)^2]$ is bounded by a sum of finite moments ($\mathbb{E}[z^4] + \mathbb{E}[y_{\text{true},j}^4]$), the expected squared loss is finite. This implies the variance $V_f$ is finite for both the MSE and Cross-Entropy loss objectives.

Substituting equation (38) back into our main inequality, we get

$$\|\nabla^2 f(\theta)\|_2 \leq H_A + H_B\sqrt{V_f + f(\theta)^2}.$$

Using the inequality $\sqrt{a + b} \leq \sqrt{a} + \sqrt{b}$ (for $a, b \geq 0$):

$$\|\nabla^2 f(\theta)\|_2 \leq H_A + H_B(\sqrt{V_f} + \sqrt{f(\theta)^2})$$
$$\leq (H_A + H_B\sqrt{V_f}) + H_B f(\theta),$$

which is the desired form. We can now define our final, finite constants $H_0$ and $H_1$:

$$H_0 = H_A + H_B \sqrt{V_f} < \infty$$
$$H_1 = H_B < \infty$$

This completes the proof. $\qquad\square$

## E   NEURAL NETWORKS ARE IN GENERAL NOT $(L_0, L_1)$-SMOOTH

In this section, we demonstrate that neural networks still violate the $(L_0, L_1)$-smoothness, even in the presence of L2 regularization or weight balancedness. We start with an example of a simple 2-layer neural network with L2 regularization when $(L_0, L_1)$-smoothness is violated for $L_0, L_1 \geq 0$.

**Proposition E.1.** *We consider a simple 2-layer neural network with MSE loss*

$$f(u, v) = \frac{1}{2}(u\sigma(v))^2 + \frac{\lambda_1}{2}u^2 + \frac{\lambda_2}{2}v^2,$$

*such that $\sigma(0) = 0, \sigma'(0) \neq 0$[3]. Then $(L_0, L_1)$-smoothness does not hold for any $L_0, L_1 \geq 0$.*

*Proof.* For this example, the gradient and the Hessian are

$$\nabla f(u, v) = \begin{bmatrix} u\sigma^2(v) + \lambda_1 u \\ u^2\sigma(v)\sigma'(v) + \lambda_2 v \end{bmatrix}, \ \nabla^2 f(u, v) = \begin{bmatrix} \sigma^2(v) + \lambda_1 & 2u\sigma(v)\sigma'(v) \\ 2u\sigma(v)\sigma'(v) & u^2((\sigma'(v))^2 + \sigma(v)\sigma''(v)) + \lambda_2. \end{bmatrix}$$

Let us evaluate them at the point $(u, 0)$. Note that $\sigma(0) = 0, \sigma'(0) \neq 0$ by the assumption of the proposition. We obtain

$$\nabla f(u, v) = \begin{bmatrix} \lambda_1 u \\ 0 \end{bmatrix}, \ \nabla^2 f(u, v) = \begin{bmatrix} \lambda_1 & 0 \\ 0 & u^2(\sigma'(0))^2 + \lambda_2. \end{bmatrix}$$

Therefore, we obtain that

$$\|\nabla^2 f(u, 0)\|_2 = \max\{\lambda_1, u^2(\sigma'(0))^2 + \lambda_2\}, \quad \|\nabla f(u, 0)\| = \lambda_1|u|.$$

Thus, if $(L_0, L_1)$-smoothness was true for this function, then there were constants $L_0, L_1 \geq 0$ such that

$$\|\nabla^2 f(u, 0)\|_2 = \max\{\lambda_1, u^2(\sigma'(0))^2 + \lambda_2\} \leq L_0 + L_1\lambda_1|u|. \tag{39}$$

Let $u \geq \frac{\sqrt{\lambda_1}}{|\sigma'(0)|}$. Then $\|\nabla^2 f(u, 0)\|_2 = u^2(\sigma'(0))^2 + \lambda_2$. Therefore, dividing both sides of (39) by $u$ we obtain

$$u(\sigma'(0))^2 \leq \frac{L_0}{u} + L_1\lambda_1.$$

Taking $u \to +\infty$, we get that LHS goes to $+\infty$ while RHS goes to a constant. Therefore, $(L_0, L_1)$-smoothness is violated for any $L_0, L_1 \geq 0$. $\qquad\square$

Next, we demonstrate that $(L_0, L_1)$-smoothness is violated under a balancedness condition as well.

**Proposition E.2.** *We consider a 2-layer neural network with MSE loss*

$$f(W_1, W_2) = \|Y - W_1\phi(W_2 X)\|_F^2$$

*and leaky-ReLU or linear activation function, i.e. $\phi(x) = \max\{x, bx\}$, with $0 < b \leq 1$. Then, $(L_0, L_1)$-smoothness does not hold under weak balancedness for any $L_0, L_1 \geq 0$.*

---

[3]These assumptions are satisfied for several activation functions such as `tanh`, `GELU`, `SiLU`.

*Proof.* Take $X = \begin{bmatrix} 1 & 0 & 0 \\ 0 & 1 & 0 \\ 0 & 0 & 1 \end{bmatrix}$ and $Y = \begin{bmatrix} 1 & 0 & 0 \\ 0 & 2 & 0 \\ 0 & 0 & 3 \end{bmatrix}$. Take also $W_1 = \begin{bmatrix} t & 0 \\ 0 & 0 \\ 0 & 0 \end{bmatrix}$ and $W_2 = \begin{bmatrix} \frac{1}{t} & 0 & 0 \\ \sqrt{t^2 - 1/t^2} & 0 & 0 \end{bmatrix}$, for $t > 1$ (notice that the entries of $W_2$ are positive, thus it is not affected by leaky-ReLU). It holds $\|W_1\|_F = t = \|W_2\|_F$, thus we indeed satisfy the weak balancedness condition. It also holds

$$Y - W_1 W_2 X = \begin{bmatrix} 0 & 0 & 0 \\ 0 & 2 & 0 \\ 0 & 0 & 3 \end{bmatrix}.$$

We can use that to compute

$$W_1^T (Y - W_1 W_2 X) = \begin{bmatrix} \frac{1}{t} & 0 & 0 \\ 0 & 0 & 0 \end{bmatrix} \begin{bmatrix} 0 & 0 & 0 \\ 0 & 2 & 0 \\ 0 & 0 & 3 \end{bmatrix} = \begin{bmatrix} 0 & 0 & 0 \\ 0 & 0 & 0 \end{bmatrix}$$

and

$$(Y - W_1 W_2 X) X^T W_2^T = \begin{bmatrix} 0 & 0 & 0 \\ 0 & 2 & 0 \\ 0 & 0 & 3 \end{bmatrix} \begin{bmatrix} \frac{1}{t} & \sqrt{t^2 - 1/t^2} \\ 0 & 0 \\ 0 & 0 \end{bmatrix} = \begin{bmatrix} 0 & 0 \\ 0 & 0 \\ 0 & 0 \end{bmatrix}.$$

Since $\nabla_{W_1} f(W_1, W_2) = (Y - W_1 W_2 X) X^T W_2^T$ (by equation (33)) and $\nabla_{W_2} f(W_1, W_2) = W_1^T (Y - W_1 W_2 X)$ (by equation (34)), we have $\|\nabla f\| = 0$, while the Frobenius norm of the Hessian (thus also its spectral norm) goes to infinity at $t$ goes to infinity by equation (35), since

$$W_1^T W_1 = \begin{bmatrix} t^2 & 0 \\ 0 & 0 \end{bmatrix}.$$

□

**Remark:** For a network like the one of Proposition E.1, Proposition 3.3 guarantees that an $(H_0, H_1)$-condition holds. Similarly, for a network like the one of Proposition E.2, Proposition 3.2 guarantees that an $(H_0, H_1)$-condition holds as well.

## F   USEFUL LEMMAS

To obtain convergence results, we need to bound the smoothness between two arbitrary points $w, y \in \mathbb{R}^d$. This can be done using $(H_0, H_1)$-smoothness. If we parametrize $w(t) := x + t(y - w)$, then from the new smoothness assumption, the smoothness constant along the segment $[x, w(t)]$ can be bounded by a function of $\chi(t) := \int_0^t (H_0 + H_1(f(w(\theta)) - f^*)) d\theta$; see the next derivations for detailed proof. In particular, the bound on the smoothness constant along $[x, y]$ is related to $\chi(1)$. Our proof techniques are inspired by the results in Li et al. (2023). However, due to a more general smoothness inequality, our derivations involve a more careful analysis, because we need to deal with additional terms that do not appear in the case of $(L_0, L_1)$-smoothness. This highlights the difficulty of obtaining convergence guarantees, in particular, a gradient upper bound (Lemma 6) and quadratic upper bound (Lemma 7).

We start with a restatement of Lemma A.3 in Li et al. (2023), which can be seen as a generalization of Grönwall's inequality.

The next lemma provides sufficient conditions when the smoothness constant along the segment $[x, y]$ can be bounded by some constant, which uses information at $w$ only. The concurrent work Liu et al. (2025) provides similar derivations based on Li et al. (2023).

**Lemma 4** (Lemma A.3 in Li et al. (2023)). *Let $\alpha: [a, b] \to [0, \infty)$ and $\beta: [0, \infty) \to (0, \infty)$ be two continuous functions. Suppose $\alpha'(t) \leq \beta(\alpha(t))$ almost everywhere over $(a, b)$. We have for all $K \in [a, b]$,*

$$\int_{\alpha(a)}^{\alpha(t)} \frac{1}{\beta(u)} du \leq t - a.$$

**Lemma 5.** *Let $f$ be $(H_0, H_1)$-smooth, $\chi(t) := \int_0^t (H_0 + H_1(f(w(\theta)) - f^*))\mathrm{d}\theta$, and define $c_1 := H_0 + H_1(c_2 + f(w) - f^*)$ for some $c_2 > 0$. Then it holds*

$$\|\nabla f(w)\|\cdot\|y - w\| + \frac{c_1}{2}\|y - w\|^2 \le c_2 \implies \chi(1) \le c_1.$$

*Proof.* Let $w(t) = x + t(y - w)$. Then, by Taylor's theorem for a gradient, we have

$$\|\nabla f(w(t)) - \nabla f(w)\| = \left\| \int_0^t \nabla^2 f(w(\theta))(w(t) - w)\mathrm{d}\theta \right\|$$

$$\overset{(i)}{\le} \int_0^t \|\nabla^2 f(w(\theta))\|\mathrm{d}\theta \cdot \|w(t) - w\|$$

$$\overset{(ii)}{\le} \|w(t) - w\| \int_0^t (H_0 + H_1(f(w(\theta)) - f^*))\mathrm{d}\theta = \|w(t) - w\|\chi(t), \tag{40}$$

where $(i)$ follows from the definition of a spectral norm, $(ii)$ — from Definition 3.1, and $\chi(t) := \int_0^t (H_0 + H_1(f(w(\theta)) - f^*))\mathrm{d}\theta$. Eventually, we want to bound $\|\nabla f(y) - \nabla f(w)\|$ for any $y = w(1)$ and $w$. Therefore, our goal is to bound $\chi(1)$. Differentiating $\chi(t)$ we have

$$\chi'(t) = (H_0 + H_1(f(w(t)) - f^*))$$
$$= H_0 + H_1(f(w) - f^*) + H_1(f(w(t)) - f(w)). \tag{41}$$

Now we need to bound the difference $f(w(t)) - f(w)$ using Taylor's theorem for the function

$$f(w(t)) - f(w) = \int_0^t \langle \nabla f(w(\theta)), w(t) - x \rangle \mathrm{d}\theta \tag{42}$$

$$\overset{(iii)}{\le} \langle \nabla f(w), w(t) - w \rangle + \|w(t) - w\| \int_0^t \|\nabla f(w(\theta)) - \nabla f(w)\|\mathrm{d}\theta$$

$$\overset{(iv)}{\le} \langle \nabla f(w), w(t) - w \rangle + \|w(t) - w\| \int_0^t \chi(\theta)\|w(\theta) - w\|\mathrm{d}\theta$$

$$\overset{(v)}{\le} \langle \nabla f(w), w(t) - w \rangle + \|w(t) - w\|\chi(t) \int_0^t \theta\|y - w\|\mathrm{d}\theta$$

$$\overset{(vi)}{\le} \langle \nabla f(w), w(t) - w \rangle + \frac{1}{2}\|y - w\|^2\chi(t), \tag{43}$$

where $(iii)$ follows from Cauchy-Shawrz inequality, $(iv)$ — from (40), $(v)$ — from monotonicity of $\chi(t)$ by definition, $(vi)$ — from the fact that $t \in [0, 1]$. Using Cauchy-Schwarz inequality again, we obtain

$$f(w(t)) - f(w) \le \|\nabla f(w)\|\cdot\|w(t) - w\| + \frac{1}{2}\|y - w\|^2\chi(t) \le \|\nabla f(w)\|\cdot\|y - w\|$$
$$+ \frac{1}{2}\|y - w\|^2\chi(t), \tag{44}$$

Therefore, we obtain from (41) and (44) that

$$\chi'(t) \le H_0 + H_1(f(w) - f^*) + H_1\|\nabla f(w)\|\cdot\|y - w\| + \frac{1}{2}H_1\|y - w\|^2\chi(t).$$

Now we use Lemma 4 with $\alpha(t) = \chi(t), \beta(t) := H_0 + H_1(f(w) - f^*) + H_1\|\nabla f(w)\|\cdot\|y - w\| + \frac{t}{2}H_1\|y - w\|^2, a = 0, b = 1$. We obtain

$$\int_0^{\chi(1)} \frac{1}{\beta(u)}\mathrm{d}u \le 1.$$

Note that since $\beta(t)$ is monotonically increasing in $t$, then for any $c_1 > 0$ such that $\beta(c_1) \le c_1$ we have

$$\int_0^{\chi(1)} \frac{1}{\beta(u)}\mathrm{d}u \le 1 \le \frac{c_1}{\beta(c_1)} \le \int_0^{c_1} \frac{1}{\beta(u)}\mathrm{d}u.$$

This implies that $\chi(1) \leq c_1$ by monotonicity of the integral. Note that $\beta(c_1) \leq c_1$ can be rewritten as

$$\beta(c_1) \leq c_1$$

$$H_0 + H_1(f(w) - f^*) + H_1\|\nabla f(w)\|\cdot\|y - w\| + \frac{c_1}{2}H_1\|y - w\|^2 \leq H_0 + H_1(c_2 + f(w) - f^*)$$

$$\|\nabla f(w)\|\cdot\|y - w\| + \frac{c_1}{2}\|y - w\|^2 \leq c_2.$$

Therefore, $\chi(1) \leq c_1$ if $\beta(c_1) \leq c_1$ which is equivalent to

$$\|\nabla f(w)\|\cdot\|y - w\| + \frac{c_1}{2}\|y - w\|^2 \leq c_2.$$

$\square$

Our next lemma provides a gradient bound which can be seen as a generalization of the classic $\|\nabla f(w)\|^2 \leq 2L(f(w) - f^*)$ inequality, which holds in the $L$-smooth regime. The proof of Lemma 6 requires a careful choice of constants $c_1$ and $c_2$ from Lemma 5.

**Lemma 6.** *Let $f$ be $(H_0, H_1)$-smooth. Then we have*

$$\|\nabla f(w)\|^2 \leq \frac{9}{4}(H_0 + 3H_1(f(w) - f^*))(f(w) - f^*).$$

*Proof.* Let $c_2 := 2(f(w) - f^*)$, then $c_1 = H_0 + H_1(c_2 + f(w) - f^*) = H_0 + 3H_1(f(w) - f^*)$. From Lemma 5 we obtain that $\chi(1) \leq c_1$ if

$$\|\nabla f(w)\|\cdot\|y - w\| + \frac{c_1}{2}\|y - w\|^2 \leq 2(f(w) - f^*) \Longleftrightarrow$$

$$\|\nabla f(w)\|\cdot\|y - w\| + \frac{1}{2}(H_0 + 3H_1(f(w) - f^*))\|y - w\|^2 \leq 2(f(w) - f^*).$$

This is a quadratic polynomial of $\|y - w\|$ which has one negative and one positive solution. Therefore, to satisfy the constraint above, we should have

$$\|y - w\| \leq \frac{-\|\nabla f(w)\| + \sqrt{\|\nabla f(w)\|^2 + 4c_1(f(w) - f^*)}}{c_1} =: r.$$

In other words, we have $\chi(1) \leq c_1$ if points $y$ and $w$ satisfy the constraint above: $\|y - w\| \leq r$.

We choose $y := w - \eta\frac{\nabla f(w)}{\|\nabla f(w)\|}$, with $\eta \leq r$. Then, we have $\|y - w\| \leq r$, thus it holds $\chi(1) \leq c_1$.

From (43), the bound $\chi(1) \leq c_1$ implies

$$f(y) - f(w) \leq \langle \nabla f(w), y - w \rangle + \frac{c_1}{2}\|y - w\|^2.$$

Since $f(y) \geq f^*$ we can continue the inequality above as follows

$$f^* - f(w) \leq \langle \nabla f(w), y - w \rangle + \frac{c_1}{2}\|y - w\|^2.$$

Substituting $y = w - \eta\frac{\nabla f(w)}{\|\nabla f(w)\|}$, we get

$$-(f(w) - f^*) \leq -\eta\|\nabla f(w)\| + \frac{c_1}{2}\eta^2 \Longleftrightarrow \frac{c_1}{2}\eta^2 - \eta\|\nabla f(w)\| + (f(w) - f^*) \geq 0. \quad (45)$$

(45) is a quadratic polynomial of $\eta$, which attains the minimum at $\eta = \frac{\|\nabla f(w)\|}{c_1}$. We consider two cases:

1. $\frac{\|\nabla f(w)\|}{c_1} \leq r$ : Plugging this bound into (45) implies

$$\frac{c_1}{2}\frac{\|\nabla f(w)\|^2}{c_1^2} - \|\nabla f(w)\|\frac{\|\nabla f(w)\|}{c_1} + (f(w) - f^*) \geq 0 \Leftrightarrow \|\nabla f(w)\|^2 \leq 2c_1(f(w) - f^*).$$

2. $\frac{\|\nabla f(w)\|}{c_1} > r$ : In this case, the minimum of the polynomial in (45) for $\eta = [0, r]$ is attained at $\eta = r$. Thus, from (45) we obtain

$$\frac{c_1}{2}r^2 - r\|\nabla f(w)\| + (f(w) - f^*) \geq 0 \Leftrightarrow \|\nabla f(w)\| \leq \frac{c_1}{2}r + \frac{f(w) - f^*}{r}.$$

We plug in the definition of $r$ in the bound above and obtain

$$\|\nabla f(w)\| \leq \frac{-\|\nabla f(w)\| + \sqrt{\|\nabla f(w)\|^2 + 4c_1(f(w) - f^*)}}{2}$$

$$+ \frac{c_1(f(w) - f^*)}{-\|\nabla f(w)\| + \sqrt{\|\nabla f(w)\|^2 + 4c_1(f(w) - f^*)}},$$

$$2\|\nabla f(w)\|(-\|\nabla f(w)\| + \sqrt{\|\nabla f(w)\|^2 + 4c_1(f(w) - f^*)})$$

$$\leq (-\|\nabla f(w)\| + \sqrt{\|\nabla f(w)\|^2 + 4c_1(f(w) - f^*)})^2 2c_1(f(w) - f^*),$$

$$-2\|\nabla f(w)\|^2 + 2\|\nabla f(w)\|\sqrt{\|\nabla f(w)\|^2 + 4c_1(f(w) - f^*)} \leq \|\nabla f(w)\|^2 + \|\nabla f(w)\|^2$$

$$+ 4c_1(f(w) - f^*) - 2\|\nabla f(w)\|\sqrt{\|\nabla f(w)\|^2 + 4c_1(f(w) - f^*)} + 2c_1(f(w) - f^*)$$

$$4\|\nabla f(w)\|\sqrt{\|\nabla f(w)\|^2 + 4c_1(f(w) - f^*)} \leq 4\|\nabla f(w)\|^2 + 6c_1(f(w) - f^*)$$

$$4\|\nabla f(w)\|^2(\|\nabla f(w)\|^2 + 4c_1(f(w) - f^*)) \leq 4\|\nabla f(w)\|^4 + 9c_1^2(f(w) - f^*)^2$$

$$+ 12c_1\|\nabla f(w)\|^2(f(w) - f^*),$$

$$4c_1\|\nabla f(w)\|^2(f(w) - f^*) \leq 9c_1^2(f(w) - f^*)^2,$$

$$\|\nabla f(w)\|^2 \leq \frac{9c_1}{4}(f(w) - f^*).$$

We obtain that, in both cases, we have the following bound

$$\|\nabla f(w)\|^2 \leq \max\left\{\frac{9}{4}, 2\right\}(H_0 + 3H_1(f(w) - f^*))(f(w) - f^*)$$

$$\leq \frac{9}{4}(H_0 + 3H_1(f(w) - f^*))(f(w) - f^*).$$

$\square$

Finally, we provide a quadratic upper bound on the change of the function value. This inequality will be useful later to demonstrate that GD returns iterates with decreasing function value.

**Lemma 7.** *Let $f$ be $(H_0, H_1)$-smooth. Then for any $y, w \in \mathbb{R}^d$ such that $\|y - w\| \leq \frac{1}{6\sqrt{H_1}}$ we have*

$$f(y) \leq f(w) + \langle \nabla f(w), y - w \rangle + \frac{2H_0 + 2H_1(f(w) - f^*)}{2}\|y - w\|^2.$$

*Proof.* From Lemma 5 we know that $\chi(1) \leq c_1$ if

$$\|\nabla f(w)\| \cdot \|y - w\| + \frac{c_1}{2}\|y - w\|^2 \leq c_2. \tag{46}$$

If we choose $y$ and $w$ such that

$$\|y - w\| \leq \min\left\{\frac{c_2}{2\|\nabla f(w)\|}, \sqrt{\frac{c_2}{c_1}}\right\} =: r,$$

then (46) is satisfied. Indeed, this is due to

$$\|\nabla f(w)\| \cdot \|y - w\| \leq \|\nabla f(w)\| \frac{c_2}{2\|\nabla f(w)\|} = \frac{c_2}{2},$$

and

$$\frac{c_1}{2}\|y - w\|^2 \leq \frac{c_1}{2}\left(\sqrt{\frac{c_2}{c_1}}\right)^2 = \frac{c_2}{2}.$$

Therefore,

$$\|\nabla f(w)\|\cdot\|y - w\|+\frac{c_1}{2}\|y - w\|^2\le c_2.$$

Let $a_1 := \frac{9}{4}(H_0 + 3H_1(f(w) - f^*))(f(w) - f^*)$ and $a_2 = c_2(H_0 + H_1(c_2 + f(w) - f^*)) = c_1 c_2$. Then from Lemma 6 we have $\|\nabla f(w)\|^2 \le a_1$. Therefore, we obtain that

$$r \ge \frac{2c_2}{\sqrt{a_1} + 2\sqrt{c_1 c_2}} = \frac{2c_2}{\sqrt{a_1} + 2\sqrt{a_2}}.$$

Let

$$c_2 = \max\left\{f(w) - f^*, \frac{H_0}{3H_1}\right\}.$$

1. If $H_0 \le 3H_1(f(w) - f^*)$, then $\frac{H_0}{3H_1} \le f(w) - f^*$. Therefore, $c_2 = f(w) - f^*$. This implies that $a_1 \le \frac{27}{2}H_1(f(w) - f^*)^2$, and

$$a_2 = c_2(H_0 + H_1(c_2 + f(w) - f^*)) = (f(w) - f^*)(H_0 + 2H_1(f(w) - f^*))$$
$$\le 5H_1(f(w) - f^*)^2.$$

Moreover, in this case,

$$c_1 = H_0 + H_1(c_2 + f(w) - f^*) = H_0 + 2H_1(f(w) - f^*) \le 5H_1(f(w) - f^*).$$

Therefore, in this case, we obtain

$$r \ge \frac{2(f(w) - f^*)}{\frac{3}{2}\sqrt{6H_1}(f(w) - f^*) + 2\cdot\sqrt{5H_1}(f(w) - f^*}} = \frac{2}{\frac{3}{2}\sqrt{6H_1} + 2\sqrt{5H_1}} \ge \frac{0.245}{\sqrt{H_1}}$$
$$\ge \frac{1}{5\sqrt{H_1}}.$$

2. If $H_0 > 3H_1(f(w) - f^*)$, then $\frac{H_0}{3H_1} > f(w) - f^*$. Therefore, $c_2 = \frac{H_0}{3H_1}$. This implies that

$$a_1 = \frac{9}{4}(H_0 + 3H_1(f(w) - f^*))(f(w) - f^*) \le \frac{9}{2}H_0(f(w) - f^*) \le \frac{3H_0^2}{2H_1},$$

$$a_2 = c_2(H_0 + H_1(c_2 + f(w) - f^*)) = \frac{H_0}{3H_1}(H_0 + H_1(H_0/3H_1 + f(w) - f^*))$$

$$\le \frac{H_0}{3H_1}(H_0 + H_1 \cdot 2H_0/3H_1) = \frac{5H_0^2}{9H_1},$$

and

$$c_1 = H_0 + H_1(c_2 + f(w) - f^*) \le H_0 + H_1\left(\frac{H_0}{3H_1} + \frac{H_0}{3H_1}\right) = \frac{5}{3}H_0.$$

Therefore, in this case, we obtain

$$r \ge \frac{2\frac{H_0}{3H_1}}{\frac{H_0}{2}\sqrt{\frac{6}{H_1}} + \frac{2H_0}{3}\sqrt{\frac{5}{H_1}}} = \frac{2/3}{\sqrt{H_1}(\frac{\sqrt{6}}{2} + \frac{2\sqrt{5}}{3})} \ge \frac{0.193}{\sqrt{H_1}} \ge \frac{1}{6\sqrt{H_1}}.$$

Combining both cases, we obtain that if

$$\|y - w\|\le \min\left\{\frac{1}{5\sqrt{H_1}}, \frac{1}{6\sqrt{H_1}}\right\} = \frac{1}{6\sqrt{H_1}} \le r,$$

we obtain $\chi(1) \le c_1 \le \max\{H_0 + 2H_1(f(w) - f^*), 5H_0/3\} \le \frac{5H_0}{3} + 2H_1(f(w) - f^*) \le 2H_0 + 2H_1(f(w) - f^*)$. From (43), this implies

$$f(y) - f(w) \le \langle\nabla f(w), y - w\rangle + \frac{2H_0 + 2H_1(f(w) - f^*)}{2}\|y - w\|^2.$$

$\square$

## G MISSING PROOFS FOR SECTION 4

### G.1 CONVERGENCE FOR GENERAL NON-CONVEX FUNCTIONS

**Theorem G.1.** *Let $f$ be $(H_0, H_1)$-smooth. Then the iterates of GD $w_{k+1} = w_k - \eta_k \nabla f(w_k)$ where $\eta_k = \frac{1}{10H_0 + 20H_1(f(w_k) - f^*)}$ satisfy*

$$\min_{k < K} \|\nabla f(w_k)\|^2 \leq \frac{20(H_0 + 2H_1(f(w_0) - f^*))(f(w_0) - f^*)}{K} \frac{1}{1 + \frac{10H_1(f(w_0) - f^*)(K-1)(K-2)}{K^2(10H_0 + 20H_1(f(w_0) - f^*))}}.$$

*If $K \geq 6$, then the rate can be simplified*

$$\min_{k < K} \|\nabla f(w_k)\|^2 \leq \frac{20(H_0 + 2H_1(f(w_0) - f^*))(f(w_0) - f^*)}{K} \frac{1}{1 + \frac{H_1(f(w_0) - f^*)}{(2H_0 + 4H_1(f(w_0) - f^*))}}.$$

*Proof.* Note that $\|w_{k+1} - w_k\| = \eta_k \|\nabla f(w_k)\|$. Now we use Lemma 6 to obtain

$$\eta_k \|\nabla f(w_k)\| \leq \eta_k \frac{3}{2} \sqrt{(H_0 + 3H_1(f(w_k) - f^*))(f(w_k) - f^*)}.$$

1. If $H_0 \leq 3H_1(f(w_k) - f^*)$, then

$$\frac{3}{2} \eta_k \sqrt{6H_1(f(w_k) - f^*)^2} \leq \frac{3}{2} \eta_k \sqrt{6H_1}(f(w_k) - f^*). \tag{47}$$

We need to upper bound the above by $\frac{1}{6\sqrt{H_1}}$ to be able to use Lemma 7. We satisfy (47) by the choice of the step-size $\eta_k$

$$\eta_k \|\nabla f(w_k)\| \leq \frac{3}{2} \eta_k \sqrt{6H_1}(f(w_k) - f^*) \leq \frac{1}{6\sqrt{H_1}} \Leftrightarrow \eta_k \leq \frac{1}{\frac{15}{2}\sqrt{6}H_1(f(w_k) - f^*)},$$

where the last inequality is satisfied since

$$\eta_k = \frac{1}{10H_0 + 20H_1(f(w_k) - f^*)} \leq \frac{1}{20H_1(f(w_k) - f^*)} \leq \frac{1}{\frac{15}{2}\sqrt{6}H_1(f(w_k) - f^*)}.$$

2. If $H_0 > 3H_1(f(w_k) - f^*)$, then

$$\eta_k \|\nabla f(w_k)\| \leq \frac{3}{2} \eta_k \sqrt{2H_0(f(w_k) - f^*)} \leq \frac{3}{2} \eta_k \sqrt{2H_0 \cdot \frac{H_0}{3H_1}} = \eta_k \frac{\sqrt{3}H_0}{\sqrt{2H_1}}. \tag{48}$$

We need to upper bound the above by $\frac{1}{6\sqrt{H_1}}$ to be able to use Lemma 7. We satisfy (48) by the choice of the step-size $\eta_k$

$$\eta_k \|\nabla f(w_k)\| \leq \eta_k \frac{\sqrt{3}H_0}{\sqrt{2H_1}} \leq \frac{1}{6\sqrt{H_1}} \Leftrightarrow \eta_k \leq \frac{\sqrt{6}}{18H_0},$$

where the last inequality is satisfied since

$$\eta_k = \frac{1}{10H_0 + 20H_1(f(w_k) - f^*)} \leq \frac{1}{10H_0} \leq \frac{\sqrt{6}}{18H_0}.$$

Therefore, the choice of the step-size allows to use Lemma 7 since the restriction $\|w_{k+1} - w_k\| \leq \frac{1}{6\sqrt{H_1}}$ is satisfied. Therefore, we have

$$f(w_{k+1}) \overset{(i)}{\leq} f(w_k) + \langle \nabla f(w_k), w_{k+1} - w_k \rangle + \frac{2H_0 + 2H_1(f(w_k) - f^*)}{2} \|w_{k+1} - w_k\|^2$$

$$\overset{(ii)}{\leq} f(w_k) - \eta_k \|\nabla f(w_k)\|^2 + (H_0 + H_1(f(w_k) - f^*))\eta_k^2 \|\nabla f(w_k)\|^2$$

$$= f(w_k) - \eta_k \|\nabla f(w_k)\|^2 (1 - \eta_k(H_0 + H_1(f(w_k) - f^*)))$$

$$\overset{(iii)}{\leq} f(w_k) - \frac{\eta_k}{2} \|\nabla f(w_k)\|^2, \tag{49}$$

where $(i)$ follows from Lemma 7, $(ii)$ — from Lemma 6, $(iii)$ — from the choice of the step-size $\eta_k \leq \frac{1}{10H_0+20H_1(f(w_k)-f^*)}$. This implies that GD achieves a monotone decrease of the function value. By the choice of the step-size $\eta_k = \frac{1}{10H_0+20H_1(f(w_k)-f^*)}$, we obtain that $\eta_k$ is increasing with $k$. Rearranging the last inequality we obtain $\|\nabla f(w_k)\|^2 \leq \frac{2}{\eta_k}(f(w_k) - f(w_{k+1}))$. Summing this inequality over iterations $\{0, \ldots, K-1\}$ we obtain

$$
\begin{aligned}
\frac{1}{K}\sum_{k=0}^{K-1}\|\nabla f(w_k)\|^2 &\leq \frac{1}{K}\sum_{k=0}^{K-1}\frac{2}{\eta_k}(f(w_k) - f(w_{k+1})) \\
&= \frac{1}{K}\sum_{k=0}^{K-1}(20H_0 + 40H_1(f(w_k) - f^*))(f(w_k) - f(w_{k+1})) \\
&= \frac{20H_0}{K}\sum_{k=0}^{K-1}f(w_k) - f(w_{k+1}) \\
&\qquad + \frac{40H_1}{K}\sum_{k=0}^{K-1}(f(w_k) - f^*)^2 - (f(w_k) - f^*)(f(w_{k+1}) - f^*) \\
&\stackrel{(iv)}{\leq} \frac{20H_0}{K}\sum_{k=0}^{K-1}f(w_k) - f(w_{k+1}) \\
&\qquad + \frac{40H_1}{K}\sum_{k=0}^{K-1}(f(w_k) - f^*)^2 - (f(w_{k+1}) - f^*)^2 \\
&\leq \frac{20H_0(f(w_0) - f^*)}{K} + \frac{40H_1(f(w_0) - f^*)^2}{K}.
\end{aligned}
$$

The current rate is the same as with a constant step-size $\eta = \frac{1}{10H_0+20H_1(f(w_0)-f^*)}$, i.e., we do not show improvement. Now our goal is to obtain a tighter rate for GD using the fact that the sequence $\{\eta_k\}$ is increasing. By (49), we obtain

$$
f(w_k) \leq f(w_0) - \sum_{j=0}^{k-1}\frac{\eta_j}{2}\|\nabla f(w_j)\|^2 \Rightarrow f(w_k) - f^* \leq (f(w_0) - f^*) - \sum_{j=0}^{k-1}\frac{\eta_j}{2}\|\nabla f(w_j)\|^2.
$$

Therefore,

$$
\frac{1}{\sum_{k=0}^{K-1}\eta_k}\sum_{k=0}^{K-1}\eta_k\|\nabla f(w_k)\|^2 \leq \frac{2(f(w_0) - f^*)}{\sum_{k=0}^{K-1}\eta_k}.
$$

To provide a tighter bound, we should take into account that the step-sizes are increasing since $f(w_k) - f^*$ is decreasing. Remember that $\eta_k = \frac{1}{10H_0+20H_1(f(w_k)-f^*)}$, then

$$
\begin{aligned}
\sum_{k=0}^{K-1}\eta_k &= \sum_{k=0}^{K-1}\frac{1}{10H_0 + 20H_1(f(w_k) - f^*)} \\
&\geq \sum_{k=0}^{K-1}\frac{1}{10H_0 + 20H_1\left(f(w_0) - f^* - \sum_{j=0}^{k-1}\frac{\eta_j}{2}\|\nabla f(w_j)\|^2\right)}.
\end{aligned}
$$

Let us denote $\Lambda_k = \sum_{j=0}^{k-1}\eta_j\|\nabla f(w_j)\|^2$, then

$$
\sum_{k=0}^{K-1}\eta_k \geq \sum_{k=0}^{K-1}\frac{1}{10H_0 + 20H_1(f(w_0) - f^*) - 10H_1\Lambda_k}.
$$

Since the function $u \to g(u) := \frac{1}{10H_0+20H_1(f(w_0)-f^*)-10H_1u}$ is convex in the set $\{u \in \mathbb{R} \mid g(u) > 0\}$, then by Jensen's inequality we have

$$
\frac{1}{K}\sum_{k=0}^{K-1}g(\Lambda_k) \geq g\left(\frac{1}{K}\sum_{k=0}^{K-1}\Lambda_k\right).
$$

In our case, we obtain

$$\sum_{k=0}^{K-1} \eta_k \geq \sum_{k=0}^{K-1} g(\Lambda_k) \geq \frac{K}{10H_0 + 20H_1(f(w_0) - f^*) - \frac{10H_1}{K}\sum_{k=0}^{K-1}\Lambda_k}.$$

Now we estimate

$$\sum_{k=0}^{K-1}\Lambda_k = \sum_{k=0}^{K-1}\sum_{j=0}^{k-1}\eta_j\|\nabla f(w_j)\|^2 \geq \min_{k<K}\|\nabla f(w_k)\|^2 \sum_{k=0}^{K-1}\sum_{j=0}^{k-1}\eta_j \geq \min_{k<K}\|\nabla f(w_k)\|^2 \eta_0 \frac{(K-1)K}{2},$$

where we use the fact that $\eta_0 \leq \eta_k$ for all $k \geq 0$. This leads to the following bound

$$\min_{k<K}\|\nabla f(w_k)\|^2 \leq \frac{1}{\sum_{k=0}^{K-1}\eta_k}\sum_{k=0}^{K-1}\eta_k\|\nabla f(w_k)\|^2$$

$$\leq \frac{2(f(w_0) - f^*)}{\frac{K}{10H_0 + 20H_1(f(w_0)-f^*) - \frac{10H_1}{K}\eta_0\frac{(K-1)(K-2)}{2}\min_k\|\nabla f(w_k)\|^2}}$$

$$\leq \frac{2(10H_0 + 20H_1(f(w_0)-f^*))(f(w_0)-f^*)}{K}$$

$$- \frac{10H_1(f(w_0)-f^*)(K-1)(K-2)\eta_0\min_k\|\nabla f(w_k)\|^2}{K^2}.$$

Rearranging the terms, we obtain

$$\min_{k<K}\|\nabla f(w_k)\|^2 \leq \frac{20(H_0 + 2H_1(f(w_0)-f^*))(f(w_0)-f^*)}{K}\frac{1}{1 + \frac{10H_1(f(w_0)-f^*)(K-1)(K-2)}{K^2(10H_0 + 20H_1(f(w_0)-f^*))}}.$$

If $K \geq 6$, then $\frac{10(K-1)(K-2)}{K^2} \geq 5$, which leads to the simplified rate. $\qquad\square$

## G.2 Convergence under Aiming Condition

**Theorem 4.2.** *Assume that $f$ is $(H_0, H_1)$-smooth, and it satisfies the Aiming condition with constant $\theta$ around the set of global minimizers $\mathcal{S}$. Then the iterates of* `GD` *with adaptive step-size $\theta \cdot \eta_k$ satisfy*

$$f(w_K) - f^* \leq \varepsilon \quad \text{after at most} \quad \frac{40H_0\mathrm{dist}(w_0,\mathcal{S})^2}{\theta^2\varepsilon} + \frac{40H_1\mathrm{dist}(w_0,\mathcal{S})^2}{\theta^2} \quad \text{iterations.}$$

*Proof.* We start by (49)

$$f(w_{k+1}) \leq f(w_k) - \frac{\eta_k}{2}\|\nabla f(w_k)\|^2 = f(w_k) - \frac{\theta}{20H_0 + 40H_1(f(w_k)-f^*)}\|\nabla f(w_k)\|^2. \quad (50)$$

Next, we show that the distance to the set of global minimizers $\mathcal{S}$ of the function $f$ does not increase. Indeed, we have

$$\mathrm{dist}(w_{k+1},\mathcal{S})^2 \overset{(i)}{=} \|w_{k+1} - \pi_{\mathcal{S}}(w_k)\|^2$$

$$= \|w_k - \pi_{\mathcal{S}}(w_k)\|^2 - 2\eta_k\langle w_k - \pi_{\mathcal{S}}(w_k), \nabla f(w_k)\rangle + \eta_k^2\|\nabla f(w_k)\|^2$$

$$\overset{(ii)}{\leq} \mathrm{dist}(w_k,\mathcal{S})^2 - 2\eta_k\theta(f(w_k)-f^*) + \eta_k^2\|\nabla f(w_k)\|^2$$

$$\overset{(iii)}{\leq} \mathrm{dist}(w_k,\mathcal{S})^2 - 2\eta_k\theta(f(w_k)-f^*)$$

$$+ \frac{9\eta_k^2}{4}(H_0 + 3H_1(f(w_k)-f^*))(f(w_k)-f^*)$$

$$= \mathrm{dist}(w_k,\mathcal{S})^2 - 2\eta_k(f(w_k)-f^*)\left(\theta - \frac{9}{8}\eta_k(H_0 + 3H_1(f(w_k)-f^*))\right),$$

where $(i)$ follows from the definition of the projection, $(ii)$ follows from the definition of the Aiming condition, $(iii)$ — from Lemma 6. Now we use the choice of the step-size $\eta_k = \frac{\theta}{10H_0 + 20H_1(f(w_k) - f^*)}$ to obtain

$$\text{dist}(w_{k+1}, \mathcal{S})^2 \le \text{dist}(w_k, \mathcal{S})^2 - \eta_k \theta(f(w_k) - f^*). \tag{51}$$

Therefore, we have that $\text{dist}(w_{k+1}, \mathcal{S})^2 \le \text{dist}(w_k, \mathcal{S})^2$ for any $k \ge 0$. Now we consider two cases:

- $f(w_k) - f^* \ge \frac{H_0}{2H_1}$ (large function value). In this case, we can lower bound the step-size as

$$\eta_k = \frac{\theta}{10H_0 + 20H_1(f(w_k) - f^*)} \ge \frac{\theta}{40H_1(f(w_k) - f^*)}.$$

Therefore, from (51), we obtain

$$\begin{aligned}
\text{dist}(w_{k+1}, \mathcal{S})^2 &\le \text{dist}(w_k, \mathcal{S})^2 - \eta_k \theta(f(w_k) - f^*) \\
&\le \text{dist}(w_k, \mathcal{S})^2 - \frac{\theta}{40H_1(f(w_k) - f^*)}\theta(f(w_k) - f^*) \\
&= \text{dist}(w_k, \mathcal{S})^2 - \frac{\theta^2}{40H_1}.
\end{aligned}$$

Since $\text{dist}(w_k, \mathcal{S})^2 \ge 0$, we can stay in this regime at most $T$ iterations, such that

$$0 \le \text{dist}(w_T, \mathcal{S})^2 \le \text{dist}(w_0, \mathcal{S})^2 - \frac{\theta^2}{40H_1}T \Rightarrow T := \frac{40H_1\text{dist}(w_0, \mathcal{S})^2}{\theta^2}.$$

- $f(w_k) - f^* \le \frac{H_0}{2H_1}$ (small function value). In this case, we can lower bound the step-size as

$$\eta_k = \frac{\theta}{10H_0 + 20H_1(f(w_k) - f^*)} \ge \frac{\theta}{20H_0}.$$

Therefore, from (51), we obtain

$$\begin{aligned}
\text{dist}(w_{k+1}, \mathcal{S})^2 &\le \text{dist}(w_k, \mathcal{S})^2 - \eta_k \theta(f(w_k) - f^*) \\
&\le \text{dist}(w_k, \mathcal{S})^2 - \frac{\theta^2}{40H_0}(f(w_k) - f^*).
\end{aligned}$$

Rearranging the terms, we obtain

$$f(w_k) - f^* \le \frac{40H_0}{\theta^2}(\text{dist}(w_k, \mathcal{S})^2 - \text{dist}(w_{k+1}, \mathcal{S})^2). \tag{52}$$

Averaging the inequalities (52) for $k \in \{T, \ldots, K\}$, we obtain

$$\begin{aligned}
\frac{1}{K - T + 1}\sum_{k=T}^{K}(f(w_k) - f^*) &\le \frac{40H_0(\text{dist}(w_0, \mathcal{S})^2 - \text{dist}(w_{K+1}, \mathcal{S})^2)}{\theta^2(K - T + 1)} \\
&\le \frac{40H_0\text{dist}(w_0, \mathcal{S})^2}{\theta^2(K - T + 1)}.
\end{aligned}$$

Since $f(w_k) - f^*$ is decreasing by (50), we have

$$f(w_K) - f^* \le \frac{40H_0\text{dist}(w_0, \mathcal{S})^2}{\theta^2(K - T + 1)}.$$

To achieve $\varepsilon$ accuracy, we need the number of iterations $K$ to be

$$\begin{aligned}
f(w_K) - f^* \le \frac{40H_0\text{dist}(w_0, \mathcal{S})^2}{\theta^2(K - T + 1)} \le \varepsilon &\Rightarrow K \ge \frac{40H_0\text{dist}(w_0, \mathcal{S})^2}{\theta^2\varepsilon} + T \\
&= \frac{40H_0\text{dist}(w_0, \mathcal{S})^2}{\theta^2\varepsilon} + \frac{40H_1\text{dist}(w_0, \mathcal{S})^2}{\theta^2}.
\end{aligned}$$

$\square$

The next theorem demonstrates that when the function sub-optimality is large, we should expect a linear decrease. This gives another intuition behind the improvement from the warm-up schedule. This result demonstrates that linear convergence can be expected even beyond the PL case.

**Theorem G.2.** *Assume that $f$ is $(H_0, H_1)$-smooth, and it satisfies the Aiming condition with constant $\theta$ around the set of global minimizers $\mathcal{S}$. Assume that $f(w_k) - f^* \geq \frac{H_0}{2H_1}$. Then the iterates of GD $w_{k+1} = w_k - \eta_k \nabla f(w_k)$ with a step-size $\eta_k = \frac{\theta}{10H_0 + 20H_1(f(w_k) - f^*)}$ satisfy*

$$f(w_{k+1}) - f^* \leq \left(1 - \frac{\theta^3}{80H_1 \text{dist}(w_0, \mathcal{S})^2}\right)(f(w_k) - f^*).$$

*Proof.* First, we use the previously derived decrease in the function value (50)

$$f(w_{k+1}) - f^* \leq f(w_k) - f^* - \frac{\theta}{20H_0 + 40H_1(f(w_k) - f^*)}\|\nabla f(w_k)\|^2,$$

and in the distance (51)

$$\text{dist}(w_{k+1}, \mathcal{S})^2 \leq \text{dist}(w_k, \mathcal{S})^2 - \eta_k \theta(f(w_k) - f^*).$$

In particular, $\text{dist}(w_k, \mathcal{S})^2 \leq \text{dist}(w_0, \mathcal{S})^2$. From the Aiming condition, we have

$$\theta(f(w_k) - f^*) \leq \langle \nabla f(w_k), w_k - \pi_{\mathcal{S}}(w_k)\rangle \leq \|\nabla f(w_k)\| \cdot \text{dist}(w_k, \mathcal{S})$$
$$\leq \|\nabla f(w_k)\| \cdot \text{dist}(w_0, \mathcal{S}). \tag{53}$$

Therefore, we obtain

$$f(w_{k+1}) - f^* \leq f(w_k) - f^* - \frac{\theta}{20H_0 + 40H_1(f(w_k) - f^*)}\|\nabla f(w_k)\|^2$$
$$\overset{(i)}{\leq} f(w_k) - f^* - \frac{\theta}{80H_1(f(w_k) - f^*)}\|\nabla f(w_k)\|^2$$
$$\overset{(ii)}{\leq} f(w_k) - f^* - \frac{\theta}{80H_1(f(w_k) - f^*)}\frac{\theta^2(f(w_k) - f^*)^2}{\text{dist}(w_0, \mathcal{S})^2}$$
$$= \left(1 - \frac{\theta^3}{80H_1 \text{dist}(w_0, \mathcal{S})^2}\right)(f(w_k) - f^*).$$

where $(i)$ follows from the bound $f(w_k) - f^* \geq \frac{H_0}{2H_1}$, $(ii)$ – from (53). $\qquad\square$

### G.3 CONVERGENCE UNDER POLYAK-ŁOJASIEWICZ CONDITION

**Theorem 4.3.** *Assume that $f$ is $(H_0, H_1)$-smooth, and it satisfies $\mu$-PL condition. Then the iterates of GD with adaptive step-size $\eta_k$ satisfy*

$$f(w_K) - f^* \leq \varepsilon \quad \text{after at most} \quad \frac{40H_1}{\mu}(f(w_0) - f^*) + \frac{20H_0}{\mu}\log\frac{H_0}{2H_1\varepsilon} \quad \text{iterations.}$$

*Proof.* We start with the equation (49) and use $\mu$-PL inequality

$$f(w_{k+1}) \leq f(w_k) - \frac{\eta_k}{2}\|\nabla f(w_k)\|^2$$
$$\leq f(w_k) - \mu\eta_k(f(w_k) - f^*)$$
$$= f(w_k) - \frac{\mu(f(w_k) - f^*)}{10H_0 + 20H_1(f(w_k) - f^*)}.$$

Now we consider two cases.

- $f(w_k) - f^* \geq \frac{H_0}{2H_1}$ (large function value). In this case, we have

$$f(w_{k+1}) \leq f(w_k) - \frac{\mu(f(w_k) - f^*)}{10H_0 + 20H_1(f(w_k) - f^*)}$$
$$\leq f(w_k) - \frac{\mu(f(w_k) - f^*)}{40H_1(f(w_k) - f^*)}$$
$$= f(w_k) - \frac{\mu}{40H_1}.$$

Since GD decreases the function value (see (49)), we have $f(w_t) - f^* \geq \frac{H_0}{2H_1}$ for all $K \in \{0, \ldots, k\}$. Therefore,

$$f(w_{k+1}) - f^* \leq f(w_0) - f^* - \frac{\mu}{40H_1}(k+1).$$

However, we cannot reduce the function value infinitely many times, since it is lower bounded. We can stay in this regime as long as $f(w_t) - f^* \geq \frac{H_0}{2H_1}$, therefore, GD stays in this regime for at most $k \leq \frac{40H_1}{\mu}\left(f(w_0) - f^* - \frac{H_0}{2H_1}\right) - 1 \leq \frac{40H_1}{\mu}(f(w_0) - f^*) - \frac{20H_0}{\mu}$ iterations. In other words, the cardinality of the set $\mathcal{T} := \{k \in \{0, \ldots, K-1\} : f(w_k) - f^* \geq \frac{H_0}{2H_1}\}$ is bounded by $T = \frac{40H_1}{\mu}(f(w_0) - f^*) - \frac{20H_0}{\mu}$.

- $f(w_k) - f^* \leq \frac{H_0}{2H_1}$ (small function value). In this case, we have

$$f(w_{k+1}) \leq f(w_k) - \frac{\mu(f(w_k) - f^*)}{10H_0 + 20H_1(f(w_k) - f^*)}$$
$$\leq f(w_k) - \frac{\mu(f(w_k) - f^*)}{20H_0}. \tag{54}$$

Since the function along the trajectory of GD does not increase (see (49)), we stay in this regime for the rest of the training. Therefore, summing up (54) for all iterations $k \in \{T, \ldots, K-1\}$ we obtain

$$f(w_K) - f^* \leq \left(1 - \frac{\mu}{20H_0}\right)(f(w_{K-1}) - f^*)$$
$$\leq \ldots$$
$$\leq \left(1 - \frac{\mu}{20H_0}\right)^{K-T}(f(w_T) - f^*).$$

Since $f(w_T) - f^* \leq f(w_0) - f^* - \frac{\mu T}{40H_1}$, we get the rate

$$f(w_K) - f^*$$
$$\leq \left(1 - \frac{\mu}{20H_0}\right)^{K-T}\left(f(w_0) - f^* - \frac{\mu}{40H_1}\left(\frac{40H_1}{\mu}(f(w_0) - f^*) - \frac{20H_0}{\mu}\right)\right)$$
$$= \left(1 - \frac{\mu}{20H_0}\right)^{K-T}\frac{H_0}{2H_1}.$$

To achieve $f(w_K) - f^* \leq \varepsilon$ we need to satisfy

$$f(w_K) - f^* \leq \left(1 - \frac{\mu}{20H_0}\right)^{K-T}\frac{H_0}{2H_1} \leq \varepsilon \Rightarrow K \geq T + \frac{20H_0}{\mu}\log\frac{H_0}{2H_1\varepsilon}$$
$$= \frac{40H_1}{\mu}(f(w_0) - f^*) + \frac{20H_0}{\mu}\log\frac{H_0}{2H_1\varepsilon}.$$

$\square$

### G.4 Convergence in the Stochastic Setting

**Theorem 4.4.** *Assume that the problem (∗) satisfies the interpolation condition. Assume that each $f_i$ is $(H_0, H_1)$-smooth and satisfies the Aiming condition around the set of global minimizers $\mathcal{S}$. Then the iterates of SGD $w_{k+1} = w_k - \eta_k\nabla f_{S_k}(w_k)$ with a step-size $\eta_k = \frac{\theta}{10H_0 + 20H_1(f_{S_k}(w_k) - f^*_{S_k})}$ and batch $S_k \subseteq [n]$ satisfy*

$$\frac{1}{K+1}\sum_{k=0}^{K}\mathbb{E}\left[\min\left\{f(w_k) - f^*, \frac{H_0}{2nH_1}\right\}\right] \leq \frac{20H_0\mathrm{dist}(w_0, \mathcal{S})^2}{\theta^2(K+1)}.$$

*Proof.* We show that the distance to the set of global minimizers $\mathcal{S}$ of the function $f$ does not increase. Indeed, we have

$$
\begin{aligned}
\text{dist}(w_{k+1}, \mathcal{S})^2 &= \|w_{k+1} - \pi_{\mathcal{S}}(w_{k+1})\|^2 \\
&\leq \|w_{k+1} - \pi_{\mathcal{S}}(w_k)\|^2 \\
&= \|w_k - \pi_{\mathcal{S}}(w_k)\|^2 - 2\eta_k \langle w_k - \pi_{\mathcal{S}}(w_k), \nabla f_{S_k}(w_k)\rangle + \eta_k^2 \|\nabla f_{S_k}(w_k)\|^2 \\
&\overset{(i)}{\leq} \|w_k - \pi_{\mathcal{S}}(w_k)\|^2 - 2\theta\eta_k(f_{S_k}(w_k) - f_{S_k}^*) + \eta_k^2 \|\nabla f_{S_k}(w_k)\|^2 \\
&\overset{(ii)}{\leq} \text{dist}(w_k, \mathcal{S})^2 - 2\theta\eta_k(f_{S_k}(w_k) - f_{S_k}^*) \\
&\qquad\qquad + \frac{9}{4}\eta_k^2(H_0 + 3H_1(f_{S_k}(w_k) - f_{S_k}^*))(f_{S_k}(w_k) - f_{S_k}^*) \\
&\overset{(iii)}{=} \text{dist}(w_k, \mathcal{S})^2 - 2\eta_k(f_{S_k}(w_k) - f_{S_k}(w^*)) \\
&\qquad\qquad + \frac{9}{4}\eta_k^2(H_0 + 3H_1(f_{S_k}(w_k) - f_{S_k}(w^*)))(f_{S_k}(w_k) - f_{S_k}(w^*)) \\
&= \text{dist}(w_k, \mathcal{S})^2 - 2\eta_k(f_{S_k}(w_k) - f_{S_k}(w^*))\left(\theta - \frac{9}{8}\eta_k(H_0 + 3H_1(f_{S_k}(w_k) - f_{S_k}(w^*)))\right)
\end{aligned}
$$

where $(i)$ follows from Definition 4.1, $(ii)$ — from Lemma 6, $(iii)$ — from the interpolation condition. Now we use the choice of the step-size

$$
\eta_k = \frac{\theta}{10H_0 + 20H_1(f_{S_k}(w_k) - f_{S_k}^*)} = \frac{\theta}{10H_0 + 20H_1(f_{S_k}(w_k) - f_{S_k}(w^*))}
$$

to obtain

$$
\text{dist}(w_{k+1}, \mathcal{S})^2 \leq \text{dist}(w_k, \mathcal{S})^2 - \eta_k\theta(f_{S_k}(w_k) - f_{S_k}(w^*)). \tag{55}
$$

Therefore, we have that $\text{dist}(w_{k+1}, \mathcal{S})^2 \leq \text{dist}(w_k, \mathcal{S})^2$ for any $k \geq 0$. Now we consider two cases:

- $f_{S_k}(w_k) - f_{S_k}(w^*) \geq \frac{H_0}{2H_1}$ (large function value). In this case, we can lower bound the step-size $\eta_k$ as

$$
\eta_k = \frac{\theta}{10H_0 + 20H_1(f_{S_k}(w_k) - f_{S_k}(w^*))} \geq \frac{\theta}{40H_1(f_{S_k}(w_k) - f_{S_k}(w^*))}.
$$

  Therefore, from (55), we obtain

$$
\begin{aligned}
\text{dist}(w_{k+1}, \mathcal{S})^2 &\leq \text{dist}(w_k, \mathcal{S})^2 - \eta_k\theta(f_{S_k}(w_k) - f_{S_k}(w^*)) \\
&\leq \text{dist}(w_k, \mathcal{S})^2 - \frac{\theta^2}{40H_1(f_{S_k}(w_k) - f_{S_k}(w^*))}(f_{S_k}(w_k) - f_{S_k}(w^*)) \\
&= \text{dist}(w_k, \mathcal{S})^2 - \frac{\theta^2}{40H_1}. \tag{56}
\end{aligned}
$$

- $f_{S_k}(w_k) - f_{S_k}(w^*) \leq \frac{H_0}{2H_1}$ (small function value). In this case, we can lower bound the step-size $\eta_k$ as

$$
\eta_k = \frac{\theta}{10H_0 + 20H_1(f_{S_k}(w_k) - f_{S_k}(w^*))} \geq \frac{\theta}{20H_0}.
$$

  Therefore, from (55), we obtain

$$
\begin{aligned}
\text{dist}(w_{k+1}, \mathcal{S})^2 &\leq \text{dist}(w_k, \mathcal{S})^2 - \eta_k\theta(f_{S_k}(w_k) - f_{S_k}(w^*)) \\
&\leq \text{dist}(w_k, \mathcal{S})^2 - \frac{\theta^2}{20H_0}(f_{S_k}(w_k) - f_{S_k}(w^*)). \tag{57}
\end{aligned}
$$

To combine descent inequalities (56) and (57), we introduce the even $E(w_k) :=$ $\left\{ f_{S_k}(w_k) - f_{S_k}(w^*) \geq \frac{H_0}{2H_1} \mid w_k \right\}$ for given $w_k$ and its indicator function $\mathbb{1}_{E(w_k)}$, i.e., for given $w_k$, $\mathbb{1}_{E(w_k)} = 1$ if $f_{S_k}(w_k) - f_{S_k}(w^*) \geq \frac{H_0}{2H_1}$, and $\mathbb{1}_{E(w_k)} = 0$ if $f_{S_k}(w_k) - f_{S_k}(w^*) < \frac{H_0}{2H_1}$. Then the descent in the general case can be written as

$$\text{dist}(w_{k+1}, \mathcal{S})^2 \leq \text{dist}(w_k, \mathcal{S})^2 - \mathbb{1}_{E(w_k)} \frac{\theta^2}{40H_1} - (1 - \mathbb{1}_{E(w_k)}) \frac{\theta^2}{20H_0} (f_{S_k}(w_k) - f_{S_k}(w^*)). \tag{58}$$

We denote $\mathbb{E}_k[\cdot]$ as $\mathbb{E}[\cdot \mid w_k]$ – the expectation conditioned on $w_k$. Thus, we have from (58) that

$$\mathbb{E}_k \left[ \text{dist}(w_{k+1}, \mathcal{S})^2 \right] \leq \text{dist}(w_k, \mathcal{S})^2 - \frac{\theta^2}{20H_0} \mathbb{E}_k \left[ (1 - \mathbb{1}_{E(w_k)})(f_{S_k}(w_k) - f_{S_k}(w^*)) \right]$$

$$- \mathbb{E}_k \left[ \mathbb{1}_{E(w_k)} \right] \frac{\theta^2}{40H_1}$$

$$= \text{dist}(w_k, \mathcal{S})^2 - \frac{\theta^2}{20H_0} \mathbb{E}_k \left[ (1 - \mathbb{1}_{E(w_k)})(f_{S_k}(w_k) - f_{S_k}(w^*)) \right]$$

$$- p_k \frac{\theta^2}{40H_1}, \quad (59)$$

where $p_k := \mathbb{E}_k \left[ \mathbb{1}_{E(w_k)} \right] = \mathbb{P}(E(w_k)) = \mathbb{P}(f_{S_k}(w_k) - f_{S_k}(w^*) \geq \frac{H_0}{2H_1})$. We emphasize that $p_k$ is a random variable. If $p_k > 0$, then there is at least one $i \in [n]$, so that $f_i(w_k) - f_i(w^*) \geq \frac{H_0}{2H_1}$ for given $w_k$. Thus, we have $p_k \geq \frac{1}{n}$. In the opposite case, we have $p_k = 0$, and $1 - \mathbb{1}_{E(w_k)} = 1$ for given $w_k$. Putting all together, we continue as follows

$$\mathbb{E}_k \left[ \text{dist}(w_{k+1}, \mathcal{S})^2 \right] \leq \text{dist}(w_k, \mathcal{S})^2 - \frac{\theta^2}{20H_0} \mathbb{1}_{\{p_k = 0\}}(f(w_k) - f(w^*)) - \mathbb{1}_{\{p_k > 0\}} p_k \frac{\theta^2}{40H_1}$$

$$\leq \text{dist}(w_k, \mathcal{S})^2 - \frac{\theta^2}{20H_0} \mathbb{1}_{\{p_k = 0\}}(f(w_k) - f(w^*)) - \mathbb{1}_{\{p_k > 0\}} \frac{\theta^2}{40nH_1}$$

$$\leq \text{dist}(w_k, \mathcal{S})^2 - \min \left\{ \frac{\theta^2}{20H_0}(f(w_k) - f(w^*)), \frac{\theta^2}{40nH_1} \right\}.$$

Taking full expectation and rearranging terms, we obtain

$$\sum_{k=0}^{K} \mathbb{E} \left[ \min \left\{ \frac{\theta^2}{20H_0}(f(w_k) - f(w^*)), \frac{\theta^2}{40nH_1} \right\} \right] \leq \sum_{k=0}^{K+1} \mathbb{E} \left[ \text{dist}(w_k, \mathcal{S})^2 \right] - \mathbb{E} \left[ \text{dist}(w_{k+1}, \mathcal{S})^2 \right]$$

$$\leq \text{dist}(w_0, \mathcal{S})^2.$$

Dividing both sides by $\frac{\theta^2}{20H_0(K+1)}$, we obtain

$$\frac{1}{K+1} \sum_{k=0}^{K} \mathbb{E} \left[ \min \left\{ f(w_k) - f(w^*), \frac{H_0}{2nH_1} \right\} \right] \leq \frac{20H_0 \text{dist}(w_0, \mathcal{S})^2}{\theta^2(K+1)}.$$

The rate above implies

$$\min_{k < K+1} \mathbb{E} \left[ \min \left\{ f(w_k) - f(w^*), \frac{H_0}{2nH_1} \right\} \right] \leq \frac{20H_0 \text{dist}(w_0, \mathcal{S})^2}{\theta^2(K+1)}.$$

$\square$

# H    Missing Proofs for GD in the Convex Setting

In this case, we demonstrate the convergence to the minimizer $w^*$ of the convex function $f$.

*Proof.* The proof mainly follows the proof of Theorem 4.2 by setting $\theta = 1$ and $\mathcal{S} = \{w^*\}$.

$\square$

## I LOWER BOUNDS

**Theorem 4.1.** *Let $f$ belong to the class $\mathcal{H}$ of $(H_0, H_1)$-smooth functions. Then it holds:*

1. *To satisfy $\|\nabla f(w_K)\| \leq \varepsilon$ for a general non-convex function $f$, GD with constant step-size initialized at $w_0$, needs at least*

$$K \geq \frac{H_1(f(w_0)-f^*)}{\log(f(w_0)-f^*)+1} \frac{f(w_0)-f^*-2\epsilon^2}{8\epsilon^2} \quad iterations.$$

2. *To satisfy $f(w_K) - f^* \leq \varepsilon$ for convex function $f$, GD with constant step-size initialized at $w_0$, needs at least*

$$K \geq \frac{H_1(f(w_0)-f^*)}{\log(f(w_0)-f^*)+1} \frac{f(w_0)-f^*-\epsilon}{4\epsilon} \quad iterations.$$

3. *To satisfy $f(w_K) - f^* \leq \varepsilon$ for $\mu$-PL function $f$ (but not necessarily convex), GD with constant step-size initialized at $w_0$, needs at least*

$$K \geq \frac{H_1}{4\mu} \frac{(f(w_0)-f^*)}{\log(f(w_0)-f^*)+1} \log\left(\frac{f(w_0)-f^*}{\epsilon}\right) \quad iterations.$$

*Proof.* Consider constants $H_1, M > 1$ and the function

$$f(w) = \begin{cases} \frac{e^{-\sqrt{H_1}w}}{e}, & \text{if } w < -\frac{1}{\sqrt{H_1}} \\ \frac{H_1 w^2}{2} + \frac{1}{2}, & \text{if } w \in \left[-\frac{1}{\sqrt{H_1}}, \frac{1}{\sqrt{H_1}}\right] \\ \frac{e^{\sqrt{H_1}w}}{e}, & \text{if } w > \frac{1}{\sqrt{H_1}}. \end{cases}$$

This function is $(H_0, H_1)$-smooth with $H_0 = H_1/2$ and convex, thus it also belongs to the objective function class.

We consider GD for the function $f$ starting from the point

$$w_0 = \frac{\log M + 1}{\sqrt{H_1}} > 1.$$

Notice that $f(w_0) = M$ and $\|\nabla f(w_0)\| = M\sqrt{H_1}$.

If we choose the step-size $\eta$ of GD larger than $2w_0/M\sqrt{H_1}$, it holds

$$w_1 = w_0 - \eta\nabla f(w_0) < w_0 - (2w_0/M\sqrt{H_1})M\sqrt{H_1} = -w_0.$$

Thus, $w_1$ is negative and further from the optimum (which is 0) compared to $w_0$.

By the structure of the function, we can show that $x_2$ will be even further. Since the function is totally symmetric, the effect of one step of GD starting from $w_1$ is the same as if it would start from $-w_1$. Thus, it suffices to show that $\tilde{w}_1 = -w_1 - \eta\nabla f(-w_1)$ is further from 0 compared to $-w_1$. Since $|w_1| > |w_0|$, it holds $-w_1 > w_0$. We consider the function

$$g(y) = |y - \eta\nabla f(y)| - |y|$$

for $y > \frac{1}{\sqrt{H_1}}$. Then, we have

$$g(y) = \left|y - \eta\sqrt{H_1}\frac{e^{\sqrt{H_1}y}}{e}\right| - |y|.$$

It is simple to see that in the part where this function is positive and $y > \frac{1}{\sqrt{H_1}}$, it is also increasing. Since $g(w_0) > 0$, $w_0 > \frac{1}{\sqrt{H_1}}$ and $-w_1 > w_0$, we have that $g(-w_1) > 0$. This means that $|\tilde{w}_1| > |w_1|$. Using an induction argument, we can show that the iterates of GD under such step-size diverge.

We conclude, that the step-size $\eta$ for our function class must satisfy

$$\eta \leq \frac{2w_0}{M\sqrt{H_1}} = \frac{2\log f(w_0) + 2}{f(w_0)H_1}. \tag{60}$$

This step-size bound will be used to derive the lower complexity bounds in all cases.

To establish lower bounds for the general and convex cases, we construct a function that contains a long, flat "runway" region where the gradient is small but non-zero. This forces any first-order method to take many small steps to traverse it.

For a parameter $\delta > 0$ (to be chosen later) and $H_0, H_1 > 0$, we define the following function $f_\delta(w)$;

The function is symmetric, $f_\delta(w) = f_\delta(-w)$, and defined for $x \geq 0$ as:

$$f_\delta(w) = \begin{cases} \frac{H_0}{2}w^2 & \text{if } 0 \leq w \leq X_1 \\ m(w - X_1) + \delta & \text{if } X_1 < w \leq X_2 \\ Ae^{\sqrt{H_1}(w - X_2)} + B & \text{if } w > X_2. \end{cases} \tag{61}$$

To make this function twice differentiable, we choose

$$m = \sqrt{2H_0\delta}$$
$$X_1 = \sqrt{2\delta/H_0}$$
$$X_2 = X_1 + (1 - \delta)/m$$
$$A = m/\sqrt{H_1}$$
$$B = 1 - A.$$

$f$ is $(H_0, H_1)$-smooth and its minimum is $f^* = f(0) = 0$.

**Lower bound in the general non-convex case:** We look for a point $w_K$ such that $\|\nabla f(w_K)\| \leq \epsilon$. To establish the lower bound, we set the gradient on the runway to be slightly larger than our target $\epsilon$, for instance, $\|\nabla f(w)\| = m = 2\epsilon$.

This choice requires us to set the construction parameter $\delta$ as follows:

$$\sqrt{2H_0\delta} = 2\epsilon \implies \delta = \frac{2\epsilon^2}{H_0}.$$

An algorithm must traverse the linear runway to enter the quadratic bowl, which is the only region where $\|\nabla f(w)\| \leq \epsilon$ is achievable.

`GD` update on the runway is $w_{k+1} = w_k - \eta\nabla f(w_k) = w_k - \eta m$, which implies that
$$w_K = w_0 - \eta K m.$$

Thus, if $w_0 = X_2$ (we start at the beginning of the runway) and $K < \frac{X_2 - X_1}{\eta m}$, then $w_K > X_1$ and we get $\|\nabla f(w_K)\| = 2\epsilon > \epsilon$. Thus, in order to get $\|\nabla f(w_K)\| \leq \epsilon$, we need to have

$$K \geq \frac{X_2 - X_1}{\eta m} = \frac{1 - \delta}{\eta m^2} = \frac{1 - \frac{2\epsilon^2}{H_0}}{4\eta\epsilon^2}.$$

Choosing $H_0 = 1$ (we can choose any positive constant) and plugging in the upper bound (60) for the step-size $\eta$, we get that $K$ must satisfy

$$K \geq \frac{f(w_0)H_1}{8(\log f(w_0) + 1)} \frac{1 - 2\epsilon^2}{\epsilon^2}.$$

Noticing that $f(w_0) = 1$ and $f^* = 0$, it holds $f(w_0) - f^* = 1$ and we get the desired lower bound:

$$K \geq \frac{H_1(f(w_0) - f^*)}{\log(f(w_0) - f^*) + 1} \frac{f(w_0) - f^* - 2\epsilon^2}{8\epsilon^2}.$$

**Lower bound in the convex case:** For this scenario, the target accuracy $\epsilon$ directly maps to our construction parameter. We set $\delta = \epsilon$ (61). The function $f_\epsilon(w)$ is convex and is constructed such that the linear runway begins at the point $(X_1, \epsilon)$. An algorithm starting at some point $w_0 = X_2$ where $f(w_0) = 1$ must traverse the runway from $X_2$ down to $X_1$ to achieve the desired accuracy.

On this runway, the gradient has a constant magnitude $m = \sqrt{2H_0\epsilon}$. Similarly as before, we have that if $K < \frac{X_2 - X_1}{\eta m}$, then $w_K > X_1$ and we get $f(w_K) - f^* > \epsilon$. Thus, we need to have

$$K \geq \frac{X_2 - X_1}{\eta m} = \frac{1 - \epsilon}{\eta m^2} = \frac{f(w_0) - f^* - \epsilon}{2\eta H_0 \epsilon}$$

to achieve $\epsilon$ accuracy for the function value.

Substituting, the upper bound (60) for $\eta$ and $H_0 = 1$, we get the desired result.

**Lower bound in the PL case:** The linear runway construction is not $\mu$-PL. For the third case, we need to construct a different function. We construct a fixed function, independent of $\epsilon$.

Let $C_0 > 0$ and $0 < \mu \leq 1$. We define a fixed connection point $w_c = \sqrt{2C_0/\mu}$. The function is symmetric and defined for $w \geq 0$ as:

$$f(w) = \begin{cases} \frac{\mu}{2}w^2 & \text{if } 0 \leq w \leq w_c \\ Ae^{\sqrt{H_1}(x - w_c)} + B & \text{if } w > w_c \end{cases} \tag{62}$$

where $A = \sqrt{2C_0\mu/H_1}$ and $B = C_0 - A$ are chosen to ensure the function is $C^1$ at $w_c$. This function is $\mu$-strongly convex (thus also $\mu$-PL) and belongs to the class of $(H_0, H_1)$ functions.

Our goal is to find again a point $w_K$ such that $f(w_K) - f^* \leq \epsilon$.

We analyze the performance of GD on the quadratic part of this function, $f(w) = \frac{\mu}{2}w^2$. An algorithm starting at $w_0 = w_c$ will have an initial function value of $f(w_0) = C_0$. The update rule with a fixed step size $\eta$ is:

$$w_{k+1} = w_k - \eta\nabla f(w_k) = w_k - \eta(\mu w_k) = (1 - \eta\mu)w_k.$$

After $K$ iterations, we have $w_K = (1 - \eta\mu)^K w_0$. We want to find the number of iterations $K$ needed to ensure $f(w_K) \leq \epsilon$.

$$f(w_K) = \frac{\mu}{2}w_K^2 = \frac{\mu}{2}(1 - \eta\mu)^{2K}w_0^2 = f(w_0)(1 - \eta\mu)^{2K} \leq \epsilon.$$

For this to hold, we need

$$f(w_0)(1 - \eta\mu)^{2K} \leq \epsilon \implies (1 - \eta\mu)^{2K} \leq \frac{\epsilon}{f(w_0)}.$$

Taking the logarithm of both sides and using the inequality $\log(1 - z) \leq -z$:

$$2K\log(1 - \eta\mu) \leq \log\left(\frac{\epsilon}{f(w_0)}\right), \quad \text{if } -2K(\eta\mu) \leq -\log\left(\frac{f(w_0)}{\epsilon}\right).$$

Solving for $K$, we get:

$$K \geq \frac{1}{2\eta\mu}\log\left(\frac{f(w_0)}{\epsilon}\right).$$

Substituting the upper bound (60) for the step-size $\eta$ and $f^* = 0$, we get the desired lower complexity bound.

$\square$

Table J.1: Detailed training details of language models and model configurations for the results in Figures 3 and 4. The implementation is based on Ajroldi (2024).

| Model | Configuration | MLP Type | Backbone | Normalization | Position Embeddings | Precision | Dropout |
|---|---|---|---|---|---|---|---|
| 70M | # Layers: 6
# heads: 8
hidden size: 512
seq. length: 1024
batch size: 256
weight decay: 0
cooldown steps: 20 %
grad clip: 1.0
tokens: 1.2B | SwiGLU (Shazeer, 2020) | PreLN transformer (Xiong et al., 2020) with skip connections | RMSnorm (Zhang & Sennrich, 2019) | MLP and Attention layers with variance: $0.02/\sqrt{\# \text{ layers}}$ Other layers: 0.02 std. dev. Biases are always initialized at zero | Mixed precision FP16 | Disabled for both hidden and attention layers |
| 160M | # Layers: 12
# heads: 12
hidden size: 1024
seq. length: 2048
batch size: 256
weight decay: 0.1
cooldown steps: 20 %
grad clip: 1.0
tokens: 1.2B | SwiGLU (Shazeer, 2020) | PreLN transformer (Xiong et al., 2020) with skip connections | RMSnorm (Zhang & Sennrich, 2019) | MLP and Attention layers with variance: $0.02/\sqrt{\# \text{ layers}}$ Other layers: 0.02 std. dev. Biases are always initialized at zero | Mixed precision FP16 | Disabled for both hidden and attention layers |
| 410M | # Layers: 6
# heads: 8
hidden size: 512
seq. length: 2048
batch size: 256
weight decay: 0.1
cooldown steps: 20 %
grad clip: 1.0
tokens: 3.2B | SwiGLU (Shazeer, 2020) | PreLN transformer (Xiong et al., 2020) with skip connections | RMSnorm (Zhang & Sennrich, 2019) | MLP and Attention layers with variance: $0.02/\sqrt{\# \text{ layers}}$ Other layers: 0.02 std. dev. Biases are always initialized at zero | Mixed precision FP16 | Disabled for both hidden and attention layers |

Table J.2: Detailed training details of image classification and model configurations for the results in Figure 5. The implementation is based on Ajroldi (2025).

| Model | Configuration | MLP Type | Backbone | Normalization | Position Embeddings | Stochastic Depth via DropPath |
|---|---|---|---|---|---|---|
| ViT-Tiny | # Patch size: 4
# heads: 8
Embedding size: 192
# layers: 12
# heads: 3
MLP ratio: 3
Class token: True
Drop path rate: 0.1
grad clip: Null | GELU (Hendrycks & Gimpel, 2016) | PreLN transformer (Xiong et al., 2020) with skip connections | LayerNorm (Ba et al., 2016) | LayerNorm: 1 Biases: 0 Other layers: 0.02 std. dev. | Residual branches are randomly dropped with a linearly increasing drop rate across depth |

## J EXPERIMENTAL DETAILS AND ADDITIONAL ABLATIONS

### J.1 EXPERIMENTAL SETUP

**Language Modeling.** Our training of language models is based on the Plain LM GitHub repository (Ajroldi, 2024) with small changes. The implementation is based on NanoGPT (Karpathy, 2022), and it includes recent improvements such as RMSNorm (Zhang & Sennrich, 2019), Rotational Positional Embeddings (Su et al., 2024), and SwiGLU activations (Shazeer, 2020). All details are reported in Table J.1.

**Image Classification.** The implementation of vision tasks is based on the GitHub repository (Ajroldi, 2025) with minor changes. Similarly, we report the training details of ViT training in Table J.2. It includes LayerNorm (Ba et al., 2016), GELU activations (Hendrycks & Gimpel, 2016), and drop path.

**Remark J.1.** *The results in Figures 1 and 2 are done with gradient clipping 1.0 and a small LR $10^{-4}$ to make small steps in the loss landscape from the initialization. Such an approach allows for tracking better the smoothness-loss dependency around the initialization.*

### J.2 ADDITIONAL RESULTS ON VERIFICATION OF THE PROPOSED CONDITION

### J.3 RESULTS VARYING RANDOM SEED

In this section, we demonstrate that the obtained results in Figures 1 and 2 are consistent when changing the random seed. Random seed changes the initialization of the models, thus leading to exploration of various parts of the landscape. We report the results in Figure J.1. According to them, in all the cases, the linear decay of the smoothness with the train loss is observed at the beginning.

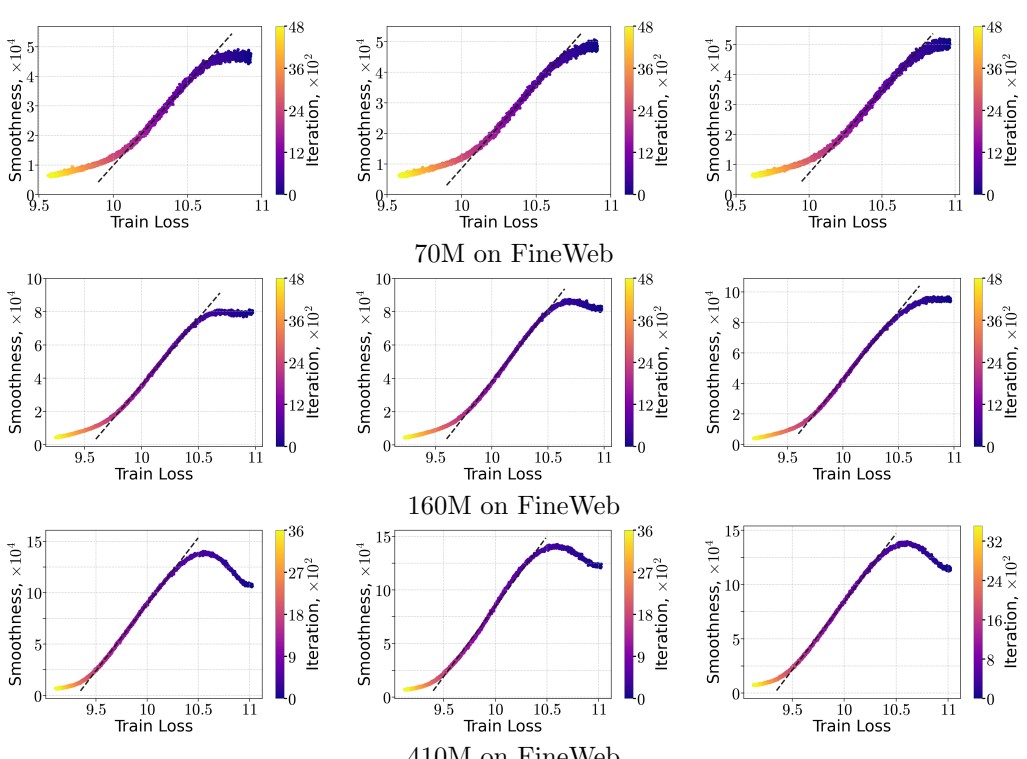

Figure J.1: Local smoothness approximation versus training loss for language models of varying sizes and random seed on the FineWeb dataset. Models are trained with SGD at a constant learning rate of $10^{-4}$. Each dot represents the estimated local smoothness and stochastic training loss at a given iteration, with color indicating training progress, while the black dashed line shows the best linear fit. For much of early training, the relation is well-approximated by a line, aside from the very initial phase where smoothness behaves differently. This deviation likely arises because the linear fit reflects only an upper bound, suggesting that a more complex functional dependence may be necessary.

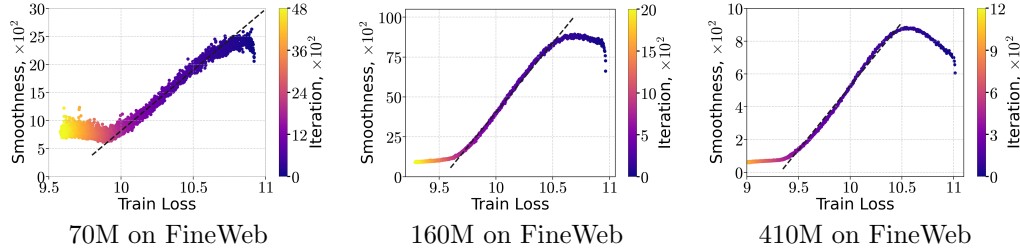

Figure J.2: Local smoothness approximation versus training loss for language models of varying sizes and random seed on the FineWeb dataset. Models are trained with `Adam` at a constant learning rate of $10^{-7}$. Each dot represents the estimated local smoothness and stochastic training loss at a given iteration, with color indicating training progress, while the black dashed line shows the best linear fit. For much of early training, the relation is well-approximated by a line, aside from the very initial phase where smoothness behaves differently. This deviation likely arises because the linear fit reflects only an upper bound, suggesting that a more complex functional dependence may be necessary.

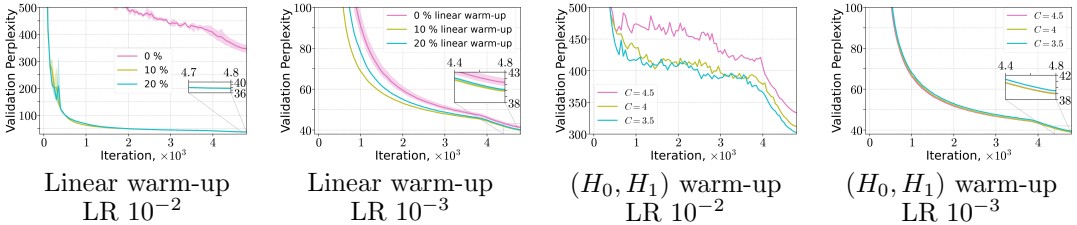

Figure J.3: Training of 70M language model on FineWeb dataset varying the length of linear warm-up (two left figures) and threshold $C$ of $(H_0, H_1)$ warm-up (two right figures) for the peak learning rate $10^{-2}$ and $10^{-3}$.

### J.3.1 Verification with Adam

Next, we switch to `Adam` optimizer to verify the proposed $(H_0, H_1)$-smoothness condition. We test the results on language models of size 70M, 160M, and 410M. The results are reported in Figure J.2. Similar to the setting in the main body, we use a small constant learning rate $10^{-7}$, which allows moving slowly in the landscape. We observe that `Adam` also demonstrates a linear dependency between local smoothness approximation and train loss. However, we observe that `Adam` stays in this linear decaying part of the landscape for fewer iterations, especially for larger models, than `SGD` does. This might suggest that for `Adam` the warm-up phase should be shorter.

## J.4 Ablation Studies

### J.4.1 Performance Varying Warm-up Length

**Language Modeling.** In this section, we investigate how warm-up length influences training. As shown in Figures J.3-J.5, using a 10–20% linear warm-up yields the best validation perplexity, demonstrating that warm-up improves the final performance of the models. We also find that warm-up enables convergence even with relatively large peak learning rates $10^{-2}$ for the 70M model and $3 \cdot 10^{-3}$ for the 160M model, whereas training without warm-up performs significantly worse at these values. Similar trends have been reported by Wortsman et al. (2023). Finally, we observe that the $(H_0, H_1)$ warm-up is less robust to the choice of peak learning rate for the 70M model, resulting in higher validation perplexity. However, once the peak learning rate is properly tuned (within $10^{-3}$–$3 \cdot 10^{-3}$), it becomes less sensitive to the choice of the constant $C$.

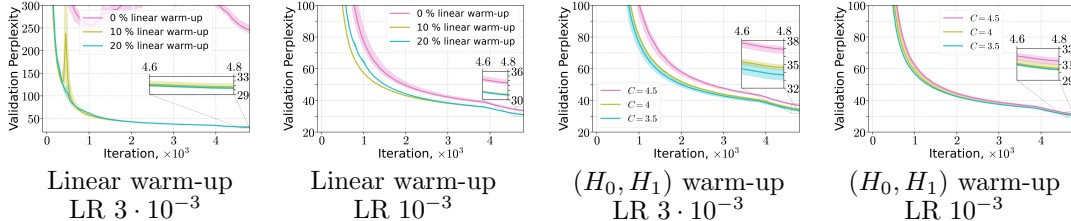

Figure J.4: Training of 160M language model on FineWeb dataset varying the length of linear warm-up (two left figures) and threshold $C$ of $(H_0, H_1)$ warm-up (two right figures) for the peak learning rate $3 \cdot 10^{-3}$ and $10^{-3}$.

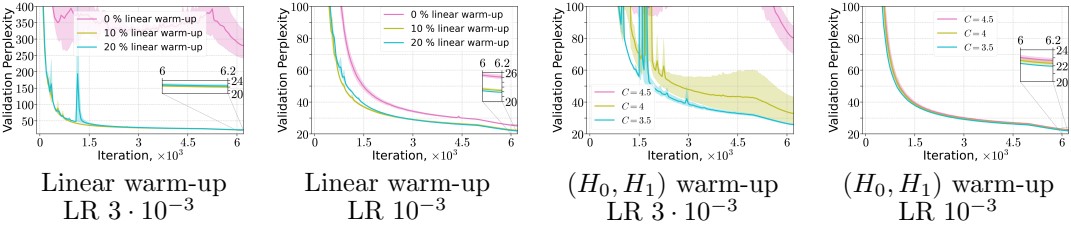

Figure J.5: Training of 410M language model on FineWeb dataset varying the length of linear warm-up (two left figures) and threshold $C$ of $(H_0, H_1)$ warm-up (two right figures) for the peak learning rate $3 \cdot 10^{-3}$ and $10^{-3}$.

**Image Classification with ViT.** Now we turn to the same test, but when training the ViT model on the ImageNet32 dataset. In contrast to language modeling results, ViT with linear and $(H_0, H_1)$ warm-up strategies demonstrates similar performance. We report the results in Figure J.6.

### J.4.2 PERFORMANCE VARYING PEAK LEARNING RATE

**Language Modeling.** We now present performance curves under different peak learning rates for all warm-up strategies: 10% linear warm-up and $(H_0, H_1)$ warm-up with $C = 4$. As shown in Figure J.7, smaller models are less sensitive to high peak learning rates when using $(H_0, H_1)$ warm-up. However, for the largest 410M model, even slightly exceeding the optimal peak learning rate produces large spikes with $(H_0, H_1)$ warm-up, though `AdamW` eventually recovers. In contrast, linear warm-up proves more robust to peak learning rate selection.

**Image Classification with ViT.** Now we conduct similar tests as in the previous section. We report the results for three warm-up strategies: 5% linear warm-up and $(H_0, H_1)$ warm-up with $C = 3$. In this case, we observe that both warm-up schedules achieve similar performance; see Figure J.8.

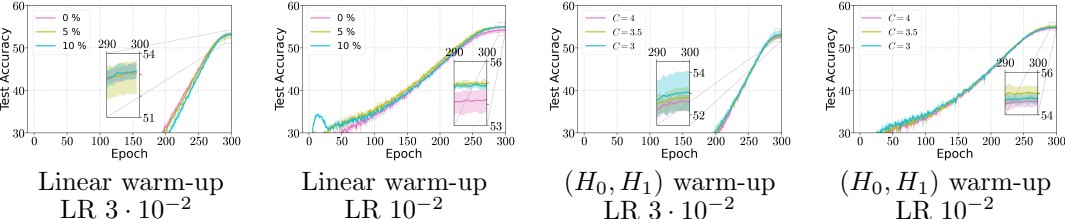

Figure J.6: Training of ViT model on ImageNet32 dataset varying the length of linear warm-up (two left figures) and threshold $C$ of $(H_0, H_1)$ warm-up (two right figures) for the peak learning rate $3 \cdot 10^{-2}$ and $10^{-2}$.

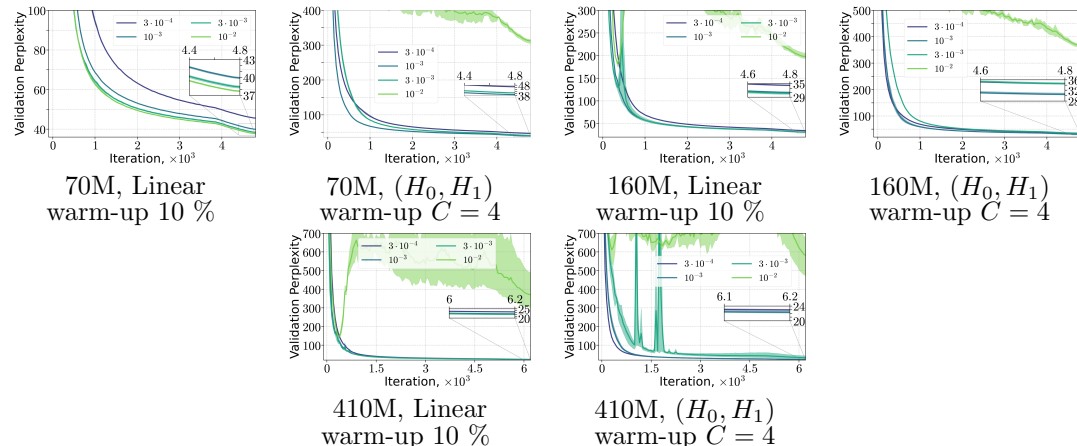

Figure J.7: Training of 70M and 160M language models on FineWeb dataset, varying the peak learning rate with 10 % linear warm-up and $(H_0, H_1)$ warm-up with $C = 4$.

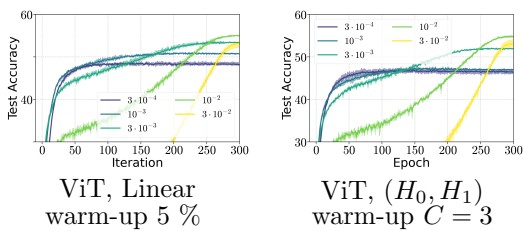

Figure J.8: Training of ViT model on ImageNet32 dataset, varying the peak learning rate with 5 % linear warm-up and $(H_0, H_1)$ warm-up with $C = 3$.

# K    INITIALIZATION AND PROGRESSIVE SHARPENING

## K.1    A THEORETICAL ANALYSIS OF THE DIFFERENT SHARPENING PHASES IN A SIMPLE MODEL

We analyze the *derivative of the spectral norm of the Hessian* of a simple $1 \times 1 \times 1$ network with linear activation $\phi(x) = x$:

$$f(u, v) = (y - uvx)^2$$

over the course of the gradient flow ODE. We assume a data sample $x, y \neq 0$. The parameters $u(t)$ and $v(t)$ evolve over time via gradient flow as: $\frac{du}{dt} = -g_u$ and $\frac{dv}{dt} = -g_v$, where $g_u = \frac{\partial f(u,v)}{\partial u}, g_v = \frac{\partial f(u,v)}{\partial v}$

Let $R := y - uvx$ be the residual error. We have

$$g_u = \frac{\partial f}{\partial u} = 2(y - uvx) \cdot (-vx) = -2vxR$$

$$g_v = \frac{\partial f}{\partial v} = 2(y - uvx) \cdot (-ux) = -2uxR.$$

The gradient vector is $g = \begin{pmatrix} g_u \\ g_v \end{pmatrix}$.

The Hessian matrix is $H = \nabla^2 f = \begin{pmatrix} H_{11} & H_{12} \\ H_{21} & H_{22} \end{pmatrix}$. We define its components as:

$$A = H_{11} = \frac{\partial g_u}{\partial u} = \frac{\partial}{\partial u}(-2vyx + 2uv^2x^2) = 2v^2x^2$$

$$C = H_{22} = \frac{\partial g_v}{\partial v} = \frac{\partial}{\partial v}(-2uyx + 2u^2vx^2) = 2u^2x^2$$

$$B = H_{12} = \frac{\partial g_u}{\partial v} = \frac{\partial}{\partial v}(-2vyx + 2uv^2x^2) = -2yx + 4uvx^2 = 2x(2uvx - y)$$

$$H_{21} = \frac{\partial g_v}{\partial u} = \frac{\partial}{\partial u}(-2uyx + 2u^2vx^2) = -2yx + 4uvx^2 = B.$$

So, the Hessian is:

$$H(t) = \begin{pmatrix} 2v^2x^2 & 2x(2uvx - y) \\ 2x(2uvx - y) & 2u^2x^2 \end{pmatrix}.$$

The spectral norm $\|H\|_2$ of a $2 \times 2$ symmetric matrix of the form $\begin{bmatrix} A & B \\ B & C \end{bmatrix}$ is its largest eigenvalue in absolute value. The two eigenvalues $\lambda_\pm$ are found by solving the characteristic equation $\det(H - \lambda I) = 0$:

$$\lambda^2 - (A + C)\lambda + (AC - B^2) = 0.$$

The solution of this quadratic equation gives the two eigenvalues:

$$\lambda_\pm = \frac{(A + C) \pm \sqrt{(A + C)^2 - 4(AC - B^2)}}{2} = \frac{(A + C) \pm \sqrt{(A - C)^2 + 4B^2}}{2}.$$

The spectral norm is the larger of these in magnitude, i.e.

$$\|H\|_2 = \lambda_+ = \frac{(A + C) + \sqrt{(A - C)^2 + 4B^2}}{2}.$$

We define $S_\lambda := \frac{1}{2}\sqrt{(A - C)^2 + 4B^2}$. It holds: $\|H\|_2 = \lambda_+ = \frac{1}{2}(A + C) + S_\lambda$.

We target to compute $\frac{d}{dt}\|H\|_2$ and analyze its sign.

First, we must find $\dot{H}$, which requires the 3rd derivatives of $f$:

- $\frac{\partial H_{11}}{\partial u} = \frac{\partial}{\partial u}(2v^2x^2) = 0$
- $\frac{\partial H_{11}}{\partial v} = \frac{\partial}{\partial v}(2v^2x^2) = 4vx^2$
- $\frac{\partial H_{12}}{\partial u} = \frac{\partial}{\partial u}(4uvx^2 - 2xy) = 4vx^2$
- $\frac{\partial H_{12}}{\partial v} = \frac{\partial}{\partial v}(4uvx^2 - 2xy) = 4ux^2$
- $\frac{\partial H_{22}}{\partial u} = \frac{\partial}{\partial u}(2u^2x^2) = 4ux^2$
- $\frac{\partial H_{22}}{\partial v} = \frac{\partial}{\partial v}(2u^2x^2) = 0$

Now we find the components of $\dot{H}$ using the chain rule:

$$\dot{H}_{ij} = \frac{\partial H_{ij}}{\partial u}\dot{u} + \frac{\partial H_{ij}}{\partial v}\dot{v} = -g_u\left(\frac{\partial H_{ij}}{\partial u}\right) - g_v\left(\frac{\partial H_{ij}}{\partial v}\right).$$

- $\dot{H}_{11} = -g_u(0) - g_v(4vx^2) = -g_v(4vx^2)$
- $\dot{H}_{22} = -g_u(4ux^2) - g_v(0) = -g_u(4ux^2)$
- $\dot{H}_{12} = -g_u(4vx^2) - g_v(4ux^2)$

We substitute $g_u = -2vxR$ and $g_v = -2uxR$ and we get

- $\dot{H}_{11} = -(-2uxR)(4vx^2) = 8uvx^3R$

- $\dot{H}_{22} = -(-2vxR)(4ux^2) = 8uvx^3R$

This reveals that $\dot{H}_{11} = \dot{H}_{22}$, which will be used later. We denote these terms by $D_H$:

$$D_H := \dot{H}_{11} = \dot{H}_{22} = 8uvx^3R.$$

The off-diagonal term is:

$$E_H := \dot{H}_{12} = -(-2vxR)(4vx^2) - (-2uxR)(4ux^2)$$
$$= 8v^2x^3R + 8u^2x^3R = 8x^3(u^2 + v^2)R$$

So, $\dot{H} = \begin{pmatrix} D_H & E_H \\ E_H & D_H \end{pmatrix}$.

For a $2 \times 2$ symmetric matrix $H$ with $A = H_{11}$ and $C = H_{22}$, the derivative of the largest eigenvalue $\lambda_+$ is:

$$\frac{d\lambda_+}{dt} = \frac{1}{2}(\dot{A} + \dot{C}) + \frac{(A - C)(\dot{A} - \dot{C}) + 4B\dot{B}}{4S_\lambda}.$$

Our simplification $\dot{A} = \dot{C} = D_H$ gives a cleaner expression:

$$\frac{d\lambda_+}{dt} = \frac{1}{2}(D_H + D_H) + \frac{(A - C)(0) + 4BE_H}{4S_\lambda} = D_H + \frac{BE_H}{S_\lambda}.$$

Substituting $S_\lambda = \frac{1}{2}\sqrt{(A - C)^2 + 4B^2}$, we get

$$\frac{d}{dt}\|H\|_2 = D_H + \frac{2BE_H}{\sqrt{(A - C)^2 + 4B^2}}.$$

Substituting our components $A, B, C, D_H, E_H$ gives the final formula:

$$\frac{d}{dt}\|H\|_2 = 8uvx^3R + \frac{2 \cdot [2x(2uvx - y)] \cdot [8x^3(u^2 + v^2)R]}{\sqrt{(2v^2x^2 - 2u^2x^2)^2 + 4(2x(2uvx - y))^2}}.$$

Let $B_{\text{sign}} := y - 2uvx$. Then $2uvx - y = -B_{\text{sign}}$ and plugging it in the previous formula, we have

$$\frac{d}{dt}\|H\|_2 = 8uvx^3R + \frac{2 \cdot (2x(-B_{\text{sign}})) \cdot (8x^3(u^2 + v^2)R)}{2\sqrt{x^4(v^2 - u^2)^2 + 4x^2(-B_{\text{sign}})^2}}$$

$$= 8uvx^3R - \frac{32x^4(u^2 + v^2)RB_{\text{sign}}}{2|x|\sqrt{x^2(v^2 - u^2)^2 + 4B_{\text{sign}}^2}}.$$

Let $c = u^2 - v^2$ (a constant, as we will prove later). To simplify the fraction, we use the identity $\frac{x^4}{|x|} = \frac{x^4}{x \cdot \text{sgn}(x)} = x^3 \cdot \text{sgn}(x)$ and we get

$$\frac{d}{dt}\|H\|_2 = 8uvx^3R - 16 \cdot \text{sgn}(x) \cdot x^3(u^2 + v^2)R \cdot \frac{B_{\text{sign}}}{\sqrt{x^2c^2 + 4B_{\text{sign}}^2}}.$$

Factoring out the common term $8x^3R$, we arrive at the general formula for the derivative, valid for all $x, y, u, v$:

$$\frac{d}{dt}\|H\|_2 = 8x^3R \left( uv - 2 \cdot \text{sgn}(x) \cdot (u^2 + v^2) \cdot \frac{B_{\text{sign}}}{\sqrt{x^2c^2 + 4B_{\text{sign}}^2}} \right) \tag{63}$$

Before analyzing specific paths, we prove that the quantity $u^2 - v^2$ stays constant over the course of gradient flow.

Indeed, we compute the time-derivative of the quantity $(u^2 - v^2)$:

$$\frac{d}{dt}(u^2 - v^2) = 2u\dot{u} - 2v\dot{v}.$$

From the gradient flow definition, $\dot{u} = -g_u = 2vxR$ and $\dot{v} = -g_v = 2uxR$. Substituting these in:

$$\frac{d}{dt}(u^2 - v^2) = 2u(2vxR) - 2v(2uxR) = 4uvxR - 4uvxR = 0.$$

Because the time-derivative is always zero, the quantity $(u^2 - v^2)$ is a conserved quantity. Any trajectory is constrained to a manifold where $u^2 - v^2 = c$, with $c$ is determined by the random initial conditions $c = u(0)^2 - v(0)^2$.

We now pass to the analysis of the sign of $\frac{d}{dt}\|H\|_2$. For notational simplicity, let $S = u^2 + v^2$ and we have

$$\frac{d}{dt}\|H\|_2 = \underbrace{8x^3R}_{\text{Term 1}} \cdot \underbrace{\left( uv - \frac{2\operatorname{sgn}(x) \cdot S \cdot B_{\text{sign}}}{\sqrt{x^2c^2 + 4B_{\text{sign}}^2}} \right)}_{\text{Term 2}}$$

The trajectory now consists of three phases, defined by the boundaries $x^2c^2 = 4B_{\text{sign}}^2$ and $B_{\text{sign}} = 0$.

**First phase:** We define the initial regime by the condition $x^2c^2 \leq 4B_{\text{sign}}^2$. This holds near the origin because $c \approx 0$ and $B_{\text{sign}} \approx y \neq 0$. In this regime: $\operatorname{sgn}(R) \approx \operatorname{sgn}(y)$ and $\operatorname{sgn}(B_{\text{sign}}) \approx \operatorname{sgn}(y)$.

1. **Term 1:** $\operatorname{sgn}(\text{Term 1}) = \operatorname{sgn}(x^3R) \approx \operatorname{sgn}(x^3y) = \operatorname{sgn}(xy)$.

2. **Term 2:** Let $F = \frac{2S|B_{\text{sign}}|}{\sqrt{x^2c^2 + 4B_{\text{sign}}^2}}$. We use the identity $uv = \operatorname{sgn}(uv) \cdot \frac{1}{2}\sqrt{S^2 - c^2}$ (easy to prove).

We must analyze two scenarios

- **Case A:** $\operatorname{sgn}(uv) = \operatorname{sgn}(xy)$. Then,

$$\text{Term 2} = \operatorname{sgn}(xy)\left[ \frac{1}{2}\sqrt{S^2 - c^2} - |F| \right].$$

We have $\frac{1}{2}\sqrt{S^2 - c^2} \leq \frac{S}{2}$. We also have $|F| \geq \frac{2S|B_{\text{sign}}|}{\sqrt{8B_{\text{sign}}^2}} = \frac{S}{\sqrt{2}} > \frac{S}{2}$. Thus, the term inside the bracket is negative and

$$\operatorname{sgn}\left( \frac{d}{dt}\|H\|_2 \right) = -\operatorname{sgn}(xy)^2 = -1.$$

- **Case B:** $\operatorname{sgn}(uv) = -\operatorname{sgn}(xy)$

$$\text{Term 2} = -\operatorname{sgn}(xy)\left[ \frac{1}{2}\sqrt{S^2 - c^2} + |F| \right].$$

In this case, it holds

$$\operatorname{sgn}\left( \frac{d}{dt}\|H\|_2 \right) = -\operatorname{sgn}(xy)^2 \operatorname{sgn}\left( \frac{1}{2}\sqrt{S^2 - c^2} + |F| \right) = -1.$$

In both cases, the derivative of the spectral norm is negative, which means that we enter a flattening phase.

**Second phase:** This initial flattening phase is only guaranteed as long as our assumption $x^2c^2 \leq 4B_{\text{sign}}^2$ holds. As the flow moves, $uvx$ increases, so $B_{\text{sign}} = y - 2uvx$ gets smaller. Eventually, we will enter an ambiguous phase where $x^2c^2 > 4B_{\text{sign}}^2$, but we still have $\operatorname{sgn}(B_{\text{sign}}) = \operatorname{sgn}(y)$ (because the flow has not yet reached $uvx = y/2$). In this region, our bounding logic for $|F|$ is inconclusive, and the sign of the derivative is unknown.

**Third phase:** The flow then crosses the mid-point at $uvx = y/2$, which means $\text{sgn}(B_{\text{sign}})$ finally flips to $-\text{sgn}(y)$. The flow is now in a "correct" quadrant (having been repelled by the origin), so $\text{sgn}(uv) = \text{sgn}(xy)$.

- $\text{sgn}(\text{Term 1}) = \text{sgn}(x^3 R) = \text{sgn}(x^3 y) = \text{sgn}(xy)$ (since $\text{sgn}(R) = \text{sgn}(y)$ always).
- The sign of $F$ is now $\text{sgn}(F) = \text{sgn}(x \cdot B_{\text{sign}}) = \text{sgn}(x) \cdot (-\text{sgn}(y)) = -\text{sgn}(xy)$.
- This means $\text{sgn}(uv)$ and $\text{sgn}(F)$ are now opposites.
- Term $2 = uv - F$.
  - If $\text{sgn}(xy) = +1$: $\text{sgn}(uv) = +1$ and $\text{sgn}(F) = -1$. Term 2 is $(\text{pos}) - (\text{neg}) = +$.
  - If $\text{sgn}(xy) = -1$: $\text{sgn}(uv) = -1$ and $\text{sgn}(F) = +1$. Term 2 is $(\text{neg}) - (\text{pos}) = -$.
- In both sub-cases, $\text{sgn}(\text{Term 2})$ is $\text{sgn}(xy)$.

**Total Sign for Phase 3:**

$$\text{sgn}\left(\frac{d}{dt}\|H\|_2\right) = \text{sgn}(\text{Term 1}) \cdot \text{sgn}(\text{Term 2}) = \text{sgn}(xy) \cdot \text{sgn}(xy) = +1.$$

Thus, a sharpening phase is guaranteed after the $uvx = y/2$ boundary.

**Remark:** If $c = 0$, the initial flattening phase lasts exactly until the mid-point, where it gives place to sharpening.

### K.2 EFFECT OF INITIALIZATION

In this section, we examine how the choice of initialization influences the empirical verification of our condition on the 70M language model trained with clipping 1.0 and a fixed learning rate. We evaluate two initialization strategies. The first follows the approach used in modern GPT-style architectures, where the variance of the weights in the MLP or GLU blocks and in the attention output layers is scaled as 0.02/n˙layer, with n˙layer denoting the layer index. We refer to this strategy as "scaling with depth." The second strategy uses a fixed variance of 0.02, independent of depth, which is the default choice in many implementations. We refer to this strategy as "no scaling with depth." We estimate the smoothness throughout training using the same methodology as in Section 5. We highlight that in this set of experiments, we use gradient clipping to 1, which is a standard training technique used in practice, which allows to restrict the update magnitude and make constant steps in the landscape.

We present the results in Figure K.1 varying the fixed learning rate hyperparameter of `SGD`. We observe the following results

- After an initial sharpness reduction (flattening) phase, `SGD` enters a gradual sharpening phase where the sharpness grows. In this regime, our condition does not describe the smoothness well anymore. After the gradual sharpening phase, `SGD` enters `EoS` stage where the sharpness oscillates around 2/LR stability threshold (only observed for LR 1e-2).
- Importantly, the transition from sharpness reduction to the gradual sharpening phase happens at the same loss value regardless of LR choice, which indicates a strong connection between the smoothness and the loss value.
- Our condition describes well the sharpness reduction phase, also observed in (Kalra & Barkeshli, 2024). In contrast to that work, we describe the reduction phase analytically. Using our theoretically derived learning-rate warm-up strategy allows to avoid instabilities due to high values of the sharpness at the beginning and leads to better final performance.
- "Scaling with depth" strategy initializes the model closer to the origin. This results in a larger initial sharpness in comparison with "no scaling with depth" scheme. This aligns with our theoretical calculations in K.1. We hypothesize that GPT-style

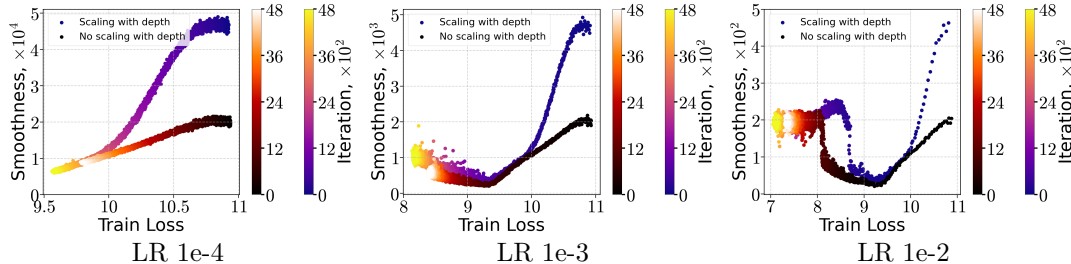

Figure K.1: Training of 70M model on FineWeb dataset with `SGD` varying fixed learning rate and initialization scheme. Each left color bar corresponds to "no scaling with depth" initialization scheme, while the right one — to "scaling with depth" strategy.

initialization requires a longer learning warm-up phase due to such high values of the sharpness at the beginning.

