# OpenReview forum: "Why Do We Need Warm-up? A Theoretical Perspective"
_ICLR.cc/2026/Conference — Submitted to ICLR 2026_

### Official Review · Reviewer_ryKS · 2025-10-25

**Soundness:** 2
**Presentation:** 3
**Contribution:** 3
**Rating:** 6
**Confidence:** 4

**Summary:**

This work theoretically analyzes learning rate phenomena from a theoretical perspective. Specifically, they study a new constraint on the loss landscape geometry that assumes that the the top eigenvalue of the loss Hessian $\lambda$ is bounded by the distance to the minima, i.e., $\lambda \leq H_0 + H_1 (f(w) - f^*)$. Through examples, the authors show that this condition theoretically holds in multiple cases. Next, the authors analyze the convergence of GD under the learning rate motivated by the $H_0-H_1$ condition. Finally, the authors test their learning rate warmup schedule in practice and show it works on par with linear warmup commonly used in practice.

**Strengths:**

* The paper is clearly written and the main results are easy to follow
* Strong theoretical analysis of learning rate warmup under the $H_0-H_1$ condition
* The authors propose a warmup schedule that performs on with linear warmup.
* The proposed warmup schedule is practical, as it only depends on the current loss and one hparam $C$. This warmup strategy can be perhaps an alternative to the linear schedule used in practice.

**Weaknesses:**

**$H_0-H_1$ condition**: Definition 3.1 states that the largest eigenvalue of the Hessian $\lambda$ is bounded by $H_0 + H_1 (f - f^*)$. This implies that as the loss decreases the upper bound of $\lambda$ reduces. The authors show that this condition holds in realistic situations (Figures 1, 2, I.1, I.2). However, there are empirical evidence against it. In full batch setting, its well (empirically) established that $\lambda$ increases throughout training [1], which is at odds with the submitted work that claims the exact opposite. In mini-batch setting, its known that $\lambda$ increases during training, albeit with a slower rate.

There can me multiple reasons why this happens, which I detail below:
1. The authors use a proxy for the max eigenvalue (line 262), which may have a different behavior than sharpness
2. Its known that very early in training, $\lambda$ may decrease early in training (Appendix A of [1]), which is rather short (10 steps) compared to the warmup duration (1000 steps). The authors use a very small learning rate (1e-04 for SGD, 1e-07 for Adam), which restricts the training to this $\lambda$ decrease phase. I would request the authors to rerun these experiments with a typical learning rate schedule and check if the decrease in curvature is observed during the entire warmup phase (both with their schedule and linear warmup).
3. For unusually large initializations, $\lambda$ may decrease throughout training [4], which aligns with the results in the paper. I would like the authors to clarify if the experiments in the submitted work are operating in this regime.

**The gap between theory and practice**: The theoretical analysis of the submitted work assumes $\lambda$ decreasing through the warmup phase, which corresponds to the 'natural sharpness reduction' phase described in [3]. However, as mentioned above, in realistic settings, $\lambda$ increases during training. Regardless of whether $\lambda$ increases or decreases, the effect of increasing learning rate is to reduce $\lambda$ [2, 3].

This creates a causal gap between the theory and practice:
1. **Theory**: $\lambda$ decreases during training, so increase $\eta$
2. **Practice**: $\eta$ increases which causes reduction in $\lambda$.

Furthermore, the proposed schedule does not care about the causality anymore. It increases the learning rate depending on loss change and does not care about whether the curvature increases or decreases. This aligns with the standard linear learning rate schedule used in practice.

[1] Gradient Descent on Neural Networks Typically Occurs at the Edge of Stability, 2021

[2] A Loss Curvature Perspective on Training Instability in Deep Learning, 2021

[3] Why Warmup the Learning Rate? Underlying Mechanisms and Improvements, 2024

[4] Universal Sharpness Dynamics in Neural Network Training: Fixed Point Analysis, Edge of Stability, and Route to Chaos, 2023

**Questions:**

* Line 67: "we provide empirical guarantees". I think there is a typo here. It should be empirical evidence rather than a guarantee.
* Equation 1: where do the constants 10, 20 come from?
* (Comment, not a question) Line 407: convergence is stochastic setting requires interpolation condition, whereas much of the realistic experiments (language modeling) are in underparameterized setting
* Can you check Figure 1, 2, I.1, I.2 for standard learning rates? How long the decrease phase lasts?
* Line 443, how did the authors come this practice schedule. It would be helpful to guide the reader.

---

> ### Author Response · Authors · 2025-11-21
> **Rebuttals (part I)**
>
> We sincerely thank the reviewer for the constructive feedback and the positive remarks on our presentation. We also appreciate the recognition of the strength of our theoretical analysis and the acknowledgment of the practical value of our warm-up schedule, which can serve as an alternative to the standard linear warm-up.
>
> We hope that the following clarifications will help re-evaluate our submission and potentially adjust the score.
>
> **W1, $(H_0,H_1)$ condition:**
>
> ***A:*** Thank you, that's a very interesting observation. For the sake of clarity, we structure our answer into several points.
>
> - **a.** Our proxy for measuring the smoothness constant is well established and used in prior work [1,2], since computing the exact Hessian norm is expensive for large models. Such a choice comes from the Taylor expansion of the gradient
> $\frac{\\|\nabla f(w\_{k+1})-\nabla f(w\_k)\\|}{\\|w\_{k+1}-w\_k\\|} = \frac{\\|\int\_0^1\nabla^2f(w\_k+\tau\eta(w\_{k+1}-w\_k))(w\_{k+1}-w\_k)d\tau\\|}{\\|w_{k+1}-w_k\\|}\underset{w_{k+1}\to w_k}{\to} \frac{\\|\int_0^1\nabla^2f(w_k)(w_{k+1}-w_k)d\tau\\|}{\\|w_{k+1}-w_k\\|}\le\lambda_{\max}(\nabla^2f(w_k)).$ The approximation is based on the assumption that the Hessian does not change fast (which is the case for small $\eta$).
>
> - **b.** In Appendix K, we calculate the derivative of the Hessian norm for a simple 2-layer network with 2 parameters and MSE loss. We show that the change of derivative can be split into 3 phases: initial flattening, followed by a mid uncharacterized phase, and finally, gradual sharpening. Such a split of the behaviour depends on the initialization: if the model is initialized close to the origin, then the initial sharpness is high and the reduction phase is more visible. These are, in general, consistent with standard results in the area ([3], Section 3).
>
> - **c.** Following the reviewer's request, we also rerun our experiments with a larger LR. We refer to Appendix K.2 for the convergence results. They confirm the reviewer's intuition under point 2. We observe that a small LR restricts SGD in the sharpness-reduction phase for more iterations, while increasing it allows SGD to enter progressive sharpening and EoS after.
>
> - **d.** We would also like to point out that measuring the duration of the reduction phase by counting iterations is not very informative, since it varies substantially with the optimizer and its hyperparameters. A more meaningful and fair comparison can be obtained by measuring this duration in terms of the loss. In our experiments, the sharpness-reduction phase spans the loss interval from approximately 11 down to 9.5 (see Figure K.1), while the final loss is typically around 3. This indicates that the sharpness-reduction phase accounts for roughly 20% of the overall training process.
>
>
> - **e.** We also examine how initialization affects the verification, both theoretically and empirically (see Section K). Bringing the model closer to the origin increases the sharpness at initialization. This effect is particularly evident for GPT-style initialization schemes, where the initialization variance decreases with depth. A higher initial sharpness could partly explain the central importance of learning-rate warm-up in the training of large language models. In contrast, using a constant-variance initialization places the model at a lower initial sharpness. We hypothesize that this difference may be one of the reasons why the sharpness-reduction phase has not always been observed in the EoS literature.
>
>
> [1] Zhang et al. Why gradient clipping accelerates training: A theoretical justification for adaptivity. arXiv preprint arXiv:1905.11881 (2019)
>
> [2] Riabinin et al. Gluon: Making Muon & Scion Great Again!(Bridging Theory and Practice of LMO-based Optimizers for LLMs). arXiv preprint arXiv:2505.13416 (2025).
>
> [3] Kalra, Dayal Singh, and Maissam Barkeshli. Why warmup the learning rate? underlying mechanisms and improvements, NeurIPS 2024.

---

> ### Author Response · Authors · 2025-11-21
> **Rebuttals (part II)**
>
> **W2: The gap between theory and practice:**
>
> ***Answer:*** Thank you for raising this important point. We would like to emphasize a few clarifications.
>
> - **a.** The theory is not assuming a causality that contradicts practice. Our analysis is intentionally restricted to the early phase of training, where the sharpness decreases before the progressive sharpening regime begins. This is a well-documented empirical phenomenon. In this regime, the optimality condition for stable descent requires adapting the learning rate to the changing curvature, and our analysis shows that warm-up provides exactly the appropriate scaling to remain within the stable region. In other words, we are characterizing why this initial reduction in $\lambda$ allows increasingly large steps.
>
> - **b.** Our analysis does not rely on this causal interpretation. Instead, it establishes that if the loss and curvature trajectories evolve according to the sharpness-reduction regime observed empirically, then warm-up is theoretically justified. The direction of causality is not required: the stability condition ultimately depends on the current curvature, whether it decreased "naturally" (the regime studied) or decreased effectively due to a slightly larger step size (practice). Hence, there is no contradiction between theory and practice.
>
> - **c.** Alignment with practical schedules: Finally, we agree with the reviewer that our schedule behaves similarly to a standard linear warm-up. This is a positive outcome: our contribution is precisely to provide a theoretical mechanism explaining why warm-up works in the early phase, not to replace all practical heuristics. The role of our analysis is to justify warm-up during the sharpness-reduction phase; the later progressive-sharpening phase lies outside the scope of our paper (where there is already a broad literature).
>
>
> **Q1: Pointing to a typo.**
>
> ***Answer:*** We thank the reviewer for this comment. We will adjust the list of contributions accordingly.
>
> **Q2: On constants in Eq. 1**
>
> ***Answer:*** These constants are taken to simplify calculations and can be chosen more carefully to derive tigther bounds w.r.t numerical constants. Such a choice of constants is mainly needed to satisfy inequalities of the form of (50), i.e., to obtain a descent.
>
> **Q3: On convergence in the stochastic regime**
>
> ***Answer:*** This is an important point. We acknowledge that obtaining convergence guarantees without assuming the interpolation condition is a significant question. By following the derivations used in the proof of Theorem 4.4, with only minor adjustments, we can show that
> $\frac{1}{K+1}\sum\_{k=0}^K\mathbb{E}\min\left\\{f(w\_k)-f^*,\frac{H\_0}{2nH\_1}\right\\}\le \mathcal{O}\left(\frac{H\_0\mathrm{dist}(w\_0,\mathcal{S})^2}{\theta^2(K+1)} + \sigma^2\_{\mathrm{int}}\right)$,
>
> where $\sigma^2\_{\mathrm{int}} := \mathbb{E}\_S[f^* - f\_S^*]$. This shows convergence to a neighbourhood of the solution that does not vanish, with $\sigma^2_{\mathrm{int}}$ capturing the expected gap between the global optimum and the local optima reached by SGD. However, we do not yet have a formal justification for whether this term must appear in general or whether it is simply a consequence of our proof strategy.
>
> We are also not aware of any existing convergence guarantees for SGD in finite–sum minimization setting under $(L_0, L_1)$-smoothness without the interpolation condition. Establishing such a result appears to require a more refined analysis and new proof techniques, which we leave to future work.
>
> **Q4: Results for larger LRs**
>
> ***Answer:*** Following the reviewer's request, we run experiments with several values of LR. We provide the results and discussion in Appendix K. We observe that for small learning rates, SGD stays in the reduction phase for more iterations, while increasing LR pushes SGD to enter gradual sharpening and EoS regimes, as expected. However, the transition between the phases always happens at the same point.
>
> **Q5: Derivation of new warm-up schedule.**
>
> ***Answer:*** Thank you for this valuable comment. Let us clarify the intuition behind the stepsize in the stochastic case. The  stepsize dictated by theory is $\eta_k = \frac{1}{10H\_0 + 20H\_1(f\_{S\_k}(w\_k)-f\_{S\_k}^*)}$, where $f\_{S_k}$ is the current stochastic loss.
>
> One can show that the theoretical guarantees by using $\eta_k$ are similar to
>
> $\tilde{\eta}\_k = \frac{1}{\max\\{10H\_0, 20H\_1(f\_{S\_k}(w\_k)-f\_{S\_k}^*)\\} } $
>
> $= \frac{1}{10H\_0}\cdot \frac{1}{\max\left\\{1,\frac{f\_{S\_k}(w\_k) - f\_{S\_k}^*}{H\_0/2H\_1} \right\\}}$ up to numerical constants.
>
> In most cases, the minimum of the stochastic loss can be well approximated by 0. Therefore, we have $\tilde{\eta}\_k  = \frac{\eta}{\min\left\\{1, \frac{f\_{S\_k}(w\_k)}{C}\right\\}},$ where $C \approx \frac{H\_0}{H\_1}$ and $\eta \approx \frac{1}{H\_0}$.

---

### Official Review · Reviewer_MWmf · 2025-10-30

**Soundness:** 3
**Presentation:** 1
**Contribution:** 3
**Rating:** 4
**Confidence:** 3

**Summary:**

This paper investigates why learning-rate warm-up is so effective in large-scale deep learning and proposes a new theoretical explanation based on a novel $(H_{0},H_{1})$-smoothness framework. Unlike the traditional $(L_{0},L_{1})$ condition, which ties curvature to gradient norm and contradicts empirical behavior in early training, the $(H_{0},H_{1})$ condition bounds curvature by the loss suboptimality, matching observed linear relations between curvature and loss. Building on this, the authors derive a theoretically motivated warm-up schedule and compare it against the standard linear warm-up across experiments on language models and vision tasks (ResNet50, ViT-Tiny). Their results show that both linear and $(H_{0},H_{1})$ warm-up improve convergence over no warm-up, with the proposed schedule performing competitively while offering theoretical justification.

**Strengths:**

1. **Novel theoretical contribution**: The introduction of $(H_0,H_1)$-smoothness provides a fresh and insightful proxy for curvature, offering a more accurate explanation of warm-up dynamics than prior $(L_0,L_1)$-based analyses.

2. **Balanced theoretical and empirical support**: The paper combines rigorous convergence proofs with clear empirical validation on both language and vision benchmarks, lending credibility to the theoretical claims.

3. **Clear and well-structured presentation**: The exposition is logically coherent and easy to follow, making complex theoretical ideas accessible and enhancing the overall readability of the work.

**Weaknesses:**

1. **Severe formatting issues**: The manuscript does not comply with the official ICLR template. In particular, it uses an incorrect font throughout and shows evidence of space compression (e.g., between Figures 3 and 4), which detracts from professionalism and readability.
2. **Limited theoretical scope**: While the $(H_0,H_1)$-smoothness analysis shows that warm-up accelerates convergence, it does not establish that warm-up leads to a better final convergence outcome. In principle, similar effects of convergence could be obtained by simply extending training without warm-up. A stronger illustration for the necessity of warm-up still relies on arguments about training instability, which are not captured by this framework.
3. **Missed validation opportunity**: The proposed theory enables regression of $f^\ast$ from empirical training trajectories, yet the paper does not attempt such regression or verify the recovered $f^\ast$ against the true optimum. Including this check would provide a valuable validation of the framework’s practical accuracy.

**Questions:**

1. **Tightness of the bound**: Figures 1–2 display an almost perfect linear relationship between smoothness and loss suboptimality, whereas the $(H_0,H_1)$ framework only provides an upper bound. Is this near equality theoretically expected for certain model classes, or is it an artifact of the optimization trajectory and estimation method? Clarifying this would strengthen the interpretation of the empirical evidence.
2. **Applicability to attention layers**: Since attention is the fundamental component of Transformer models, does $(H_0,H_1)$-smoothness formally hold for a single attention block? Can the authors provide a proof or at least a theoretical justification beyond empirical observation?
3. **Combining proxies for curvature**: Curvature is inherently a second-order property. $(L_0,L_1)$-smoothness approximates it via a first-order proxy (gradients), while $(H_0,H_1)$-smoothness uses a zeroth-order proxy (loss values). Would a hybrid framework that leverages both first- and zeroth-order information yield a tighter or more general characterization of smoothness?

---

> ### Author Response · Authors · 2025-11-21
> **Rebuttals (part I)**
>
> We sincerely thank the reviewer for the constructive feedback and for the positive remarks on the well-structured and clear presentation. We also appreciate the recognition of the novelty of our theoretical analysis and the balanced theoretical and empirical support for our claims.
>
> We hope that the following clarifications will help re-evaluate our submission and potentially adjust the score.
>
> **W2: Limited theoretical scope:**
>
> ***Answer:***  For clarity, we break down our answer below.
>
> - **a.** We kindly disagree with this claim. We emphasize that our convergence analysis demonstrates asymptotic convergence to $\varepsilon$ error. Our theory can be applied as follows: If the transition from sharpness-reduction to progressive sharpening occurs at the loss level $f_{tr}$, then by setting $\varepsilon=f_{tr}-f^*$ in our analysis, we obtain that GD equipped with our schedule reaches this transition point more quickly
>
> - **b.** Prior studies do not provide a quantitative way to measure or explain the stability issue; they mostly rely on empirical observations. Our condition helps to fill this gap. The underlying intuition behind stability is the following: Classical optimization theory tells us that the stepsize must remain below a value on the order of $1/L$, where $L$ is the smoothness constant, otherwise the method eventually diverges. In our setting, the smoothness constant is not fixed but depends on the current loss value, yet the same principle still applies. This means that the step size must be scaled inversely with the instantaneous smoothness, or alternatively chosen extremely small from the beginning, which is impractical. If neither is done, gradient descent will ultimately diverge and thus become unstable. Our condition, therefore, provides a concrete, quantitative explanation for why learning rate warm-up stabilizes training.
>
> **W3: Missed validation opportunity:**
>
> ***Answer:*** Thank you for this comment. Our theory predicts a linear dependency between the sharpness and loss value. Our experiments show that the predicted linear relation is tight at the beginning of training at the sharpness-reduction phase. Importantly, verifying this linear relation does not require knowledge of $f^*$.
>
> Recovering $f^*$, however, would require the condition to remain tight along the entire optimization trajectory up to convergence, as well as accurate estimates of the constants $H_0$ and $H_1$. In practice, such estimates are loose because our condition is tight only near initialization and becomes increasingly conservative in the later stages of training.
>
> **Q1: Tightness of the bound:**
>
> **Answer:** Understanding the tightness of our condition is an important yet challenging task. In our theoretical analysis, we demonstrate that such a condition is an upper bound for simple networks, but its tightness remains unclear. Therefore, we provide empirical evidence that our condition approximates the smoothness well at the beginning of training. Proving that our condition is completely tight theoretically seems to be a nearly impossible task. It would be at least as hard as showing that a certain smoothness constant obtained for a practical ML model is theoretically the best.
>
> **Q2: Applicability to attention layers:**
>
> ***Answer:*** Thank you for a valuable comment. Indeed, analyzing the condition for attention layers is an important question as it is the main block of transformer models. In the revised version of our work, we demonstrate that our condition can be extended for a simplified attention layer as well. We refer the reviewer to our new Section 3.1.2 for an exact statement and to Appendix D for its proof. Long story short, our condition is also applicable to transformers of only one attention layer, extending its generality.

---

> > ### Author Response · Authors · 2025-11-21
> > **Rebuttals (part II)**
> >
> > **Q3: Combining proxies for curvature:**
> >
> > ***Answer:*** This is a good point! Considering the smoothness condition $\\|\nabla^2f(w)\\|\le H_0 + L_1\\|\nabla f(w)\\| + H_1(f(w)-f^*)$ is an interesting question for future work. We believe it will still characterize the same class of functions since $(L_0, L_1)$-smooth functions are a subclass of $(H_0, H_1)$-smooth functions: adding the gradient norm to the $(H_0,H_1)$-condition will not change the class. Nonetheless, the condition might indeed be tighter w.r.t. the choice of constants $H\_0, L\_1, H\_1$.
> >
> > The expected theoretical stepsize would be $\eta\_k \sim \frac{\eta}{H\_0+L\_1\\|\nabla f(w)\\|+H\_1(f(w)-f^*)}$. We can approximate this stepsize as follows
> >
> > $$\eta_k \approx \frac{\eta}{\max\left[1,\frac{\\|\nabla f(w)\\|}{C\_1},\frac{f(w)-f^*}{C\_2}\right]}.$$
> >
> > In other words, such a hybrid condition performs clipping w.r.t both the gradient norm and function suboptimality. Since both learning rate warm-up and gradient clipping are essential in training large models, this condition would serve as a proxy for both of them. However, we expect that the theoretical analysis might become significantly more complex, as we need to track the evolution of both gradient norm and loss.

---

### Official Review · Reviewer_k5bA · 2025-11-01

**Soundness:** 3
**Presentation:** 4
**Contribution:** 2
**Rating:** 4
**Confidence:** 4

**Summary:**

The paper introduces a generalization of the $(L_0, L_1)$ smoothness, they call $(H_0, H_1)$-smoothness. They show that under that assumption warm-up is preferrable to fixed step size. They show that under some assumptions a few neural networks satisfy this property.

**Strengths:**

The paper is well written and visually pleasing. Addresses a long standing problem in a clean way. Proofs and results are clear.
I really appreciated reading this.

The math part is well done, results are proved nicely.

**Weaknesses:**

#### On the Tightness of your Condition

You prove that under your condition LR-warm up is optimal. It is when your condition is tight! Not in general. If on a Linear Shallow network I start from zero, it does satisfy the condition but the Hessian grows towards the solution, it does not go down, thus fixed step size = the maximal step size you would pick it better.

In general neural networks seem to show the phenomenon of progressive sharpening, which is conceptually the opposite of your condition. Can you please comment on this? Do you see progressive sharpening in the models you train?


#### Explaining Warm-Up?
However, I'm a little dubious that in general warm-up is for convergence purposes. I believe in non-convex cases and large scale ML systems it is for stability. I think it is necessary to comment on that.

This is not a strong ground for rejection, but I believe one needs to account for/deal with these other explanations. I think the paper is clean and beautiful, just maybe you can conjecture *what else* warm up is needed for.

For instance, when someone assumes progressive sharpening is happening, warm up is useful to constraint the model to less sharp areas of your landscape. Can you comment on this? Do you see this when applying your theory to neural networks? Do you see this in experiments?

I think my final grade will depend on how you address this.
Precisely, making some experiments about and reworking the limitations of your work in analyzing the full picture behing warm up are explicit.


Also, I don't think it is sensible to speak about balanced neural networks for practice, because those are the flatter ones within the parameter space. That said I believe your assumption is also satisfied at standard initialization of linear networks. Maybe you can comment on how $H_0$ needs to grow to be satisfied at standard initialization.

**Questions:**

See weaknesses.

To what extent is progressive sharpening allowed under your property? Until $H_0$ I guess?

---

> ### Author Response · Authors · 2025-11-21
> **Rebuttals (part I)**
>
> We sincerely thank the reviewer for the constructive feedback, as well as for the positive remarks on the quality of the presentation, the importance of the problem we address, and the clarity and strength of our theoretical results.
>
> We hope that the following clarifications will help re-evaluate our submission and potentially adjust the score.
>
> **W1: On the Tightness of your Condition**
>
> ***Answer:*** Thank you for this valuable comment. For clarity, we break down our answer below.
> - **a.** We would like to emphasize that if a shallow network is initialized at zero, then GD does not move, as the gradient is zero. For such networks, the initialization should be close to zero, but not exactly zero. If such a network is initialized sufficiently close to 0, then it will provably first enter a stage of flattening. More precisely, we calculated the time derivative of the spectral norm of the Hessian over the course of gradient flow for a simple model (outlined in Appendix K.1), which shows the following:
>
>     - Near the origin, the derivative of the Hessian norm is negative, thus we enter a flattening phase.
>
>     - Next, the model enters a transition regime in which the sign of the derivative of the Hessian norm seems to be unclear.
>
>     - Finally, the model moves to the well-known sharpening phase, where the sharpness increases.
>
>     This calculation aligns well with standard results discussed in [1] (see Section 3). While the third phase has been studied relatively extensively in the literature related to EoS, our analysis focuses on the first phase, which remains far less developed.
>
>     In Appendix K.2, we present experiments on training the same model with different learning rates. These clearly show that progressive sharpening happens in these models, but always after an initial flattening phase which takes a considerable portion of the total training.
>
> - **b.** Thus, our condition is not in contradiction with standard phenomena appearing during the training of deep neural networks, but actually complementary. It is, to our knowledge, the only known condition that quantifies the relationship between sharpness and loss value in the initial phase and can provably provide theoretical improvement for convergence using a warm-up schedule, compared to training with a fixed learning rate. The previous discussion and the results discussed in [1] reveal that this condition does capture the loss landscape to a reasonable extent at the beginning of training.
>
> - **c.** We do not claim that the theoretical learning-rate warm-up schedule is optimal. Our point is that, for the function class under consideration, it is asymptotically better than using a constant schedule. From a practical point of view, SGD equipped with our warm-up schedule achieves a lower loss after a fixed number of iterations than SGD with a constant schedule.
> This insight can also be applied in practice. If the transition from the sharpness-reduction phase to the progressive-sharpening phase occurs at a loss level $f_{tr}$, then by choosing $\varepsilon=f_{tr}-f^*$, our theory indicates that SGD reaches this transition point more quickly when using our warm-up schedule, while avoiding instabilities during the sharpness-reduction phase.
>
>     [1] Kalra, Dayal Singh, and Maissam Barkeshli. Why warmup the learning rate? underlying mechanisms and improvements, NeurIPS 2024.
>
> **W2: Explaining Warm-Up?**
>
> ***Answer:*** Thank you for these valuable comments. We address them as follows.
>
> - **a.** We acknowledge that one motivation for using learning rate warm-up is to improve training stability, and we explicitly mention this in the related works section. However, prior studies do not provide a quantitative way to measure or explain this stability. Our condition helps to fill this gap.
>
>     The underlying intuition is straightforward. Classical optimization theory tells us that the stepsize must remain below a value on the order of $1/L$, where $L$ is the smoothness constant, otherwise the method eventually diverges. In our setting, the smoothness constant is not fixed but depends on the current loss value, yet the same principle still applies. This means that the step size must be scaled inversely with the instantaneous smoothness, or alternatively chosen extremely small from the beginning, which is impractical. If neither is done, gradient descent will ultimately diverge and thus become unstable.
>
> - **b.** Since our goal is to characterize the sharpness-reduction phase that occurs before progressive sharpening, our theory does not aim to precisely describe the behavior that follows afterward.

---

> ### Author Response · Authors · 2025-11-21
> **Rebuttals (part II)**
>
> **W3: On balancedness regularization.**
>
> ***Answer:*** Indeed, out of all the possible re-parametrizations of the weight matrices, we have the smallest spectral norm for the Hessian in the balanced case. This assumption certainly helps us to prove our condition in many cases. If one would go for instance to the other extreme and allow any possible reparametrization of the weight matrices, the $(H_0,H_1$ condition cannot hold, as the sharpness can explode while the loss value remains constant. Most importantly, balancedness is a realistic assumption, because: i) the quantities $W\_i^T W_i - W\_{i+1} W\_{i+1}^T$ and $\\|W\_i\\|_F^2 -  \\|W\_{i+1}\\|\_F^2$ are known to be preserved over the course of gradient flow, for linear and non-linear networks respectevily, ii) Weights are usually initialized randomly with a small variance around $0$, thus the aforementioned quantities will be small over the course of the training (close to 0).
>
> A glimpse of the necessary scaling for $H_0$ can be seen in Equation (17). There, the closer the initialization is to $0$, the smaller $h$ becomes and the larger $H_0$ must be. This is in full accordance with our experiments in Appendix K.2: the closer we initialize to $0$, the higher the plot starts.
>
> **Q1: Progressive sharpening under $(H_0,H_1)$-condition**
>
> ***Answer:*** Our condition effectively captures the initial sharpness-reduction phase that precedes the gradual sharpening observed later in training. Empirically, this condition is tight in the early regime: our experiments show that sharpness decreases approximately linearly with the loss. Beyond this phase, however, the condition becomes noticeably looser, and it does not describe the progressive sharpening behavior accurately.

---

### Author Response · Authors · 2025-11-21
**General response**

We would like to thank all reviewers for their precious effort and time assessing our manuscript. Below, we provide some general feedback, while we also address reviewers' main points individually. We submit a revised version of our paper, where the changes we made appear in blue.

The main point of criticism seems to be the incompatibility of our theory with a bulk of research on progressive sharpening and edge of stability (EOS). This is a fair point, and diving deeper into this line of research really helped us to refine our approach. Our new experiments (and a theoretical calculation), detailed in Appendix K, suggest that progressive sharpening does happen in all the models we look at, just at a later stage, while the initial stage is always a stage of progressive flattening. Interestingly, the turning point between flattening and sharpening does not seem to depend on the learning rate. The duration of the initial flattening can depend on the initialization, but it takes about 20% of the training from the loss drop perspective. Our theory applies exactly at this initial stage. See also the discussion we added at the end of Section 3 of the main paper.

Another major addition we made is analysis for one-layer transformers for in-context learning (Section 3.1.2 and Appendix D for proofs). This is a result we obtained recently, and since Reviewer MWmf asked about it, we thought that it would be a good addition to our revised version.

Thanks again for your hard work!

---

### Meta-Review · Area_Chair_v63g · 2025-12-26

**Summary:**

The paper aims to close the gap between theory and practice of warm-up by introducing the $(H_0, H_1)$ condition under which it is provably optimal (for training speed) to increase learning rate in the initial phase of training.

The key concern was raised by Reviewers  k5bA and ryKS. Research on Edge of Stability and Break-Even Point shows that sharpness in the initial phase of training can actually increase (progressive sharpening). Authors responded that “Our condition effectively captures the initial sharpness-reduction phase that precedes the gradual sharpening observed later in training.” My understanding is that in the experiments, due to initialization, there was no progressive sharpening observed, but this means the experiments were not fully representative of the practical setting where sharpness often increases in the early phase of training.

The connection between progressive sharpening and warm-up, while partially addressed by the Authors during discussion phase, should be more thoroughly addressed before accepting the work. Therefore at this stage I am recommending rejecting the work.

**Reviewer Concerns:**

Reviewers k5bA and ryKS noted that curvature often increases during training, which seems to contradict the $(H_0, H_1)$ condition. More precisely, research shows that sharpness can increase or decrease, depending on the Hessian at initialization.

It was also remarked by ryKS, there is existing research by Gilmer et al on the effect of warming up on curvature, which should be more clearly discussed.

It was also noted by MWmf that scope is limited to analysing training speed alone while warm-up is known to improve generalization.

These key comments were not fully addressed in my opinion during the rebuttal phase.

**Reviewer Scores:**

Two Reviewers voted to weakly reject the paper, while one voted for weak acceptance. Given that the key comments were not fully addressed in my opinion during the rebuttal phase, I do see likely that Reviewers would change significantly their scores.

---

### Decision · Program_Chairs · 2026-01-26

Reject